# Functional genomics of lipid metabolism in the oleaginous yeast *Rhodosporidium toruloides*

**Samuel T Coradetti[1†], Dominic Pinel[2†], Gina M Geiselman[2], Masakazu Ito[2], Stephen J Mondo[3], Morgann C Reilly[4,5], Ya-Fang Cheng[2], Stefan Bauer[2], Igor V Grigoriev[3,6,7], John M Gladden[4], Blake A Simmons[4,8], Rachel B Brem[1,6], Adam P Arkin[2,7,9]\*, Jeffrey M Skerker[2,8,9]\***

[1]The Buck Institute for Research on Aging, Novato, United States; [2]Energy Biosciences Institute, Berkeley, United States; [3]United States Department of Energy Joint Genome Institute, Walnut Creek, United States; [4]Joint BioEnergy Institute, Emeryville, United States; [5]Chemical and Biological Processes Development Group, Pacific Northwest National Laboratory, Richland, United States; [6]Department of Plant and Microbial Biology, University of California, Berkeley, Berkeley, United States; [7]Environmental Genomics and Systems Biology Division, Lawrence Berkeley National Laboratory, Berkeley, United States; [8]Biological Systems and Engineering Division, Lawrence Berkeley National Laboratory, Berkeley, United States; [9]Department of Bioengineering, University of California, Berkeley, Berkeley, United States

**\*For correspondence:**
aparkin@lbl.gov (APA);
skerker@berkeley.edu (JMS)

[†]These authors contributed equally to this work

**Competing interests:** The authors declare that no competing interests exist.

**Abstract** The basidiomycete yeast *Rhodosporidium toruloides* (also known as *Rhodotorula toruloides*) accumulates high concentrations of lipids and carotenoids from diverse carbon sources. It has great potential as a model for the cellular biology of lipid droplets and for sustainable chemical production. We developed a method for high-throughput genetics (RB-TDNAseq), using sequence-barcoded *Agrobacterium tumefaciens* T-DNA insertions. We identified 1,337 putative essential genes with low T-DNA insertion rates. We functionally profiled genes required for fatty acid catabolism and lipid accumulation, validating results with 35 targeted deletion strains. We identified a high-confidence set of 150 genes affecting lipid accumulation, including genes with predicted function in signaling cascades, gene expression, protein modification and vesicular trafficking, autophagy, amino acid synthesis and tRNA modification, and genes of unknown function. These results greatly advance our understanding of lipid metabolism in this oleaginous species and demonstrate a general approach for barcoded mutagenesis that should enable functional genomics in diverse fungi.
DOI: https://doi.org/10.7554/eLife.32110.001

## Introduction

*Rhodosporidium toruloides* (also known as *Rhodotorula toruloides* [*Wang et al., 2015*]) is a basidiomycete yeast (subdivision Pucciniomycotina). *Rhodotorula/Rhodosporidium* species are widely distributed in the phyllosphere and diverse soils (*Rosa and Peter, 2006*; *Sláviková et al., 2009*; *Butinar et al., 2005*; *Pulschen et al., 2015*). They accumulate high concentrations of carotenoid pigments (*Mata-Gómez et al., 2014*; *Lee et al., 2014*), giving their colonies a distinctive orange, red, or pink hue. When *R. toruloides* is cultured under nitrogen (*Zhu et al., 2012*), sulfur (*Wu et al.,*

**eLife digest** The fungus *Rhodosporidium toruloides* can grow on substances extracted from plant matter that is inedible to humans such as corn stalks, wood pulp, and grasses. Under some growth conditions, the fungus can accumulate massive stores of hydrocarbon-rich fats and pigments. A community of scientists and engineers has begun genetically modifying *R. toruloides* to convert these naturally produced fats and pigments into fuels, chemicals and medicines. These could form sustainable replacements for products made from petroleum or harvested from threatened animal and plant species.

Fungi, plants, animals and other eukaryotes store fat in specialized compartments called lipid droplets. The genes that control the metabolism – the production, use and storage – of fat in lipid bodies have been studied in certain eukaryotes, including species of yeast. However, *R. toruloides* is only distantly related to the most well-studied of these species. This means that we cannot be certain that a gene will play the same role in *R. toruloides* as in those species.

To assemble the most comprehensive list possible of the genes in *R. toruloides* that affect the production, use, or storage of fat in lipid bodies, Coradetti, Pinel et al. constructed a population of hundreds of thousands of mutant fungal strains, each with its own unique DNA 'barcode'. The effects that mutations in over 6,000 genes had on growth and fat accumulation in these fungi were measured simultaneously in several experiments. This general approach is not new, but technical limitations had, until now, restricted its use in fungi to a few species.

Coradetti, Pinel et al. identified hundreds of genes that affected the ability of *R. toruloides* to metabolise fat. Many of these genes were related to genes with known roles in fat metabolism in other eukaryotes. Other genes are involved in different cell processes, such as the recycling of waste products in the cell. Their identification adds weight to the view that the links between these cellular processes and fat metabolism are deep and widespread amongst eukaryotes. Finally, some of the genes identified by Coradetti, Pinel et al. are not closely related to any well-studied genes. Further study of these genes could help us to understand why *R. toruloides* can accumulate much larger amounts of fat than most other fungi.

The methods developed by Coradetti, Pinel et al. should be possible to implement in many species of fungi. As a result these techniques may eventually contribute to the development of new treatments for human fungal diseases, the protection of important food crops, and a deeper understanding of the roles various fungi play in the broader ecosystem.

DOI: https://doi.org/10.7554/eLife.32110.002

*2011*), or phosphorus (*Wu et al., 2010*) limitation, it can accumulate as much as 70% of cellular biomass as lipids (*Wiebe et al., 2012*), primarily as triacylglycerides (TAG).

Eukaryotes accumulate neutral lipids in complex, dynamic organelles called lipid droplets. Lipid droplets emerge from the endoplasmic reticulum (ER) membrane as a core of TAG surrounded by sterol esters, a phospholipid monolayer derived from ER phospholipids, and a targeted ensemble of proteins mediating inter-organelle interaction, protein trafficking, cellular lipid trafficking and regulated carbon flux in and out of the lipid droplet (*Walther and Farese, 2012*; *Farese and Walther, 2009*; *Gao and Goodman, 2015*). Aberrant lipid droplet formation contributes to many human diseases (*Krahmer et al., 2013a*; *Welte, 2015*) and impacts cellular processes as diverse as autophagy (*Shpilka et al., 2015*) and mitosis (*Yang et al., 2016*). The propensity of *R. toruloides* to form large lipid droplets under a variety of conditions makes it an attractive platform to study conserved aspects of the cellular biology of these important organelles across diverse eukaryotes.

*Rhodosporidium toruloides* is also an attractive host for production of sustainable chemicals and fuels from low-cost lignocellulosic feedstocks. Wild isolates of *R. toruloides* can produce lipids and carotenoids from a wide variety of carbon sources including glucose (*Wiebe et al., 2012*), xylose (*Wiebe et al., 2012*), and acetate (*Huang et al., 2016*), as well as complex biomass hydrolysates (*Fei et al., 2016*). They are relatively tolerant to many forms of stress including osmotic stress (*Singh et al., 2016*) and growth-inhibiting compounds in biomass hydrolysates (*Hu et al., 2009*; *Kitahara et al., 2014*). *Rhodosporidium toruloides* has been engineered to produce lipid-derived bioproducts such as fatty alcohols (*Fillet et al., 2015*) and erucic acid (*Fillet et al., 2017*) from

synthetic pathways. To enable more efficient production of terpene-derived and lipid-derived chemicals, it has also been engineered for enhanced carotenoid (*Lee et al., 2016*) and lipid (*Zhang et al., 2016a*) production. These efforts, while promising, have for the most part employed strategies adapted from those demonstrated in evolutionarily distant species such as *Saccharomyces cerevisiae* and *Yarrowia lipolytica*. To truly tap the biosynthetic potential of *R. toruloides*, a better understanding of the unique aspects of its biosynthetic pathways, gene regulation and cellular biology will be required.

Recently, transcriptomic and proteomic analysis of *R. toruloides* in nitrogen limited conditions (*Zhu et al., 2012*) identified over 2,000 genes with altered transcript abundance and over 500 genes with altered protein abundance during lipid accumulation. These genes included many enzymes involved in the TCA cycle, a putative PYC1/MDH2/Malic Enzyme NADPH conversion cycle (*Wynn et al., 1999*), fatty acid synthesis, fatty acid beta-oxidation, nitrogen catabolite repression, assimilation and scavenging, autophagy, and protein turnover. Proteomics of isolated lipid droplets (*Zhu et al., 2015*) identified over 250 lipid droplet-associated proteins including fatty acid synthesis genes, several putative lipases, a homolog of the lipolysis-regulating protein perilipin (*Bickel et al., 2009*), vesicle trafficking proteins such as Rab GTPases and SNARE proteins, as well as several mitochondrial and peroxisomal proteins.

While these studies were unambiguous advances for the field, significant work remains to establish the genetic determinants of lipid accumulation in *R. toruloides*. Differential transcript or protein abundance under nitrogen limitation is suggestive of function in lipid accumulation, but transcriptional regulation and gene function are often poorly correlated in laboratory conditions (*Price et al., 2013*). Similarly, sequestration in the lipid droplet may help regulate availability of some proteins for functions not necessarily related to lipid metabolism (*Cermelli et al., 2006*). More direct functional data would help the *R. toruloides* community prioritize this extensive list of genes for more detailed study and identify additional genes not identifiable by proteomic and transcriptomic methods. Finally, these studies highlighted dozens of genes with no known function, and hundreds more with only limited functional predictions. A more functional approach may yield more insights into unique aspects of *R. toruloides* biology.

Fitness analysis of gene deletion or disruption mutants within pooled populations is a flexible, powerful approach for elucidating gene function. In these experiments the relative growth rate of thousands of mutant strains are simultaneously measured by tracking the relative abundance of unique sequence identifiers for each mutant. These identifying sequences could be short sequence 'barcodes' inserted into targeted deletion mutants (*Giaever et al., 2002*), or genomic DNA flanking random transposon insertions (*Sassetti et al., 2001*). Early fitness experiments tracked strain abundance by hybridization of identifier sequences to DNA micro-arrays (*Giaever et al., 2002*; *Sassetti et al., 2001*). The advent of high-throughput sequencing and the development of broad host range transposons enabled more widespread use of fitness analysis in bacteria by direct sequencing of transposon insertion sites (TnSeq) (*Gawronski et al., 2009*; *Langridge et al., 2009*). The scalability and precision of TnSeq is improved when random sequence barcodes are added to each randomly integrated transposon (RB-TnSeq) (*Wetmore et al., 2015*). Once insertions sites have been mapped, strain abundance can then be more accurately measured with a simple, consistent PCR amplification of the barcode sequences from known priming sites (BarSeq).

TnSeq and RB-TnSeq have been employed extensively in bacteria (*Kwon et al., 2016*), and in a few eukaryotic species (*Michel et al., 2017*; *Pettitt et al., 2017*). Although some of the first barcoded fitness experiments were performed on mutant pools of *S. cerevisiae* (*Giaever et al., 2002*) and advances in TnSeq methods continue in that species (*Michel et al., 2017*), to date relatively low transformation efficiencies and a lack of functional transposon systems has limited the application of TnSeq and RB-TnSeq in most fungal species. Random mutagenesis of fungi by the bacterium *Agrobacterium tumefaciens* is one route to overcome these technical barriers. *Agrobacterium tumefaciens*, an opportunistic plant pathogen, has evolved an efficient system to transfer virulence genes into eukaryotic cells (*Gelvin, 2003*). Once in the host cell, these transfer DNAs (T-DNAs) integrate randomly into the genome (*Bundock et al., 2002*). *Agrobacterium tumefaciens*-mediated transformation (ATMT) has been used extensively in plants (*Gelvin, 2003*) and to transform diverse fungi at high efficiency (*Bundock et al., 2002*; *Michielse et al., 2005*; *Walton et al., 2005*; *Kunitake et al., 2011*; *Sullivan et al., 2002*; *Blaise et al., 2007*). Recently, Esher et al. used ATMT followed by mutant selection and high-throughput sequencing to identify several mutants with altered cell wall

biosynthesis in the basidiomycete yeast *Cryptococcus neoformans* (*Zhang et al., 2016a*). The methods they employed were only viable for characterization of a small pool of highly enriched mutants, but they demonstrated an effective paradigm to bring high-throughput functional genomics to diverse fungi.

In this study, we demonstrate the construction of a randomly barcoded, random insertion library in *R. toruloides* by ATMT and its application for functional genomics (RB-TDNAseq). We report a list of 1,337 genes, including 36 unique to basidiomycetes, that were recalcitrant to T-DNA insertion, the first full genome survey of putatively essential genes in a basidiomycete fungus. We use our barcoded mutant library to explore fatty acid catabolism in *R. toruloides*, demonstrating its utility in rapidly assessing mutant phenotypes. We show that mitochondrial beta-oxidation is important for fatty acid utilization in this species and that some members of its expanded complement of peroxisomal acyl-CoA dehydrogenases are necessary for growth on different fatty acids, suggesting substrate specificity or conditional optimality for each enzyme. We investigate perturbed lipid accumulation in the mutant pool by fractionation of the population by buoyancy and fluorescence activated cell sorting. We identify 150 genes with significant roles in lipid accumulation, notably genes involved in signaling cascades (28 genes), gene expression (15 genes), protein modification or trafficking (15 genes), ubiquitination or proteolysis (nine genes), autophagy (nine genes), and amino acid synthesis (eight genes). We also find evidence that tRNA modification affects lipid accumulation in *R. toruloides*, identifying five genes with likely roles in thiolation of tRNA wobble residues. These results significantly advance our understanding of lipid metabolism in *R toruloides*; identify key biological processes that should be explored and optimized in any oleaginous yeast engineered for lipid production; support emerging evidence of deep connections between lipid droplet dynamics, vesicular trafficking, and protein sorting; and demonstrate a general approach for barcoded mutagenesis that should enable functional genomics in a wide variety of fungal species.

## Results

### A functional genomics platform for *R. toruloides*

To enable functional genomics in *R. toruloides* IFO 0880, we first improved the existing genome assembly and annotation (*Zhang et al., 2016a*) using a combination of long-read PacBio sequencing for a more complete de novo assembly, a more comprehensive informatics approach for gene model predictions and functional annotation, and manual refinement of those models using evidence from mRNA sequencing (Genbank accession LCTV02000000), also available at the Mycocosm genome portal (*Grigoriev et al., 2014*) (see Appendix 1 for details). Summary tables of gene IDs, predicted functions, and probable orthologs in other systems are included in *Supplementary file 1*. For brevity, we will refer to *R. toruloides* genes by the common name for their *Saccharomyces cerevisiae* orthologs (e.g. *MET2*) when such orthologous relationships are unambiguous. Otherwise, we will give the Mycocosm protein ID, e.g. *RTO4_12154* and *RTO4_14576* are both orthologs of *GPD1*.

Because no method existed for high-throughput genetics in *R. toruloides*, we adapted established protocols for mapping barcoded transposon insertions (RB-TnSeq) (*Wetmore et al., 2015*), to mapping barcoded T-DNA insertions introduced with *Agrobacterium tumefaciens*-mediated transformation (ATMT). We call this method RB-TDNAseq (*Figure 1A*). In brief, we generated a diverse library of binary ATMT plasmids bearing nourseothricin resistance cassettes with ~10 million unique 20 base-pair sequence 'barcodes' by efficient Type IIS restriction enzyme cloning (*Engler et al., 2008*), introduced the library into *A. tumefaciens* EHA105 by electroporation, then transformed *R. toruloides* with ATMT. Using a TnSeq-like protocol, we mapped the unique locations of 293,613 individual barcoded T-DNA insertions in the *R. toruloides* genome (see Appendix 1 for details). Once insertion sites were associated with their barcodes, pooled fitness experiments were performed using a simple, scalable BarSeq protocol as previously described (*Wetmore et al., 2015*).

Insertions were sufficiently well dispersed to map at least one T-DNA in 93% of nuclear genes, despite some local and fine-scale biases in insertion rates (see Appendix 1 for details). Insertion density in coding regions followed an approximately normal distribution (as expected for random integration) centered on nine inserts per thousand base pairs, except for a subpopulation of genes with fewer than two inserts/kb (*Figure 1B*). These very low-insertion genes were highly enriched for orthologs of genes reported as essential in *Aspergillus nidulans* (*Arnaud et al., 2012*), *Cryptococcus*

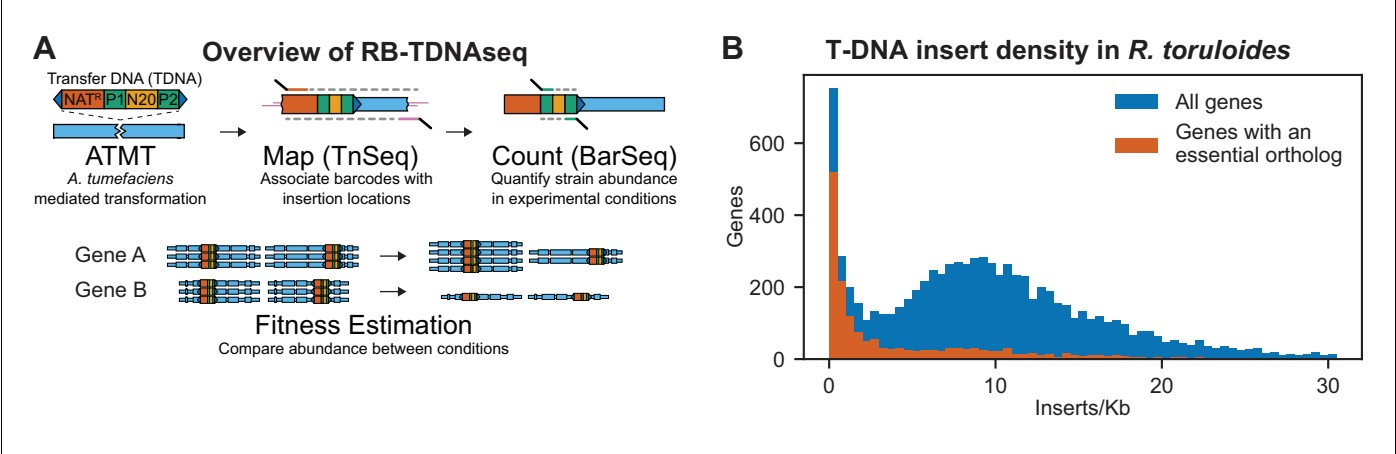

**Figure 1.** Overview of RB-TDNAseq and T-DNA insert density in *R. toruloides* coding regions. (**A**) General strategy of RB-TDNAseq. A library of binary plasmids bearing an antibiotic resistance cassette (NAT[R]) and a random 20 base-pair sequence 'barcode' (N20) flanked by specific priming sites (P1/P2) is introduced into a population of *A. tumefaciens* carrying a *vir* helper plasmid. *A. tumefaciens* efficiently transforms a T-DNA fragment into the target fungus (ATMT). NAT[R] colonies are then combined to make a mutant pool. T-DNA-genome junctions are sequenced by TnSeq, thereby associating barcodes with the location of the insertion (Map). The mutant pool is then cultured under specific conditions and the relative abundance of mutant strains is measured by sequencing a short, specific, PCR on the barcodes (BarSeq) and counting the occurrence of each sequence (Count). Finally, for each gene, count data is combined across all barcodes mapping to insertions in that gene to obtain a robust measure of relative fitness for strains bearing mutations in that gene (Fitness Estimation). (**B**) Histogram of insert density in coding regions (start codon to stop codon) for all genes, and genes with orthologs reported to be essential in *A. nidulans, C. neoformans, N. crassa, S. cerevisiae*, or *S. pombe*. The following figure supplements are available for *Figure 1*.

DOI: https://doi.org/10.7554/eLife.32110.003

The following figure supplements are available for figure 1:

**Figure supplement 1.** Schematic of TnSeq and BarSeq libraries generated using RB-TDNAseq.
DOI: https://doi.org/10.7554/eLife.32110.004

**Figure supplement 2.** Complexities of T-DNA insertions.
DOI: https://doi.org/10.7554/eLife.32110.005

**Figure supplement 3.** Observed biases in T-DNA insertion locations.
DOI: https://doi.org/10.7554/eLife.32110.006

neoformans (*Ianiri and Idnurm, 2015*), *Saccharomyces cerevisiae* (*Cherry et al., 2012*), or *Schizosaccharomyces pombe* (*Wood et al., 2012*), or for which only heterokaryons could be obtained in the *Neurospora crassa* deletion collection (*Colot et al., 2006*). We therefore infer that the majority of these genes recalcitrant to T-DNA insertion are likely essential in our library construction conditions, or at least that mutants for these genes have severely compromised growth. Based on the above criterion, we identified 1,337 probable essential genes, which we report in *Supplementary file 1*. This list includes over 400 genes not reported as essential in the above-mentioned model fungi and is enriched for genes with homologs implicated in mitochondrial respiratory chain I assembly and function, dynein complex, the Swr1 complex, and mRNA nonsense mediated decay. For a full list of GO term enrichments see *Supplementary file 1*. This list also includes 36 genes unique to basidiomycetes.

## Mapping biosynthetic pathways using RB-TDNAseq

Before investigating more novel aspects of *R. toruloides*' biology, we tested if RB-TDNAseq could be used to correctly identify gene function in well-conserved amino acid biosynthetic pathways. We cultured the mutant pool in defined medium (DM), consisting of glucose and yeast nitrogen base without amino acids and in DM supplemented with 'drop-out mix complete' (DOC), a mix of amino acids, adenine, uracil, p-aminobenzoic acid, and inositol. To establish if RB-TDNAseq could produce statistically robust results with minimal experimental replication, we recovered three independent starter cultures from frozen aliquots of the mutant pool and used each replicate to inoculate both

supplemented and non-supplemented cultures. We grew these cultures for seven generations and measured fitness across the mutant pool with BarSeq.

Secondary mutations are prevalent even in well-curated mutant collections (*Comyn et al., 2017*) and ATMT can introduce several types of confounding mutations (see Appendix 1 for details). To mitigate the influence of such mutations on our analysis, we adapted the established methods and software of Wetmore et al. (*Wetmore et al., 2015*; *Price et al., 2016*; *Cole et al., 2017*; *Sagawa et al., 2017*) for our BarSeq analysis. These algorithms compute a fitness score for each mutant strain as a $\log_2$ ratio of abundance in the experimental condition to abundance in a 'Time 0' sample from its seed culture. A composite fitness score (F) is then computed for each gene by combining multiple fitness scores from strains bearing insertions in that gene. A 'moderated T-statistic' calculated from the average and variance of strain fitness scores indicates the consistency of F across strains and experiments. See the Materials and methods section and (*Wetmore et al., 2015*) for more information on how these metrics are calculated. For more information on sequencing depth, behavior of T-statistics and detailed examples of how individual strain fitness scores contribute to F, see Appendix 1. All fitness scores and T-statistics (combined across biological replicates) are available in *Supplementary file 2* and online in a dynamic fitness browser, adapted from (*Price et al., 2016*): http://fungalfit.genomics.lbl.gov/.

Different aliquots of the mutant pool have subtly different starting compositions and experience stochastic variations in the length of lag phase as they recover from frozen stocks. Subtle variations in Illumina library preparation and sequencing for samples processed at different times may add further batch-specific biases to count data. For these reasons, direct comparisons of BarSeq counts between conditions tested in different batches and seeded from different starter cultures are not advisable. Expressing the data as F and T relative to Time 0 reduces it to a more portable format, allowing for comparisons of mutant fitness across conditions not necessarily tested in the same experiment. Given F and T in two different conditions ($F_{C1}$, $T_{C1}$ and $F_{C2}$, $T_{C2}$), we calculate relative fitness $F_{C1-C2} = F_{C1}-F_{C2}$ and relative T-statistics $T_{C1-C2} = (F_{C1}-F_{C2})/\sqrt{var(F_{C1})+var(F_{C2})}$.

Fitness scores for 6,558 genes in cultures grown on DM and DM supplemented with DOC are shown in *Figure 2A*. Mutants for 28 genes had fitness scores suggesting auxotrophy: fitness defects in non-supplemented media ($F_{DM} < -1$) with consistently different scores in supplemented versus non-supplemented media ($T_{DM-DOC} < -3$). When we grew the mutant pool in defined media with methionine or arginine supplementation (*Figure 2B*), the 28 auxotrophic mutants partitioned into 11 mutants rescued by methionine, eight mutants rescued by arginine, seven mutants rescued by neither amino acid and two mutants rescued by both amino acids. All of the identified methionine and arginine auxotrophic mutants have orthologous genes for which mutants are auxotrophic for methionine/cysteine or arginine, respectively, in *S. cerevisiae* or *A. nidulans*. Alternatively, when we hierarchically clustered the fitness scores for genes with $F < -1$ and $T < -3$ versus Time 0 in any supplementation condition (*Figure 2C*), the resulting clusters included twelve and nine mutants rescued by methionine and arginine respectively; this was a nearly complete recovery of genes with predicted functions in this pathway (shown in *Figure 2D–E* with additional discussion in Appendix 1). Based on these data, we chose $|T| > 3$ as a conservative threshold for consistent, reliable fitness scores in further BarSeq experiments.

## Fatty acid catabolism in *R. toruloides*

We next sought to understand how *R. toruloides* utilizes distinct fatty acids as growth substrates, as a window onto the complex lipid metabolism in this fungus. For this purpose, we used RB-TDNAseq to measure mutant fitness on three fatty acids as the sole carbon source: oleic acid (the most abundant fatty acid in *R. toruloides* [*Li et al., 2007*]) ricinoleic acid (a high-value fatty acid produced naturally in plants (*Dyer et al., 2008*) and synthetically in fungi [*Holic et al., 2012*]), and methyl ricinoleic acid (a ricinoleic acid derivative used in lactone production [*Endrizzi et al., 1996*]). A total of 129 genes had consistently low fitness scores on one or more fatty acids including genes implicated in beta-oxidation of fatty acids, gluconeogenesis, mitochondrial amino acid metabolism, and several other aspects of cellular metabolism and gene regulation (See *Figure 3—figure supplement 1* and Appendix 1 for a clustering analysis of fitness scores for these genes and *Supplemental file 2* for a complete list).

We were particularly interested in beta-oxidation of fatty acids in the peroxisome and mitochondria, as these pathways are critical for lipid homeostasis (*Kohlwein et al., 2013*; *Rambold et al.,*

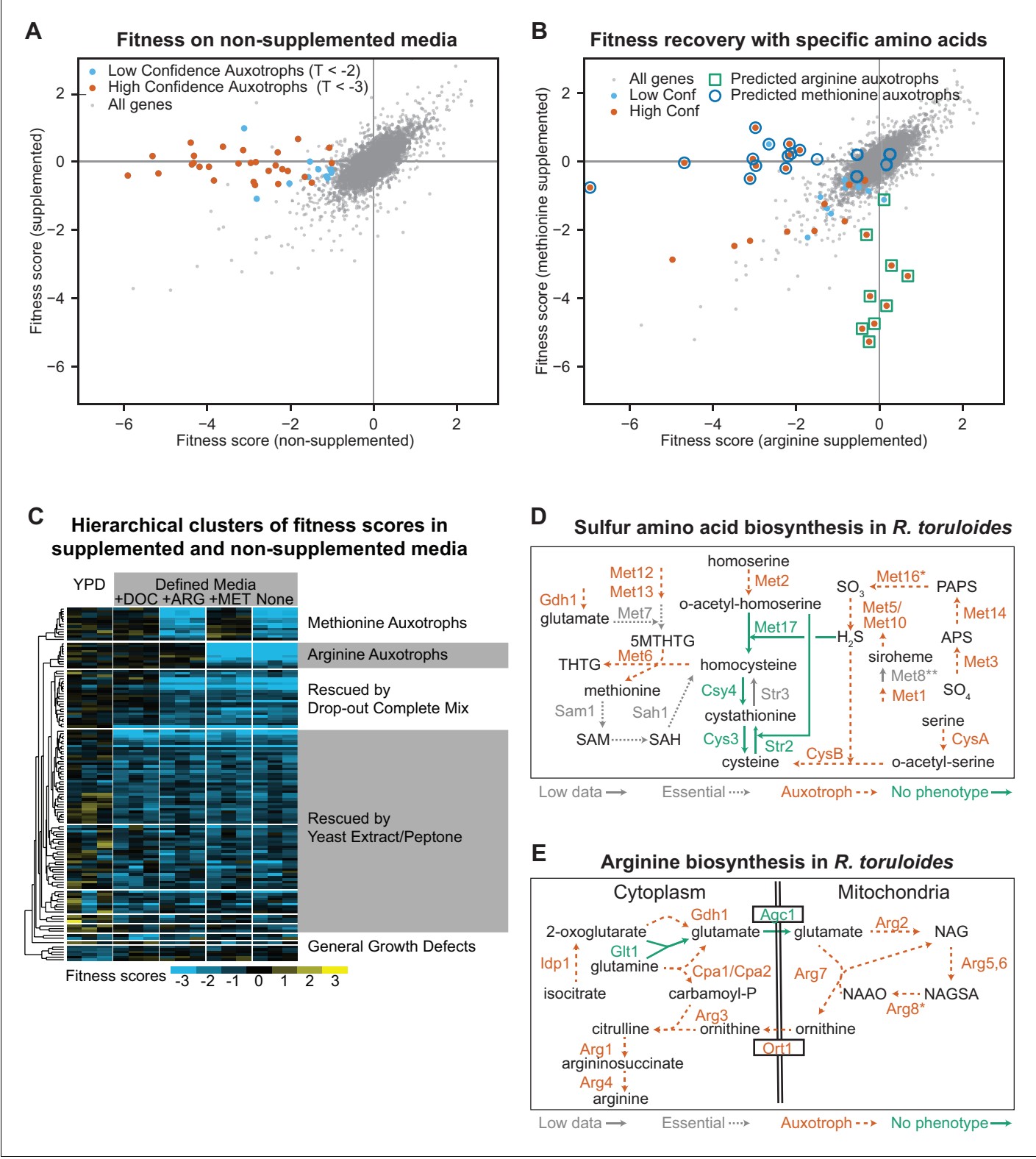

**Figure 2.** Confirmation of amino acid biosynthetic genes with high-throughput fitness experiments. (**A**) Fitness scores for 6,558 genes in media with and without amino acid supplementation (drop-out complete mix). Gene fitness scores are log ratios of final versus starting abundance averaged over multiple barcoded insertions per gene across three biological replicates. Genes that had consistently different enrichment scores between treatments (ΔF > 1, |T| statistic >3) are highlighted and represent genes for which mutant strains are auxotrophic for one or more amino acids, nucleotides, or vitamins present in the drop-out complete mixture. (**B**) Fitness scores in media supplemented with arginine or methionine. Highlighted genes are the

*Figure 2 continued on next page*

*Figure 2 continued*

same as highlighted in (A). Deletion strains for circled or boxed genes are auxotrophic for methionine or arginine, respectively, in *S. cerevisiae* or *A. nidulans*. See **Supplementary file 2** for full fitness data. (C) Hierarchical clusters of fitness scores in supplemented and non-supplemented media. Fitness scores for each biological replicate versus its Time 0 replicate for genes with a consistent fitness defect (F < −1, T < −3) in one or more of the following conditions: Yeast extract/Peptone/Dextrose media (YPD) or defined media (DM, composed of yeast nitrogen base plus glucose) with or without the following supplements: (+DOC), arginine (+ARG), or methionine (+MET). (D) Sulfur amino acid biosynthesis in *R. toruloides* as inferred from fitness experiments. CysA/CysB are named according to their *A. nidulans* orthologs, all others by orthologs in *S. cerevisiae*. Auxotrophic mutants had F < −1 in non-supplemented media (DM) and T < −3 in DM versus supplemented media (DOC). Multiple insertions were mapped in *STR3*, suggesting non-essentiality, but strain abundance was too low to estimate fitness in BarSeq data. *MET16* had fitness scores that clustered with the other auxotrophic mutants, but $T_{DM-DOC}$ was −2.7. **Fitness scores for insertions in *MET8* were not inconsistent with auxotrophy, but only two insertions were abundant enough to be tracked. 5MTHTG: 5-methyltetrahydropteroyltri-L-glutamate, THTG: tetrahydropteroyltri-L-glutamate, SAM: S-adenosyl-L-methionine, SAH: S-adenosyl-homocysteine, APS: adenylyl-sulfate, PAPS: 3'-phosphoadenylyl-sulfate. (E) Arginine biosynthesis in *R. toruloides* as inferred from fitness experiments. *ARG8* had fitness scores that clustered with the other auxotrophic mutants, but $T_{DM-DOC}$ was −2.9. NAG: N-acetylglutamate, NAGSA: N-acetylglutamate semialdehyde, NAAO: N-alpha-acetylornithine. The following figure supplements are available for *Figure 2*.

DOI: https://doi.org/10.7554/eLife.32110.007

The following figure supplements are available for figure 2:

**Figure supplement 1.** Barcode abundance in BarSeq experiments.
DOI: https://doi.org/10.7554/eLife.32110.008
**Figure supplement 2.** Contributions of individual strains to gene-level fitness scores.
DOI: https://doi.org/10.7554/eLife.32110.009
**Figure supplement 3.** Properties of T-statistics.
DOI: https://doi.org/10.7554/eLife.32110.010

2015), with major implications for both human health (**Houten et al., 2016**; **Waterham et al., 2016**) and metabolic engineering in fungi (**Dulermo and Nicaud, 2011**; **Beopoulos et al., 2014**). Fitness scores for *R. toruloides* genes homologous to enzymes with known roles in beta-oxidation of fatty acids are shown in **Figure 3A**. The localization for these enzymes is inferred mostly from observations in distantly related species, but orthologs of five enzymes localized to the predicted compartments in the basidiomycete yeast *Ustilago maydis* (**Camões et al., 2015**) adding some confidence to these predicted locations.

Mutants for mitochondrial enzymes had the most consistent fitness scores across all three fatty acids, whereas mutants for the peroxisomal enzymes and peroxins had more variable fitness scores among fatty acids. Mutants for seven peroxisomal beta-oxidation enzymes and three peroxins had different fitness scores on oleic acid versus ricinoleic acid and methylricinoleic acid (listed in Appendix 1, full fitness scores in **Supplementary file 2**), while 11 other predicted peroxisomal beta-oxidation enzymes had no consistent fitness scores at all. These results demonstrate how RB-TDNAseq can be used to rapidly identify condition-specific phenotypes among closely related members of a gene family. All together our data are consistent with a model of fatty acid beta-oxidation in *R. toruloides* in which diverse long-chain fatty acids are shortened in the peroxisome and a less structurally diverse set of short-chain fatty acids are oxidized to acetyl-CoA in the mitochondria (**Figure 3—figure supplement 2**).

To validate our fitness data on fatty acids, we made targeted deletion mutants for several predicted peroxisomal and mitochondrial proteins by homologous recombination into a non-homologous end joining deficient *YKU70Δ* strain (also known as *KU70*) (**Ninomiya et al., 2004**; **Zhang et al., 2016b**). We grew these mutant strains on oleic or ricinoleic acid media and compared their growth to the parental *YKU70Δ* strain in mid-log phase. Relative growth for the deletion strain for each gene is compared to its fitness scores in the BarSeq experiment in **Figure 3B** and **Figure 3C**. The *PEX7Δ* mutant had similar fitness defects on both fatty acids, but mutants for *RTO4_8673* (similar to *PEX11*) and *RTO4_14567* (similar to *H. sapiens ACAD11*), had stronger fitness defects on ricinoleic acid, and the mutant for acyl-CoA dehydrogenase *RTO4_8963* had stronger fitness defects on oleic acid as predicted from fitness scores. Over a 96 hr time course, the *RTO4_14567Δ* mutant failed to grow at all on ricinoleic acid, whereas the *RTO4_8963Δ* mutant and the *PEX11* homolog *RTO4_8673Δ* mutant had more subtle phenotypes, approaching the same final density of the *YKU70Δ* control strain after a longer growth phase (**Figure 3—figure supplement 3**).

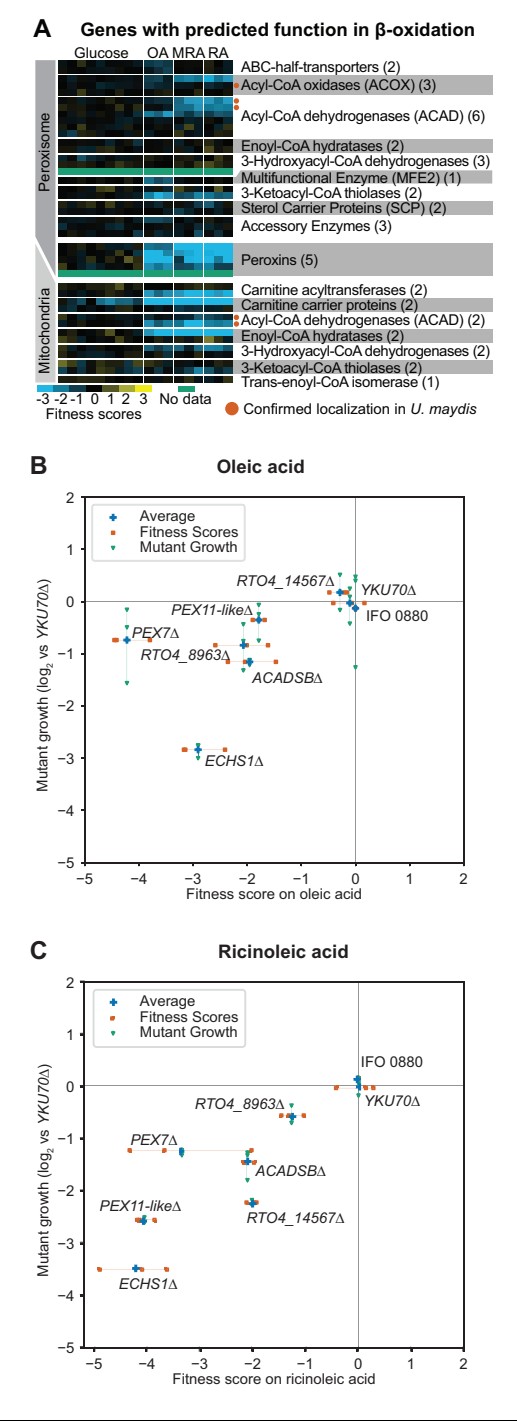

**Figure 3.** Genes with fitness defects on fatty acids. (**A**) Heatmap of fitness scores for *R. toruloides* genes with predicted roles in beta-oxidation of fatty acids. Enzyme classes and predicted locations were inferred from homologous proteins in *Ustilago maydis* as reported by Camões et al. (*Camões et al., 2015*). See *Supplementary file 2* for full fitness data. (**B**) Log$_2$ optical density ratio for single deletion mutants versus the *YKU70Δ* control strain at mid-log phase on 1% oleic acid as carbon source are plotted against the *Figure 3 continued on next page*

These data showed that BarSeq fitness scores were reliable predictors of significant growth defects for mutants in pure culture.

## Functional genomics of lipid accumulation in *R. toruloides*

To dissect the genetic basis of lipid accumulation in *R. toruloides*, we induced lipid accumulation by nitrogen limitation (*R. toruloides* lipid droplets visualized in *Figure 4A*), and used two measures of cellular lipid content to fractionate the mutant pool (*Figure 4B* and Appendix 1). We used the neutral-lipid stain BODIPY 493/503 (*Bozaquel-Morais et al., 2010*) and fluorescence activated cell sorting (FACS) to enrich populations with larger/more or smaller/fewer lipid droplets (*Terashima et al., 2015*). We also used buoyancy separation on sucrose gradients to enrich for populations with higher or lower total lipid content (*Eroglu and Melis, 2009*; *Kamisaka et al., 2006*; *Liu et al., 2015*). Because many mutations can affect cell buoyant density independent of lipid accumulation (*Novick et al., 1980*; *Bryan et al., 2010*), we also grew the mutant pool in rich media (YPD) and subjected it to sucrose gradient separation as a control for lipid-independent buoyancy phenotypes. For each pair of high and low lipid fractions, we then calculated an 'enrichment score', E, and T-statistic for each gene. E is analogous to our fitness scores based on growth, except that it is the log$_2$ ratio of abundance in the high lipid fraction to the low lipid fraction, whereas F is the log$_2$ ratio of final to initial abundance. Hierarchical clusters of enrichment scores for 271 genes for which mutants have consistently altered lipid accumulation ($|E| > 1$ and $|T| > 3$) are shown in *Figure 5A*. Enrichment scores and T-statistics for all 6,558 genes with sufficient BarSeq data are reported in *Supplementary file 2*.

To assess the reliability of these enrichment scores in predicting phenotypes for null mutants, we constructed 29 single gene deletion mutants by homologous recombination in a *YKU70Δ* strain of IFO 0880 and measured lipid accumulation by average BODIPY fluorescence for 10,000 cells from each strain using flow cytometry. *Figure 5B and C* show relative BODIPY signal for targeted deletion mutants versus the *YKU70Δ* parental strain (see Appendix 1 for more information on normalization and power analysis). When enrichment scores from both assays were strongly positive (LA1), we found that 7 of 8 deletion mutants had the expected phenotype (i.e. increased lipid accumulation). When only one assay yielded a strongly positive score (clusters LA2 and LA3),

*Figure 3 continued*

fitness scores for each gene from BarSeq experiments on 1% oleic acid. (C) Log₂ optical density ratio for single deletion mutants versus the *YKU70Δ* control strain at mid-log phase on 1% ricinoleic acid as carbon source are plotted against the fitness scores for mutants in each gene from BarSeq experiments on 1% ricinoleic acid. See *Supplementary file 2* for a statistical summary for all strains shown in (B) and (C), including P values and effect sizes. The following figure supplements are available for *Figure 3*.

DOI: https://doi.org/10.7554/eLife.32110.011

The following figure supplements are available for figure 3:

**Figure supplement 1.** K-means clusters of fitness scores for 129 genes for which mutants have specific fitness defects on fatty acids.

DOI: https://doi.org/10.7554/eLife.32110.012

**Figure supplement 2.** Model for beta-oxidation of fatty acids in *R. toruloides*.

DOI: https://doi.org/10.7554/eLife.32110.013

**Figure supplement 3.** Extended growth curves for deletion mutants on fatty acids.

DOI: https://doi.org/10.7554/eLife.32110.014

only 3 of 5 mutants had apparent increases in lipid content as measured by flow cytometry. Further, for the two mutants for genes in cluster LA3 with the greatest apparent increase in lipid content (*PMT4* and *RTO4_10302,* similar to *C. neoformans CMT1*) that measurement was likely an artifact of incomplete cell separation. Both mutants formed long chains of cells (see *Figure 7—figure supplement 1* for microscopy images), which would be analyzed as a single cell by our FACS assay. Genes in clusters LA4 and LA5 had conflicting enrichment scores between the two assays. Of three targeted deletion strains for genes in these clusters, only one (*CCC1Δ*) had a statistically significant phenotype, with decreased lipid accumulation. When the FACS assay gave a strongly negative score and there was no strong contrary buoyancy score (clusters LA6, LA7, and LA8), 11 of 13 mutants had reduced lipid accumulation. These data confirm that both separation techniques are fundamentally sound, though in isolation each method has a significant rate of false positives. In combination, the two assays identified a large set of high-confidence candidate genes with important roles in lipid accumulation.

## Diverse predicted functions for lipid accumulation mutants

We manually curated homology-based predicted functions for the 393 genes with consistent fitness or enrichment scores in this study (*Supplementary file 1*). An overview of predicted localizations and functions for genes we identified with roles in fatty acid utilization or lipid accumulation is shown in *Figure 6*, with more detail for mutants with increased and decreased lipid accumulation in *Tables 1* and *2*, respectively. Note that we have excluded genes for which only one enrichment technique indicated altered lipid accumulation from this analysis.

Mutants with increased lipid accumulation (cluster LA1, 56 genes) were most notably enriched for genes involved in signaling cascades, post-translational protein modification and trafficking, and in amino acid biosynthesis. Genes involved in signaling cascades included several homologs to G-proteins such as *RAS1* and mammalian *RAC1* and their effectors, as well as several kinases, indicating a complex signaling network regulating lipid accumulation. Genes involved in protein trafficking included P24 adapter proteins, suggesting they play an important role in delivering lipid-mobilizing genes to the lipid droplet or removing lipid biosynthesis genes from the endomembrane network. Mutants for several genes identified in our auxotrophy experiments also had increased lipid accumulation, most notably genes involved in sulfate assimilation for cysteine and methionine biosynthesis. Not all auxotrophic mutants had altered lipid accumulation, suggesting that arrested protein synthesis is not necessarily sufficient to increase lipid accumulation.

Mutants with decreased lipid accumulation (clusters LA6, LA7, and LA8, 94 genes) were most notably enriched for genes with roles in autophagy, protein phosphorylation, and tRNA-modifcation. Mutants in nine core components of autophagy were deficient for lipid accumulation, consistent with previous findings that chemical inhibition of autophagy reduced lipid accumulation in *Y. lipolytica* (*Qiao et al., 2015*). Mutants in several proteases and ubiquitin ligases also had reduced lipid accumulation, highlighting the importance of efficient recycling of cellular materials to refactor the cell for high lipid accumulation. Mutants in at least nine protein kinases, three phosphatases or their binding partners had reduced lipid accumulation; likely these genes mediate nutrient sensing cascades that stimulated lipid accumulation. Several genes with likely roles in thiolation of tRNA wobble residues had lower lipid accumulation. Though these mutants also had apparent buoyancy phenotypes on YPD, two deletion strains (*NCS6Δ* and *NCS2Δ*) had reduced lipid content in pure culture

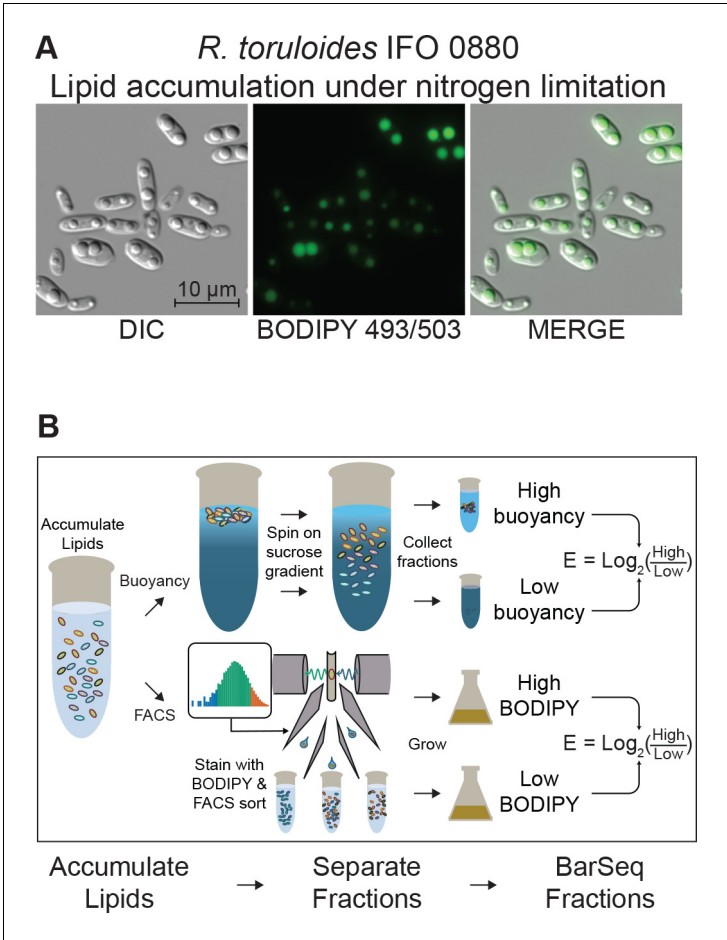

**Figure 4.** Detecting mutants with altered lipid accumulation. (**A**) Lipid accumulation in *R. toruloides* under nitrogen limitation. DIC microscopy of *R. toruloides* grown in low nitrogen media for 40 hr and stained with BODIPY 493/503 to label lipid droplets. (**B**) Two strategies to enrich populations for high or low TAG content cells. (Top) Buoyant density separation on sucrose gradients. Lipid accumulated cells are loaded onto a linear sucrose gradient and centrifuged. Cells settle at their neutral buoyancy, with the size of the low-density lipid droplet as the main driver of buoyancy differences. The gradient is then split into several fractions, and fractions representing the most and least buoyant 5–10% of the population, as well as a no-separation control are subjected to DNA extraction and strain quantification with BarSeq. For each gene an enrichment score is calculated as the $\log_2$ ratio of mutant abundance in the high buoyancy versus low buoyancy fractions. (Bottom) FACS sorting on BODIPY signal. Cells cultured in lipid accumulation conditions (limited nitrogen) are stained with BODIPY 493/503, then sorted in a FACS system. The 10% of the population with the highest and lowest BODIPY signal are sorted into enriched populations, as well as non-gated control. These small populations (10 million cells each) are then cultured for additional biomass and subjected to DNA extraction and strain quantification with BarSeq. For each gene, a FACS enrichment score is calculated as the $\log_2$ ratio of mutant abundance in the high BODIPY versus low BODIPY fractions. The following figure supplements are available for *Figure 4*.

DOI: https://doi.org/10.7554/eLife.32110.015

The following figure supplements are available for figure 4:

**Figure supplement 1.** Measuring lipid accumulation under nitrogen limitation.
DOI: https://doi.org/10.7554/eLife.32110.016

**Figure supplement 2.** Lipid accumulation and buoyancy changes under nitrogen limitation.
DOI: https://doi.org/10.7554/eLife.32110.017

(*Figure 5C*). They may play a role in regulating global carbon metabolism (*Laxman et al., 2013*). *RTO4_16381*, a distant homolog of *H. sapiens PLIN1* (perilipin), was also necessary for high lipid accumulation, consistent with its homolog's known roles in lipid body maintenance and regulation of

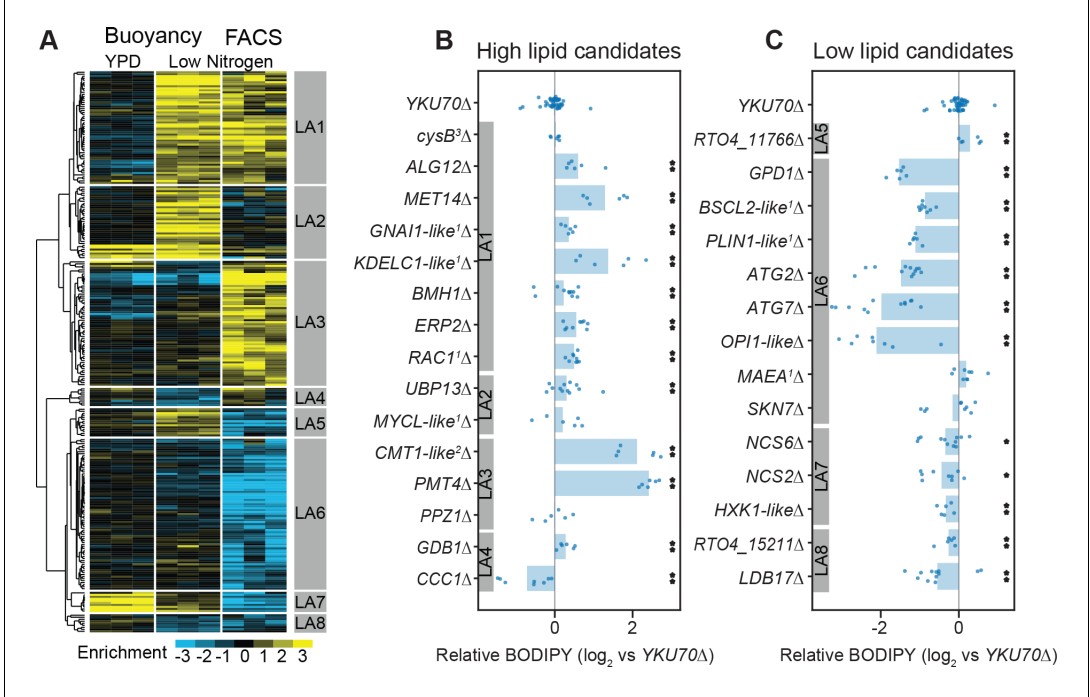

**Figure 5.** RB-TDNAseq on enriched populations identifies genes affecting lipid accumulation. (A) Hierarchical clusters of enrichment scores for 271 genes with consistent enrichment ($|E| > 1$, $|T| > 3$) in high/low fractions separated by buoyant density or FACS sorting of BODIPY stained cells after lipid accumulation on low nitrogen media. Enrichment scores for individual biological replicates (three per condition) were clustered in this analysis. Eight major clusters were identified (LA1-LA8). See *Supplementary file 2* for full enrichment data. (B and C) Relative BODIPY signal for deletion mutants. Points are the average BODIPY/cell for 10,000 cells from independent biological replicate cultures normalized to three control *YKU70Δ* cultures processed on the same day. Three biological replicates were processed for each strain in any given experiment and each strain was included in at least two experiments processed on different days (N ≥ 6). A statistical summary for all strains including N, P values, and effect sizes is included in *Supplementary file 2*. \*\*p<0.01, \*p<0.05 by homoscedastic T-test versus *YKU70Δ*. [1]Human homolog, [2]*C. neoformans* homolog, [3]*A. nidulans* homolog. The following figure supplements are available for *Figure 5*.

DOI: https://doi.org/10.7554/eLife.32110.018

The following figure supplement is available for figure 5:

**Figure supplement 1.** tRNA thiolation in *S. cerevisiae* versus lipid accumulation in *R. toruloides*.

DOI: https://doi.org/10.7554/eLife.32110.019

triglyceride hydrolysis (*Bickel et al., 2009*) and previous observations that it localized to lipid droplets in *R. toruloides* (*Zhu et al., 2015*).

## Diverse morphological phenotypes for lipid accumulation mutants

To further characterize the phenotypes of our lipid accumulation mutants, we performed differential interference contrast (DIC) and fluorescence microscopy. The mutants showed a variety of phenotypes with respect to both cellular and lipid droplet morphology. Eight examples are highlighted in *Figure 7*. While wild type cells most commonly had two lipid droplets of similar size, several high lipid accumulation mutants had qualitatively more cells with three or more lipid droplets (e.g. *MET14Δ*, *Figure 7*)) or cells with a single dominant droplet (e.g. *RAC1Δ*, *Figure 7*). *RAC1Δ* also had qualitatively larger, more spherical cells. A *KDELC-like Δ* mutant with increased lipid accumulation also showed a defect in cell separation likely reflective of combined defects in lipid accumulation, secretion, and cell wall/septum formation. All strains had a wide cell-to-cell variation in lipid droplet size, consistent with high variance in BODIPY intensity measured by flow cytometry (*Figure 4—figure supplement 2A*). Most low-lipid strains appeared morphologically similar to wild type with smaller lipid bodies (*Figure 7—figure supplement 1*). However, a *BSCL2-likeΔ* (seipin) mutant showed an even larger variation in droplet size than wild type, consistent with observations in *S. cerevisiae* mutants for the homolog *SEI1/FLD1* (*Fei et al., 2008*) and likely reflective of a conserved

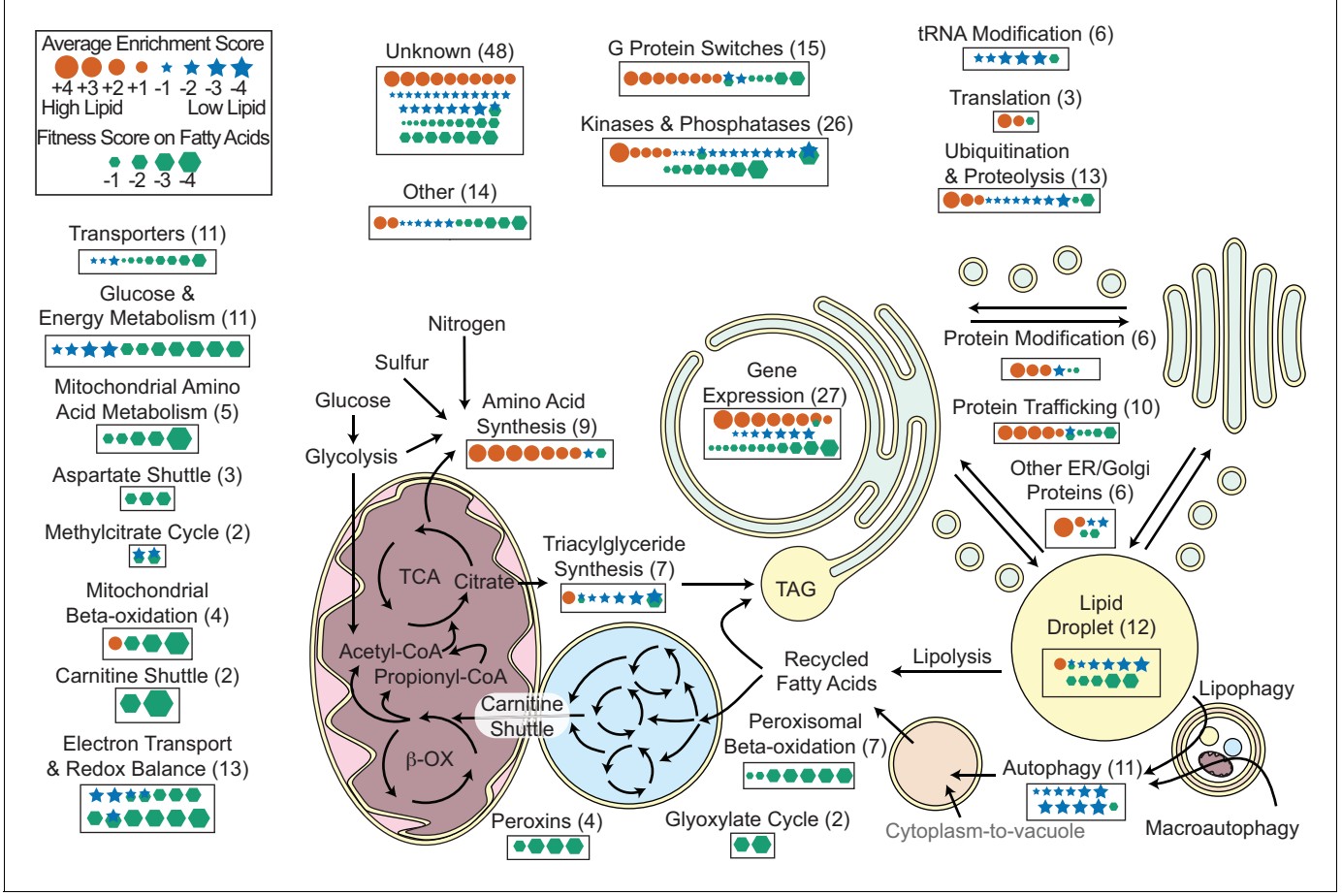

**Figure 6.** Overview of *R. toruloides* lipid metabolism. Key metabolic pathways and cellular functions mediating lipid metabolism as identified from fitness scores on fatty acid and enrichment scores from lipid accumulation screens. Fitness and/or enrichment scores for individual genes are depicted graphically by relative size of hexagonal, circular or star icons respectively. Only fitness scores for genes with consistent growth defects on at least one fatty acid (see *Supplementary file 2*) and enrichment scores from high confidence clusters (see *Figure 5* and *Supplementary file 2*) are shown. Enrichment scores were averaged between buoyancy and FACS experiments, except for genes with confounding enrichment scores in rich media conditions, for which only FACS data were averaged. Positive scores (orange circles) represent genes for which mutants have increased lipid accumulation. Negative fitness scores (blue stars) represent genes for which mutants have decreased lipid accumulation. Genes detected in proteomics of *R. toruloides* lipid droplets by Zhu et al. (*RAC1, GUT2, PLIN1, EGH1, RIP1, MGL2, AAT1, CIR2, MLS1*, and *RTO4_8963*) or found in lipid droplets of many organisms (*DGA1* and *BSCL2*) (see *Supplementary file 5*) are depicted under 'Lipid Droplet' and also their molecular functions, e.g. 'G Protein Switches' for *RAC1*. The following figure supplements are available for *Figure 6*.
DOI: https://doi.org/10.7554/eLife.32110.020

The following figure supplement is available for figure 6:

**Figure supplement 1.** Genes directly effecting TAG biosynthesis in *R. toruloides*.
DOI: https://doi.org/10.7554/eLife.32110.021

function in lipid droplet formation and efficient delivery of lipid biosynthetic proteins to the growing lipid droplet (*Wang et al., 2016*; *Pagac et al., 2016*; *Salo et al., 2016*). Autophagy mutants (*ATG2Δ*) had the most uniformly small lipid droplets in elongated cells with enlarged vacuoles. Overall, the morphological phenotypes we observed in *R. toruloides* are similar to a number of previous microscopic screens for altered lipid accumulation in diverse eukaryotes (*Fei et al., 2008*; *Szymanski et al., 2007*; *Guo et al., 2008*; *Zehmer et al., 2009*; *Ashrafi et al., 2003*).

**Table 1.** Predicted gene function: Mutants with increased lipid accumulation.

Predicted functions for genes for which mutants were high-confidence candidates for increased lipid accumulation (enrichment scores clustered in LA1, *Figure 5*).

| | Gene ID | Short name | Annotation from | Description | Enrichment BD | FACS |
|---|---|---|---|---|---|---|
| **G Protein Switches** | | | | | | |
| * | RTO4_15883 | RAS1 | S. cerevisiae | GTPase | 2.0 | 2.3 |
| | RTO4_14088 | RAC1 | H. sapiens | GTPase | 2.0 | 0.9 |
| * | RTO4_16215 | GNAI1-like | H. sapiens | GTPase | 1.6 | 1.0 |
| | RTO4_11402 | gapA | A. nidulans | GTPase-activating protein | 0.6 | 1.4 |
| | RTO4_13336 | RIC8A | H. sapiens | Guanine nucleotide exchange factor | 1.3 | 1.4 |
| | RTO4_16170 | sif-like | D. melanogaster | Guanine nucleotide exchange factor | 1.5 | 0.9 |
| | RTO4_16644 | BMH1 | S. cerevisiae | 14-3-3 protein | 1.3 | 2.2 |
| | RTO4_16068 | BMH1 | S. cerevisiae | 14-3-3 protein | 0.7 | 1.2 |
| **Kinases and Phosphatases** | | | | | | |
| | RTO4_13246 | CNA1 | S. cerevisiae | Phosphatase (Calcineurin catalytic subunit) | 0.8 | 1.2 |
| | RTO4_11675 | CNB1 | S. cerevisiae | Phosphatase (Calcineurin regulatory subunit) | 1.1 | 1.2 |
| | RTO4_11667 | PTC1 | S. cerevisiae | Phosphatase | 0.9 | 1.2 |
| | RTO4_10638 | CLA4 | S. cerevisiae | Kinase | 3.4 | 4.5 |
| * | RTO4_16605 | TPK1 | S. cerevisiae | Kinase | 1.1 | 0.5 |
| **Gene Expresssion** | | | | | | |
| | RTO4_10333 | SET1 | S. cerevisiae | Chromatin modifying | 3.0 | 1.1 |
| | RTO4_10279 | BRE2 | S. cerevisiae | Chromatin modifying | 2.5 | 1.0 |
| | RTO4_12689 | SPP1 | S. cerevisiae | Chromatin modifying | 2.0 | 1.3 |
| | RTO4_15412 | RCO1 | S. cerevisiae | Chromatin modifying | 3.5 | 1.6 |
| | RTO4_10209 | MIT1-like | S. cerevisiae | Transcripition factor | 1.4 | 0.3 |
| | RTO4_14550 | CYC8 | S. cerevisiae | Transcription factor | 3.7 | 3.8 |
| | RTO4_10274 | SKN7-like | S. cerevisiae | Transcription factor | 2.2 | 1.5 |
| | RTO4_13346 | CBC2 | S. cerevisiae | RNA splicing factor | 1.6 | 1.2 |
| **Protein Modification** | | | | | | |
| | RTO4_11272 | ALG12 | S. cerevisiae | Alpha-1,6-mannosyltransferase | 3.5 | 1.7 |
| | RTO4_14881 | CAP10-like | C. neoformans | Xylosyltransferase | 1.5 | 2.0 |
| | RTO4_16598 | LARGE1 | H. sapiens | N-acetylglucosaminyltransferase-like protein | 1.8 | 1.3 |
| **Protein Trafficking** | | | | | | |
| | RTO4_12145 | ERP1 | S. cerevisiae | COPII cargo adapter protein (p24 family) | 2.4 | 2.7 |
| | RTO4_16731 | ERP2 | S. cerevisiae | COPII cargo adapter protein (p24 family) | 1.7 | 2.0 |
| | RTO4_12521 | EMP24 | S. cerevisiae | COPII cargo adapter protein (p24 family) | 1.9 | 2.4 |
| | RTO4_14054 | BST1 | S. cerevisiae | GPI inositol-deacylase | 1.5 | 0.2 |
| * | RTO4_15883 | RAS1 | S. cerevisiae | GTPase | 2.0 | 2.3 |
| **Other ER/Golgi Proteins** | | | | | | |
| | RTO4_10371 | KDELC1-like | H. sapiens | Endoplasmic reticulum protein EP58 | 3.1 | 6.0 |

*Table 1 continued on next page*

*Table 1 continued*

| | Gene ID | Short name | Annotation from | Description | Enrichment BD | FACS |
|---|---|---|---|---|---|---|
| | RTO4_15763 | | | SH3 Domain-containing ER Protein | 1.0 | 1.5 |
| **Amino Acid Biosynthesis** | | | | | | |
| | RTO4_11050 | MET1 | S. cerevisiae | Uroporphyrinogen III transmethylase | 3.8 | 2.0 |
| | RTO4_8744 | MET5 | S. cerevisiae | Sulfite reductase | 4.4 | 2.1 |
| § | RTO4_10374 | MET10 | S. cerevisiae | Sulfite reductase | 2.5 | 1.3 |
| | RTO4_8709 | MET14 | S. cerevisiae | Adenylylsulfate kinase | 4.1 | 1.1 |
| | RTO4_11741 | MET16 | S. cerevisiae | Phosphoadenosine phosphosulfate reductase | 1.7 | 1.1 |
| | RTO4_12031 | cysB | A. nidulans | Cysteine synthase A | 3.3 | 2.1 |
| * | RTO4_16196 | ARG1 | S. cerevisiae | Argininosuccinate synthase | 1.3 | 1.8 |
| **Translation** | | | | | | |
| | RTO4_12273 | MRN1 | S. cerevisiae | RNA-binding protein | 2.5 | 1.6 |
| | RTO4_8595 | EIF4E2 | H. sapiens | Translation initiation factor | 2.0 | 0.5 |
| **Ubiquitination and Proteolysis** | | | | | | |
| | RTO4_11150 | Mub1-like | S. cerevisiae | Ubiquitin ligase complex member | 3.8 | 2.0 |
| | RTO4_15576 | CDC4 | S. cerevisiae | Ubiquitin ligase complex member | 1.7 | 1.8 |
| **Triacylglyceride Synthesis** | | | | | | |
| † | RTO4_8972 | NDE1 | S. cerevisiae | NADH dehydrogenase | 1.6 | 1.9 |
| **Lipid Droplet Associated** | | | | | | |
| | RTO4_14088 | RAC1 | H. sapiens | GTPase | 2.0 | 0.9 |
| **Mitochondrial Beta-oxidation** | | | | | | |
| | RTO4_16284 | HSD17B10 | H. sapiens | 3-hydroxyacyl-CoA dehydrogenase | 1.6 | 0.5 |
| **Other** | | | | | | |
| | RTO4_12175 | mesA | A. nidulans | Myosin binding protein | 1.3 | 1.8 |
| | RTO4_8401 | SHE4 | S. cerevisiae | Transmembrane protein involved in cell polarity | 1.0 | 1.3 |
| **Unknown Function** | | | | | | |
| | RTO4_16524 | | | Protein of unknown function | 3.1 | 1.9 |
| | RTO4_11613 | | | Protein of unknown function | 2.5 | 1.7 |
| | RTO4_12505 | | | Protein of unknown function | 2.1 | 2.1 |
| | RTO4_13512 | | | Protein of unknown function | 1.5 | 1.9 |
| | RTO4_10805 | | | Protein of unknown function | 1.2 | 1.8 |
| | RTO4_15251 | | | Protein of unknown function | 1.6 | 1.3 |
| | RTO4_15358 | | | Protein of unknown function | 2.0 | 0.5 |
| | RTO4_13513 | | | Protein of unknown function | 1.3 | 1.2 |
| | RTO4_12461 | | | Protein of unknown function | 1.5 | 0.8 |
| | RTO4_13351 | | | Protein of unknown function | 1.2 | 1.0 |

Cellular processes grouped as in **Figure 6**. BD: Enrichment score from buoyant density separation. FACS: Enrichment score from fluorescence activated cell sorting.

Protein abundance under nitrogen limitation: * increased; † increased 10-fold or more; ‡ decreased; § decreased 10-fold or more (**Zhu et al., 2012**).

DOI: https://doi.org/10.7554/eLife.32110.022

## Discussion

### Bringing functional genomics to non-model fungi with RB-TDNAseq

We employed an established method, *Agrobacterium tumefaciens*-mediated transformation, to extend barcoded insertion library techniques (**Wetmore et al., 2015**) into a non-model basidiomycetous fungus. The efficiency of *A. tumefaciens* transformation in diverse fungal species (**Michielse et al., 2005**; **Martínez-Cruz et al., 2017**; **Wu et al., 2016**; **Zhang et al., 2015**; **Liu et al., 2013**; **Zhang et al., 2014**; **Li et al., 2013**; **Han et al., 2012**; **Muniz et al., 2014**; **Rodrigues et al., 2013**; **Celis et al., 2017**) will enable use of RB-TDNAseq in many fungal species with limited genetic tools. We used RB-TDNAseq to simultaneously track mutants in over 6,500 genes for altered lipid catabolism and neutral lipid accumulation using a simple, scalable BarSeq protocol. The phenotypes measured in our high-throughput experiments were consistent with those observed for single gene deletion strains, demonstrating the reliability of this approach. In some respects *R. toruloides* was an ideal species to develop these methods. The *R. toruloides* genome is relatively compact (just over 20% of the sequence is predicted to be intergenic), and it grows as a haploid yeast. Effective BarSeq analysis on species with larger, less dense genomes will require greater sequence depth per sample. Typical fungal genomes are only modestly larger, though, around 35–45 Mb (**Mohanta and Bae, 2015**) vs 20 Mb for *R. toruloides*. Sequencing limitations are thus already minimal and will only decrease in the foreseeable future. A greater challenge will be adapting this technology in fungi that grow mainly as diploids or in filamentous, multicellular, or multinucleate forms harboring genetically distinct nuclei. Many of those species also produce haploid, uninucleate spores for sexual reproduction, asexual dispersal, or both. RB-TDNAseq can be applied to study the germination of these spores and their growth into nascent, isogenic colonies prior to their fusion into more physiologically and genetically complex networks of mycelia and fruiting bodies.

We found that genes recalcitrant to T-DNA insertion were highly enriched in orthologs for known essential genes, suggesting that most genes with very low insertion rates were likely essential in our mutagenesis conditions. Previous studies employing high-density transposon mutagenesis in fungi and bacteria have demonstrated the general utility of this approach (**Michel et al., 2017**; **Le Breton et al., 2015**). The high efficiency of *A. tumefaciens*-mediated transformation in diverse fungi should enable similar surveys in many poorly annotated fungi. We hope the provisional list of essential genes identified here will serve as a useful resource for genetics in *R. toruloides* and related species. In particular, orthologs to these genes may be potential targets for new antifungal strategies against basidiomycete pathogens, such as the closely related rusts of the Pucciniomycotina subphylum (**Singh et al., 2015**; **Park et al., 2015**) and the more distantly related human pathogen *Cryptococcus neoformans* (**May et al., 2016**).

### New insights into fatty acid catabolism in *R. toruloides*

The presence of a probable mitochondrial fatty acid beta-oxidation pathway in *R. toruloides* has been noted previously (**Zhu et al., 2012**). Our results confirm that this pathway is functional and essential for fatty acid utilization and add to mounting evidence that mitochondrial beta-oxidation is widespread in fungi (**Khan et al., 2012**). In mammals, some branched long-chain fatty acids are shortened in the peroxisome, then transferred via the acylcarnitine shuttle to the mitochondria for complete oxidation (**Wanders et al., 2015**; **Swigonová et al., 2009**), while other long-chain fatty acids are metabolized solely in the mitochondria (**Chegary et al., 2009**). *Rhodosporidium toruloides* has orthologs to the mammalian mitochondrial short, branched-chain and medium-chain acyl-CoA dehydrogenases *ACADSB* and *ACADM*, but not to the long-chain and very long-chain acyl-CoA dehydrogenases *ACADL* and *ACADVL*. *Rhodosporidium toruloides* also has several homologs to peroxisomal long chain acyl-CoA dehydrogenases *ACAD10* and *ACAD11*. In our experiments, both peroxisomal and mitochondrial beta-oxidation were necessary for robust growth on fatty acids and peroxisomal beta-oxidation enzymes had more variable fitness scores between different fatty acids.

**Table 2.** Predicted gene function: Mutants with decreased lipid accumulation.
Predicted functions for genes for which mutants were high-confidence candidates for decreased lipid accumulation (enrichment scores clustered in LA6 - LA8, *Figure 5*).

| | Gene ID | Short name | Annotation from | Description | Cluster | Enrichment BD | FACS |
|---|---|---|---|---|---|---|---|
| **tRNA thiolation** | | | | | | | |
| | RTO4_10764 | NCS2 | S. cerevisiae | tRNA 2-thiolation protein | LA7 | 0.5 | −2.3 |
| | RTO4_12817 | NCS6 | S. cerevisiae | tRNA 2-thiolation protein | LA7 | 0.7 | −2.6 |
| | RTO4_14918 | ELP2 | S. cerevisiae | Elongator complex protein | LA7 | 0.7 | −1.2 |
| | RTO4_14716 | IKI3 | S. cerevisiae | Elongator complex protein | LA7 | 0.4 | −1.1 |
| | RTO4_11341 | UBA4 | S. cerevisiae | Adenylyltransferase and sulfurtransferase | LA7 | 0.6 | −2.6 |
| **G Protein Switches** | | | | | | | |
| † | RTO4_15198 | Rab6 | H. sapiens | GTPase | LA6 | −1.3 | −1.6 |
| | RTO4_14622 | RGP1 | H. sapiens | Guanine nucleotide exchange factor | LA6 | −1.4 | −1.5 |
| **Kinases and Phosphatases** | | | | | | | |
| | RTO4_10698 | VHS1 | S. cerevisiae | Kinase | LA6 | 0.8 | −3.7 |
| | RTO4_16375 | HRK1 | S. cerevisiae | Kinase | LA6 | 0.4 | −2.2 |
| * | RTO4_11453 | GLC7 | S. cerevisiae | Kinase | LA8 | −1.2 | −0.9 |
| | RTO4_16810 | KIN1 | S. cerevisiae | Kinase | LA6 | 0.1 | −1.1 |
| | RTO4_10025 | SAT4 | S. cerevisiae | Kinase | LA7 | 1.6 | −3.6 |
| | RTO4_13327 | ATG1 | S. cerevisiae | Kinase | LA6 | 0.1 | −2.5 |
| | RTO4_14907 | SCH9 | S. cerevisiae | Kinase | LA6 | −0.6 | −2.0 |
| | RTO4_14906 | kinase-like | S. cerevisiae | Kinase | LA6 | −0.3 | −1.8 |
| | RTO4_13290 | YAK1 | S. cerevisiae | Kinase | LA8 | −1.1 | −0.9 |
| | RTO4_11732 | PPH3 | S. cerevisiae | Phosphatase 4 catalytic subunit | LA6 | 0.9 | −3.6 |
| | RTO4_12586 | PSY2 | S. cerevisiae | Phosphatase 4 regulatory subunit | LA6 | 0.2 | −1.2 |
| | RTO4_16463 | PTC7-like | S. cerevisiae | Phosphatase | LA6 | 0.1 | −2.0 |
| **Autophagy** | | | | | | | |
| | RTO4_13327 | ATG1 | S. cerevisiae | Kinase | LA6 | 0.1 | −2.5 |
| | RTO4_13598 | ATG2 | S. cerevisiae | Membrane protein | LA6 | −0.6 | −3.4 |
| | RTO4_12968 | ATG3 | S. cerevisiae | Ubiquitin-like-conjugating enzyme | LA6 | −0.8 | −4.5 |
| | RTO4_13496 | ATG4 | S. cerevisiae | Cysteine protease | LA6 | −0.1 | −2.3 |
| | RTO4_11901 | ATG7 | S. cerevisiae | Ubiquitin-like modifier-activating enzyme | LA6 | −0.8 | −4.2 |
| | RTO4_13543 | ATG8 | S. cerevisiae | Ubiquitin-like protein | LA6 | −1.0 | −4.2 |
| | RTO4_11326 | ATG9 | S. cerevisiae | Membrane protein | LA6 | 0.0 | −1.3 |
| | RTO4_9008 | ATG14 | S. cerevisiae | Autophagy-specific subunit of PtdIns3P-kinase complex | LA6 | 0.0 | −5.0 |
| | RTO4_16723 | ATG18 | S. cerevisiae | Phosphoinositide binding protein | LA6 | −0.9 | −5.8 |
| **Ubiquitination and Proteolysis** | | | | | | | |
| † | RTO4_16672 | PRB1 | S. cerevisiae | Vacuolar proteinase | LA6 | −0.2 | −1.7 |
| | RTO4_15345 | SIS1 | S. cerevisiae | Protein chaperone | LA6 | −0.4 | −1.2 |
| | RTO4_10423 | RMD5 | S. cerevisiae | GID complex E3 ubiquitin ligase | LA6 | −0.4 | −2.0 |
| | RTO4_11737 | GID8 | H. sapiens | GID complex member | LA6 | −0.1 | −1.5 |

*Table 2 continued on next page*

*Table 2 continued*

| | Gene ID | Short name | Annotation from | Description | Cluster | Enrichment BD | FACS |
|---|---|---|---|---|---|---|---|
| | RTO4_9816 | LONRF1 | H. sapiens | E3 ubiquitin ligase | LA6 | −0.5 | −4.5 |
| | RTO4_15320 | USP48 | H. sapiens | Ubiquitin carboxyl-terminal hydrolase | LA6 | 0.0 | −1.2 |
| | RTO4_9600 | COPS3 | H. sapiens | COP9 signalosome complex subunit | LA1 | 1.4 | 0.6 |
| | RTO4_11569 | GPS1 | H. sapiens | COP9 signalosome complex subunit | LA6 | 0.7 | −2.1 |
| **Triacylglyceride Synthesis** | | | | | | | |
| † | RTO4_12154 | GPD1 | S. cerevisiae | Glycerol-3-phosphate dehydrogenase | LA6 | −1.7 | −4.0 |
| | RTO4_11043 | BCSL2-like | H. sapiens | Seipin | LA6 | −0.8 | −2.9 |
| | RTO4_16460 | DGA1 | H. sapiens | Diacylglycerol acyltransferase | LA6 | −0.7 | −4.0 |
| | RTO4_14597 | ACS1 | S. cerevisiae | Acetyl-CoA synthetase | LA8 | −1.7 | −1.0 |
| | RTO4_10182 | YEF1 | S. cerevisiae | NAD+/NADH kinase | LA6 | −0.1 | −1.6 |
| ‡ | RTO4_11039 | GUT2 | S. cerevisiae | Glycerol-3-phosphate dehydrogenase | LA6 | −0.2 | −1.1 |
| **Lipid Droplet Associated** | | | | | | | |
| | RTO4_16381 | PLIN1-like | S. cerevisiae | Perilipin | LA6 | −1.7 | −4.3 |
| ‡ | RTO4_11039 | GUT2 | S. cerevisiae | Glycerol-3-phosphate dehydrogenase | LA6 | −0.2 | −1.1 |
| | RTO4_15372 | EGH1 | S. cerevisiae | Steryl-beta-glucosidase | LA6 | 0.7 | −2.5 |
| | RTO4_13614 | RIP1 | S. cerevisiae | Mitochondrial complex III iron-sulfur protein | LA6 | −0.5 | −2.8 |
| | RTO4_11043 | BCSL2-like | H. sapiens | Seipin | LA6 | −0.8 | −2.9 |
| | RTO4_16460 | DGA1 | H. sapiens | Diacylglycerol acyltransferase | LA6 | −0.7 | −4.0 |
| **Protein Modification** | | | | | | | |
| | RTO4_12670 | B3GALT1-like | H. sapiens | Beta-1,3-Galactosyltransferase | LA6 | −0.9 | −3.1 |
| **Protein Trafficking** | | | | | | | |
| † | RTO4_15198 | Rab6 | H. sapiens | GTPase | LA6 | −1.3 | −1.6 |
| **Other ER/Golgi Proteins** | | | | | | | |
| | RTO4_8838 | DNAJC4 | H. sapiens | DnaJ family chaperone | LA6 | −0.8 | −1.3 |
| | RTO4_13971 | DNAJC3 | H. sapiens | DnaJ family chaperone | LA6 | −1.1 | −2.2 |
| **Gene Expression** | | | | | | | |
| | RTO4_11333 | KLF18-like | H. sapiens | Transcription factor | LA6 | −0.2 | −1.1 |
| | RTO4_15641 | SKN7 | S. cerevisiae | Transcription factor | LA6 | 0.9 | −2.9 |
| | RTO4_14676 | LHX5-like | H. sapiens | Transcription factor | LA6 | −0.2 | −2.8 |
| | RTO4_11891 | HAP2 | S. cerevisiae | Transcription factor | LA6 | −0.8 | −2.4 |
| | RTO4_12420 | OPI1-like | S. cerevisiae | Transcription factor | LA6 | 0.0 | −3.7 |
| | RTO4_14100 | HAPX | C. neoformans | Transcription factor | LA8 | −1.2 | −1.7 |
| | RTO4_13255 | SGF73 | S. cerevisiae | SAGA-associated factor | LA6 | 0.4 | −1.5 |
| **Methylcitrate Cycle** | | | | | | | |
| | RTO4_14162 | ICL2 | S. cerevisiae | 2-methylisocitrate lyase | LA6 | −0.3 | −1.8 |
| | RTO4_12642 | PDH1 | S. cerevisiae | 2-methylcitrate dehydratase | LA6 | −0.1 | −1.7 |

Table 2 continued

| | Gene ID | Short name | Annotation from | Description | Cluster | Enrichment | |
|---|---|---|---|---|---|---|---|
| | | | | | | BD | FACS |
| **Electron Transport and Redox Balancing** | | | | | | | |
| | RTO4_11165 | CBP4 | S. cerevisiae | Mitochondrial complex III assembly factor | LA6 | −0.4 | −2.5 |
| | RTO4_13614 | RIP1 | S. cerevisiae | Mitochondrial complex III iron-sulfur protein | LA6 | −0.5 | −2.8 |
| | RTO4_13902 | AFG1 | S. cerevisiae | Mitochondrial complex IV assembly factor | LA6 | −0.3 | −1.3 |
| ‡ | RTO4_10010 | NDUFS4 | H. sapiens | Mitochondrial complex I accessory factor | LA8 | −1.3 | −0.1 |
| | RTO4_13925 | NDUFAF3 | H. sapiens | Mitochondrial complex I assembly factor | LA8 | −1.0 | −1.6 |
| **Amino Acid Biosynthesis** | | | | | | | |
| † | RTO4_12302 | CPA2 | S. cerevisiae | Large subunit of carbamoyl phosphate synthetase | LA6 | −0.4 | −2.4 |
| **Glucose and Energy Metabolism** | | | | | | | |
| | RTO4_10423 | RMD5 | S. cerevisiae | GID complex E3 ubiquitin ligase | LA6 | −0.4 | −2.0 |
| | RTO4_11737 | GID8 | H. sapiens | GID complex member | LA6 | −0.1 | −1.5 |
| | RTO4_12034 | TPS2 | S. cerevisiae | Trehalose 6-phosphate synthase | LA6 | 0.0 | −3.8 |
| * | RTO4_10264 | GLK1 | S. cerevisiae | Hexokinase | LA7 | 2.1 | −2.0 |
| **Transporters** | | | | | | | |
| † | RTO4_12909 | OAT1 | C. neoformans | Nucleobase transporter | LA6 | −0.2 | −1.1 |
| | RTO4_11397 | COT1 | S. cerevisiae | Vacuolar zinc transporter | LA6 | −0.2 | −1.1 |
| | RTO4_11924 | SNF3 | S. cerevisiae | Plasma membrane low glucose sensor | LA6 | 0.0 | −2.8 |
| **Other** | | | | | | | |
| | RTO4_12512 | cry | N. crassa | Blue-light photoreceptor cryptochrome | LA7 | 0.6 | −1.6 |
| | RTO4_14974 | | | Steroidogenesis/phosphatidylcholine transfer domain | LA6 | −0.3 | −1.2 |
| | RTO4_15889 | MAEA | H. sapiens | EMP macrophage erythroblast attacher | LA6 | −0.1 | −1.7 |
| | RTO4_16287 | CDD1 | S. cerevisiae | Cytidine deaminase | LA6 | 0.3 | −2.3 |
| | RTO4_15247 | WDR26 | H. sapiens | WD repeat protein | LA6 | −0.9 | −1.3 |
| | RTO4_8764 | MGS1 | S. cerevisiae | DNA-dependent ATPase and ssDNA annealing protein | LA6 | 0.2 | −1.2 |
| **Unknown** | | | | | | | |
| | RTO4_10431 | | | Protein of unknown function | LA6 | 0.7 | −1.6 |
| | RTO4_8973 | | | Protein of unknown function | LA8 | −0.2 | −1.1 |
| | RTO4_13195 | | | Protein of unknown function | LA6 | −0.2 | −1.1 |
| | RTO4_10367 | | | Protein of unknown function | LA6 | −0.1 | −1.3 |
| | RTO4_10102 | | | Protein of unknown function | LA6 | −0.3 | −1.2 |
| | RTO4_14926 | | | Protein of unknown function | LA6 | 0.2 | −1.7 |
| | RTO4_12045 | | | Protein of unknown function | LA6 | 0.0 | −1.5 |
| | RTO4_13600 | | | Protein of unknown function | LA6 | −0.3 | −1.3 |
| | RTO4_10976 | | | Protein of unknown function | LA6 | −0.2 | −1.5 |
| | RTO4_9970 | LDB17 | S. cerevisiae | Protein of unknown function | LA8 | −1.3 | −0.5 |
| | RTO4_13435 | | | Protein of unknown function | LA7 | 0.2 | −2.0 |
| | RTO4_9692 | | | Protein of unknown function | LA6 | −0.5 | −1.4 |
| | RTO4_15521 | | | Protein of unknown function | LA6 | 0.2 | −2.2 |
| | RTO4_8769 | | | Protein of unknown function | LA6 | −0.5 | −1.6 |

*Table 2 continued*

| Gene ID | Short name | Annotation from | Description | Cluster | Enrichment BD | FACS |
|---------|-----------|-----------------|-------------|---------|------|------|
| RTO4_8770 | | | Protein of unknown function | LA6 | −0.5 | −1.9 |
| RTO4_11259 | | | Protein of unknown function | LA7 | 0.7 | −3.3 |
| RTO4_9490 | | | Protein of unknown function | LA6 | −0.6 | −2.4 |
| RTO4_15520 | | | Protein of unknown function | LA6 | −0.5 | −2.5 |
| RTO4_8771 | | | Protein of unknown function | LA6 | −0.6 | −2.5 |
| RTO4_13452 | | | Protein of unknown function | LA6 | −1.3 | −4.0 |
| RTO4_15211 | | | Protein of unknown function | LA8 | −1.1 | −1.5 |

Cellular processes grouped as in **Figure 6**. BD: Enrichment score from buoyant density separation. FACS: Enrichment score from fluorescence activated cell sorting.

Protein abundance under nitrogen limitation: * increased; † increased 10-fold or more; ‡ decreased; § decreased 10-fold or more (**Zhu et al., 2012**).

DOI: https://doi.org/10.7554/eLife.32110.023

These observations are consistent with a model of beta-oxidation in which a large ensemble of peroxisomal enzymes shorten diverse long-chain fatty acids in the peroxisome and a smaller ensemble of enzymes metabolize short-chain fatty acids in the mitochondria. Our results demonstrate how a barcoded insertion library can accelerate discrimination of function between closely related members of a diversified gene family. Fitness assays on a much larger panel of substrates should yield further insights into the individual functions of *R. toruloides'* diverse complement of peroxisomal enzymes and guide experimental design for their biochemical characterization.

## Extending high-throughput fitness techniques to lipid production

While pooled fitness experiments have been used extensively to identify novel gene function, work so far has primarily focused on growth-based phenotypes, with only limited exploration of other phenotypes (**Sliva et al., 2016**; **Hassan et al., 2016**; **Tyo et al., 2009**). In this study we used two proven strategies for differentiating between cells with altered lipid accumulation, buoyant density centrifugation (**Eroglu and Melis, 2009**; **Kamisaka et al., 2006**; **Liu et al., 2015**) and FACS (**Terashima et al., 2015**; **Xie et al., 2014**), and applied them to our barcoded mutant pool. Inconsistencies between the two assays and with respect to independent BODIPY staining of targeted deletion strains suggests significant false positive rates for each assay in isolation. When both assays were in agreement, however, 18 of 21 deletion mutants had the expected phenotype in independent experiments. This approach identified 150 high confidence candidate genes with strong impacts on lipid accumulation under nitrogen limitation. While this set is likely incomplete, it complements previous transcriptional and proteomic studies to establish critical genes and cellular processes supporting lipid accumulation that deserve more intensive study. As has been noted in previous functional screens (**Smith et al., 2006**), there was limited overlap between genes for which mutants had a detectable lipid accumulation phenotype in our study and genes with altered protein abundance in *R. toruloides* during lipid accumulation (**Zhu et al., 2012**) (14 genes) or genes that co-purified with *R. toruloides* lipid droplets (five genes) (**Zhu et al., 2015**). The different ensembles of genes identified by each technique illustrate that these systems-level approaches complement each other.

## New insights into regulation of lipid metabolism in *R. toruloides*

Proteomic, transcriptomic, mutagenic and over-expression surveys of lipid metabolism have been carried out in several model eukaryotic systems including *S. cerevisiae* (**Bozaquel-Morais et al., 2010**; **Fei et al., 2008**; **Szymanski et al., 2007**; **Grillitsch et al., 2011**; **Fei et al., 2011**; **Ruggles et al., 2014**; **Currie et al., 2014**; **Bouchez et al., 2015**), *C. elegans* (**Ashrafi et al., 2003**; **Zhang et al., 2010**; **Liu et al., 2014**; **Lee et al., 2014**; **Lapierre et al., 2011**), *D. melanogaster* (**Cermelli et al., 2006**; **Guo et al., 2008**; **Beller et al., 2006**; **Beller et al., 2008**; **Krahmer et al., 2013b**), various mammalian cell lines (**Zehmer et al., 2009**; **Nishino et al., 2008**; **Tu et al., 2009**),

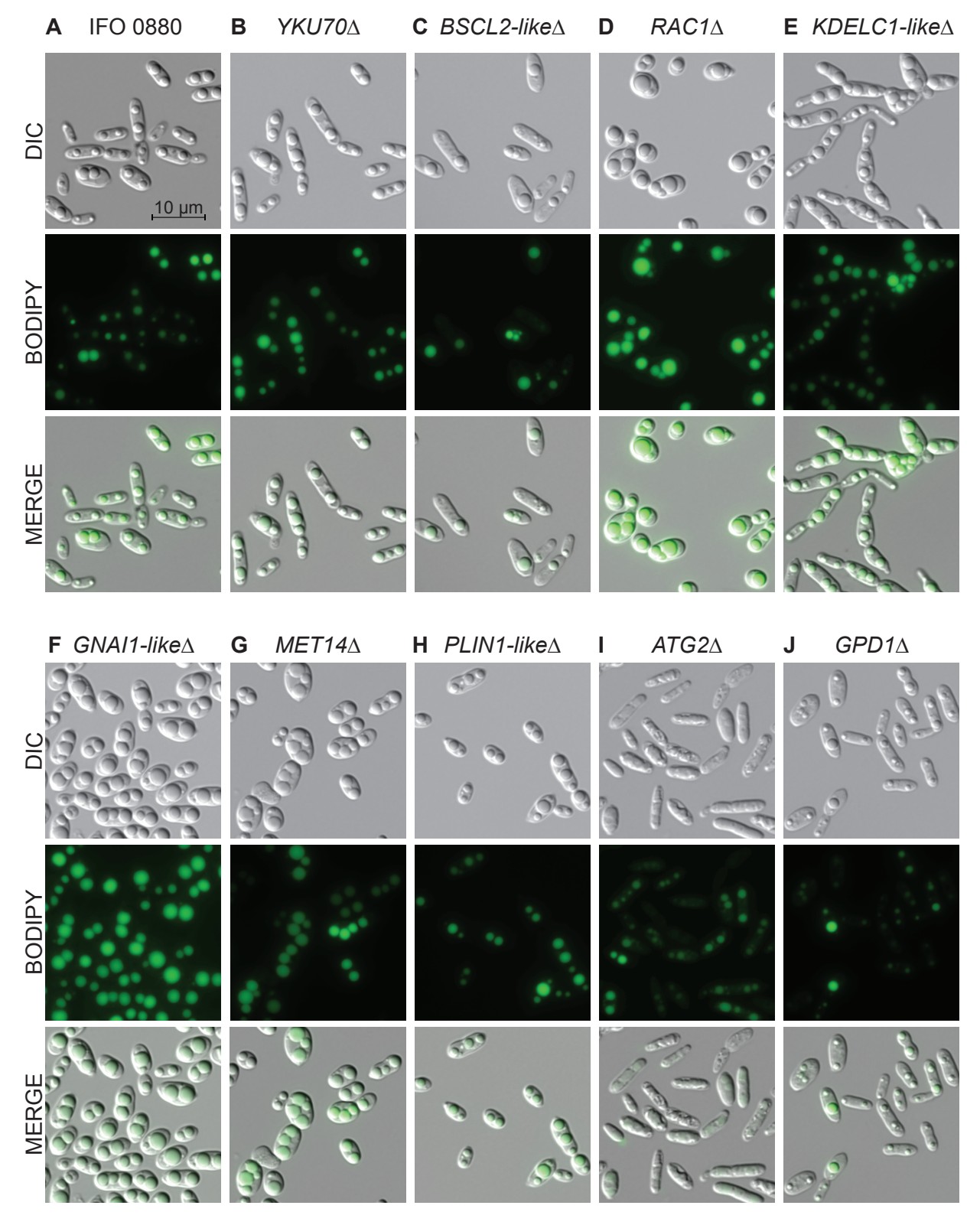

**Figure 7.** Light and fluorescence microscopy images of selected lipid accumulation mutants. DIC microscopy on eight deletion mutants for lipid accumulation genes. All deletion mutants (**C–J**) were constructed in a *YKU70Δ* background to enable homologous recombination at the targeted locus. Cells were grown 40 hr in low nitrogen lipid accumulation media. DIC, BODIPY 493/503 fluorescence, and composite images are shown for ten strains. (**A**) *R. toruloides* IFO 0880 (WT). (**B**) *RTO4_11920Δ* ortholog of *YKU70*. (**C**) *RTO4_11043Δ* similar to *H. sapiens BSCL2*. (**D**) *RTO4_14088Δ* ortholog of *H.*

*Figure 7 continued on next page*

*Figure 7 continued*

*sapiens RAC1*. (E) *RTO4_10371Δ* similar to *H. sapiens KDELC1*. (F) *RTO4_16215Δ* similar to *H. sapiens GNAI1*. (G) *RTO4_8709Δ* ortholog of *MET14*. (H) *RTO4_16381Δ* similar to *H. sapiens PLIN1*. (I) *RTO4_13598Δ* ortholog of *ATG2*. (J) *RTO4_12154Δ* ortholog of *GPD1*. The following figure supplements are available for *Figure 7*.

DOI: https://doi.org/10.7554/eLife.32110.024

The following figure supplement is available for figure 7:

**Figure supplement 1.** Additional light and fluorescence microscopy images.

DOI: https://doi.org/10.7554/eLife.32110.025

and *Y. lipolytica* (**Athenstaedt et al., 2006**; **Pomraning et al., 2017**; **Silverman et al., 2016**) (see **Supplementary file 5** for a summary of genes identified in 35 studies). These studies employed different analytical techniques and culture conditions, and identified many genes without clear orthologs across the different species used, making a granular meta-analysis extremely difficult. A few broad themes are apparent, however. Protein trafficking and organelle interaction are inextricably linked with lipid body formation, growth and mobilization. Membrane-bound G proteins in the endomembrane network have conserved roles regulating trafficking and cellular morphology in response to metabolic states. A complex network of signaling cascades, protein modifications and transcription factors mediate the transition to lipid accumulation or lipid mobilization. A major output of this regulation is amino acid metabolism. Lipid metabolism and autophagy are deeply linked in a complex manner. Our findings were consistent with these general themes, including some orthologs to genes identified in the studies above, but the importance of general functions was more conserved across species than the roles of specific orthologous gene sets. The genes and processes we identify here should be considered in any strategy to optimize lipid metabolism in *R. toruloides* specifically or oleaginous yeasts in general. Comparative study of these processes across diverse species in standardized conditions will likely be required to uncover which aspects are fundamental to lipid droplet accumulation, maintenance and variation, and which processes are integrated by specific regulatory circuits in a given organism. See Appendix 1 for a deeper discussion of the individual genes for which mutants had altered lipid accumulation in our experiments and how those observations relate to previous work.

## Uncovering function for novel genes

In this study, we identified 46 *R. toruloides* genes with no functional predictions (**Supplementary file 1**), but which had important functions in lipid metabolism as evidenced by reduced fitness when grown on fatty acids or altered lipid accumulation. These included nine genes with broad conservation across ascomycete and basidiomycete fungi and seven genes with conservation across several basidiomycete species. These genes are of particular interest for further study into their specific functions in lipid metabolism. Moreover, the mutant pool generated in this study should be an excellent tool to assign functions for uncharacterized *R. toruloides* genes. Cofitness analysis is a particularly powerful method for uncovering the function of novel genes in pathways and processes for which one or more well-characterized genes is also required (**Hillenmeyer et al., 2010**). Closely interacting genes exhibit strongly correlated fitness scores across large panels of diverse conditions. Because the T-DNA insertions in the mutant pool are barcoded, fitness experiments are inherently scalable to a large number of conditions. Because the analytical methods we employed maximize portability and scalability across large compendiums of experiments (**Wetmore et al., 2015**), individual experiments can be conducted at different times under specialized culture conditions, at different scales, and even by different laboratories, yet the data can be effectively compared, maximizing the power of cofitness analysis. We encourage the *R. toruloides* community and the broader fungal community to make use of this new resource and collaborate with us to maximize its potential.

## Conclusions

In conclusion, we believe that RB-TDNAseq holds great promise for rapid exploration of gene function in diverse fungi. Because ATMT has been demonstrated in numerous, diverse fungi, we expect this method will be portable to many non-model species. Because the fitness analysis is inherently scalable, it will enable rapid fitness analysis over large compendia of conditions. Cofitness analysis of

such compendia will accelerate the annotation of new genomes and identify new classes of genes not abundant in established model fungi. In this study, we demonstrated the application of RB-TDNAseq to the study of lipid metabolism in an oleaginous yeast that has significant potential to become a new model system for both applied and fundamental applications. We identified a large set of genes from a wide array of subcellular functions and compartments that impact lipid catabolism and accumulation. These processes and genes must be considered and addressed in any metabolic engineering strategy to optimize lipid metabolism in *R. toruloides* and other oleaginous yeasts. Deeper understanding of the extreme cell-to-cell variation in lipid accumulation seen across eukaryotes will likely require deeper mechanistic understanding of these processes and their interaction with the lipid droplet. The principles learned from exploring lipid metabolism and storage across diverse eukaryotes will inform biotechnological innovations for the production of biofuels and bioproducts, as well as new therapies for metabolic disorders.

## Materials and methods

### Strains

We used *R. toruloides* IFO 0880 (also called NBRC 0880, obtained from Biological Resource Center, NITE (NBRC), Japan) as the starting strain for all subsequent manipulations. We used *Agrobacterium tumefaciens* EHA 105 and plasmids derived from pGI2 (*Abbott et al., 2013*) for *A. tumefaciens*-mediated transformation (ATMT) of *R. toruloides* (strain and plasmid kindly provided by Chris Rao, UIUC). The barcoded mutant pool was constructed by ATMT. We made all gene deletions in a non-homologous end-joining deficient *YKU70Δ* background (*Zhang et al., 2016b*) by homologous recombination of a nourseothricin resistance cassette introduced by either ATMT or electroporation of a PCR product. For deletions made by ATMT we used flanking arms of ~1000–1500 bp for homologous recombination. We found that as few as 40 bp of flanking sequence were sufficient for homologous recombination of PCR products at many loci. All strains used in this study, and primers used for strain construction and verification are listed in *Supplementary file 4*.

### Culture conditions

For most experiments, we used optical density (OD) as measured by absorbance at 600 nm on a GENESYS 20 spectrophotometer (Thermo Fisher Scientific, 4001–000, Waltham, MA) as a metric for growth and to control inoculation density. For IFO 0880 grown in rich media, 1 OD unit represents approximately 30 million cells/mL. Unless otherwise noted, cultures were grown at 30°C in 100 mL liquid media in 250 mL baffled flasks (Kimble Chase, 25630250, Vineland, New Jersey) with 250 rpm shaking on a New Brunswick Innova 2300 platform shaker (Eppendorf, M1191-0000, Hauppauge, New York) with constant illumination using a LUMAPRO 6W LED lamp (Grainger, 33L570, San Leandro, CA). We used yeast-peptone-dextrose (YPD) media (BD Biosciences, BD242820, San Jose, CA) for general strain maintenance and rich media conditions. For auxotrophy experiments we used 0.67% w/v yeast nitrogen base (YNB) w/o amino acids (BD Biosciences, BD291940) with 111 mM glucose (Sigma-Aldrich, G7528, St. Louis, MO) as our defined media and supplemented with 75 mM L-methionine (Sigma-Aldrich, M9625), 75 mM L-arginine (Sigma-Aldrich, A5006), or 0.2% w/v drop-out mix complete (DOC), which contains all 20 amino acids, adenine, uracil, p-aminobenzoic acid, and inositol (US Biological, D9515, Salem, MA). To test growth and fitness on oleic acid (Sigma-Aldrich, O1008 and 364525), ricinoleic acid (Sigma-Aldrich, R7257), and methyl ricinoleic acid (Sigma-Aldrich, R8750), we used this same defined media formulation with 1% fatty acid (by volume) instead of glucose. For lipid accumulation experiments, we pre-cultured strains for two generations in YPD (OD 0.2 to OD 0.8) then washed them twice and resuspended them at OD 0.1 in low nitrogen medium; 0.17% w/v yeast nitrogen base (YNB) w/o amino acids or ammonium sulfate (BD Biosciences, BD233520), 166 mM D-glucose, 7 mM $NH_4Cl$ (Thermo Fisher Scientific, S25168A), 25 mM $KH_2PO_4$ (Thermo Fisher Scientific, P285-3), and 25 mM $Na_2HPO_4$ (Sigma-Aldrich, S0876). This is the C:N 120 formulation from Nicaud et al. (*Nicaud et al., 2014*). Unless otherwise specified, cultures were harvested for lipid quantification or fractionation after 40 hr of growth and lipid accumulation. In all experiments biological replicates refer to samples from independent cultures in the experimental condition. Biological replicates processed on the same day were usually inoculated from the same YPD pre-culture, except for BarSeq experiments. For BarSeq experiments we seeded

independent starter cultures in YPD and collected a 'Time 0' reference sample after two generations. In downstream fitness or enrichment analysis, we explicitly paired each sample from an experimental condition with the Time 0 sample from the starter culture replicate from which it was seeded.

## Genome sequencing and de novo assembly

To generate an improved genome assembly for IFO 0880 we prepared genomic DNA for PacBio RS II sequencing (Pacific Biosciences, Menlo Park, CA). Genomic DNA was purified using a two-step protocol, first using glass bead lysis and phenol-chloroform extraction, as previously described (*Zhang et al., 2016a*), followed by a QIAGEN Genomic-tip 100/G method (QIAGEN, 10243, Germantown, MD). All QIAGEN buffers were obtained from a Genomic DNA Buffer Set (QIAGEN, 19060). Briefly, the dry genomic DNA pellet was first resuspended in G2 buffer supplemented with 200 µg/mL RNase A (QIAGEN, 19101) and 13.5 mAU/ml Proteinase K (QIAGEN, 19131), incubated at 50°C for one hour, and then loaded on a Tip-100 column. After three washes with QC buffer and elution with QF buffer, the DNA was precipitated with isopropanol and removed by spooling using a glass Pasteur pipet. The genomic DNA was washed with 70% ethanol and after air-drying, resuspended in EB buffer (pH 7.5). DNA concentration was determined using a Qubit 3.0 fluorometer (Thermo Fisher Scientific, Q33218) and submitted to University of Maryland Genomics Resource Center for library preparation and sequencing. A 10 kb insert, size selected (BluePippin, Sage Science, Beverly, MA) SMRTbell library was prepared and sequenced on a PacBio RS II platform using P4C2 chemistry and 10 SMRT cells. De novo assembly of 610,663 polymerase reads (mean subread length of 5,193 bp) was performed using SMRT Analysis version 2.3.0.140936 (http://www.pacb.com/support/software-downloads/) and the RS_HGAP_Assembly.3 protocol (HGAP3) using default settings except for a genome size of 20,000,000 bp. The final assembly contained 30 polished contigs (mean coverage of 131-fold) with a total genome size of 20,810,536 bp. Paired-end Illumina data (17,817,326 PE100 reads, [*Zhang et al., 2016a*]) was used for error correction using Pilon version 1.13 (https://github.com/broadinstitute/pilon). As expected, the most common type of correction (569 in total) was insertion or deletion of a nucleotide in homopolymer regions. The final error corrected scaffolds were annotated by JGI and submitted to Genbank under the accession LCTV02000000. Raw sequence data (PacBio and Illumina) has been deposited in the NCBI SRA (SRP114401 and SRP058059, respectively).

## RNA sequencing and analysis

To harvest RNA for improved gene model prediction, we inoculated *R. toruloides* into 50 mL cultures in M9 Minimal Salts Solution (BD Biosciences, BD248510), 2 mM MgSO$_4$ (Sigma-Aldrich, M7506), 100 µM CaCl$_2$ (Sigma-Aldrich, C5670), and Yeast Trace Elements Solution (88 µg/mL nitrilotriacetic acid, 175 µg/mL MgSO$_4$ 7H$_2$O, 29 µg/mL MnSO$_4$ H$_2$O, 59 µg/mL NaCl, 4 µg/mL FeCl$_2$, 6 µg/mL CoSO$_4$, 6 µg/mL CaCl$_2$ 2H$_2$O, 6 µg/mL ZnSO$_4$ 7H$_2$O, 0.6 µg/mL CuSO$_4$ 5H$_2$O, 0.6 µg/mL KAl (SO$_4$)$_2$ 12H$_2$O, 6 µg/mL H$_3$BO$_3$, 0.6 µg/mL Na$_2$MoO$_4$ H$_2$O), pH 7.0, with 2% glucose (Sigma-Aldrich, D9434) or 10 mM p-coumaric acid (trans-4-hydroxycinnamic acid; Alfa Aesar, A15167, Tewksbury, MA), and incubated overnight at 30°C with 200 rpm shaking. We harvested cultures at mid-log phase, centrifuged at 3,000 RCF for 10 min at room temperature, removed the supernatant and flash-froze the cell pellet in an ethanol/dry ice bath and stored at −80°C. We lyophilized pellets overnight in a FreeZone-12 freeze dry system (Labconco, 7754030, Kansas City, MO) and extracted total RNA with a Maxwell RSC Plant RNA Kit (Promega, AS1500, Madison, WI) using a Maxwell RSC instrument (Promega, AS4500). RNA was sequenced and mapped to the *R. toruloides* IFO 0880 genome at the Department of Energy Joint Genome Institute (JGI) in Walnut Creek, CA with in-house protocols.

## Gene model predictions and curation

The improved genome assembly was annotated using the JGI Annotation pipeline (*Grigoriev et al., 2014*). Owing to relatively small intergenic spacing in the *R. toruloides* genome, fused gene models were a common problem. We hand curated over 500 gene models by searching for homology to unrelated proteins at each end of the automated gene models and inspecting agreement with assembled transcripts from our RNAseq experiments. Briefly, for all protein models over 400 amino acids long, we used the N-terminal and C-terminal 30% of each sequence in separate BLAST queries

(NCBI BLAST-plus software 2.2.30) to a custom database of proteins from 22 other eukaryotic genomes (see Orthology relationships, below). We then compared the significant alignments for each terminus of a given gene and scored them for disagreement in regards to the respective orthology groups to which each target sequence belonged with a custom Python script (*Coradetti, 2018a*; copy archived at https://github.com/elifesciences-publications/fusedgenemodels). The top-scoring 500 gene models were manually inspected for uncharacteristically long introns and for predicted introns and exons not supported by RNAseq reads and modified as required using the Mycocosm genome browser. The current genome annotation is publicly available at the JGI Mycocosm web portal (*Grigoriev et al., 2014*): http://genome.jgi.doe.gov/Rhoto_IFO0880_4

## Orthology relationships

We predicted orthologous proteins for our *R. toruloides* gene models in *H. sapiens, D. melanogaster, C. elegans, A. thaliana, C. reinhartii, S. cerevisiae,* and 16 other fungi with the orthomcl software suite version 2.0.9 (*Li et al., 2003*). See *Supplementary file 1* for a full list of ortholog groups and details on the genomes used in this analysis.

## Vector library construction

To efficiently construct a large and diverse mutant pool of barcoded mutants we first constructed a large library of barcoded vectors with an optimized Type IIS endonuclease cloning strategy (*Engler et al., 2008*). We modified the ATMT vector pGI2 (*Abbott et al., 2013*) to act as a barcode receiving vector by first removing the two pGI2 SapI sites already present on the vector backbone through SapI restriction digestion, treatment with T4 DNA polymerase for blunt end formation and subsequent blunt end ligation. Next, we introduced two divergent SapI recognition sites just inside the right border of the T-DNA (vector pDP11) as the integration site for random barcoding. We added the barcodes by synthesizing the oligonucleotide GATGTCCACGAGGTCTC TNNNNNNNNNNNNNNNNNNNNNCGTACGCTGCAGGTCGAC and amplifying with primers TCA-CACAAGTTTGTACAAAAAAGCAGGCTGGAGCTCGGCTCTTCGCCCGATGTCCACGAGGTCTCT and CTCAACCACTTTGTACAAGAAAGCTGGGTGGATCCGCTCTTCAATTGTCGACCTGCAGCG TACG. We then combined 4 μg of vector and 140 ng of barcode fragments in a 50 μl reaction with 5 μl 10x T4 ligase buffer, 5 μl 10x NEB CutSmart buffer (NEB, B7204S, Ipswich, MA), 2.5 μl T7 ligase (NEB, M0318L), and 2.5 μl of SapI (NEB, R0569S). We incubated the reaction at 37°C for 5 min, then 25 cycles of 37°C for 2 min and 20°C for 5 min, before denaturing the enzymes for 10 min at 65°C. Without cooling the product, we added 1 μl SapI and incubated for 30 min at 37°C to digest any uncut vector, then cooled to 10°C. We purified the barcoded plasmids using a Zymo DNA clean and concentrator kit (Zymo Research, D4014, Irvine, CA), eluting in 15 μl of elution buffer and pooled 10 barcoding reactions. We then transformed *E. coli* electrocompetent 10-beta cells (NEB, C3019I) according to the manufacturers specifications in 30 independent transformations. We estimated the diversity of the barcoded vector pool by performing barcode sequencing as described below, sequencing on an Illumina MiSeq system and estimating the true pool size by the relative proportion of barcodes with 1 or 2 counts. See the script Multicodes.pl from Wetmore et al. (*Wetmore et al., 2015*) for details. This yielded a barcoded pool estimated to consist of ~100 million clones.

## *Agrobacterium* mediated transformation of *R. toruloides*

We transformed the barcoded vector pool into *A. tumefaciens* EHA 105 with a protocol adapted from established methods (*Mersereau et al., 1990*). We diluted a stationary phase starter culture 1:100 in 500 ml Luria-Bertani broth (BD Biosciences, BD244620) and cultured for 6 hr at 30°C. We pelleted cells at 3,000 RCF for 10 min at 4°C, washed pellets in ice-cold 1 mM HEPES (Thermo Fisher Scientific, BP310), pH 7.0, then washed them in ice-cold 10% glycerol 1 mM HEPES, suspended cells in 5 ml ice-cold 10% glycerol 1 mM HEPES, and flash froze 50 μl aliquots in liquid nitrogen. To produce a large transformant pool of *A. tumefaciens* bearing millions of unique barcode sequences, we electroporated 5 ml of competent cells with 50 μg of plasmid DNA (50 μl per well) in a HT100 96-well plate chamber (BTX, 45-0400, Holliston, MA) with a 2.5 kV pulse, 400 ohm resistance and 25 μF capacitance from an ECM 630 wave generator (BTX, 45-0051). We recovered cells in LB for 2 hr at 30°C, and plated on LB agar with 50 μg/ml kanamycin (Sigma-Aldrich, K4000).

Approximately 14 million transformation events were scraped and collected into a mixed pool for transformation of *R. toruloides*.

We grew the barcoded *A. tumefaciens* pool to OD 1 in 50 mL YPD in a baffled flask at 30°C, then pelleted the cells and suspended in 10 mL induction medium (1 g/L NH$_4$Cl, 300 mg/L MgSO$_4$ 7H$_2$O, 150 mg/L KCl (Thermo Fisher Scientific, P267-500), 10 mg/L CaCl$_2$ (VWR, 0556, Radnor, PA), 750 µg/L FeSO$_4$ 7H$_2$O (Thermo Fisher Scientific, AC423731000), 144 mg/L K$_2$HPO$_4$ (VWR, 0705), 48 mg/L NaH$_2$PO$_4$ (Thermo Fisher Scientific, BP329), 2 g/L D-Glucose, 10 mg/L thiamine (Sigma-Aldrich, T4625), 20 mg/L acetosyringone (Sigma-Aldrich, D134406), and 3.9 g/L MES (Sigma-Aldrich, 69892), adjusted to pH 5.5 with KOH) and incubated 24 hr at room temperature in culture tubes on a roller drum. We cultured *R. toruloides* in 10 mL YPD to OD 0.8, then pelleted the cells and suspended in the induced *A. tumefaciens* culture for 5 min at room temperature. We filtered the mixed culture on a 0.45 µm membrane filter (EMD Millipore, HAWP04700, Bedford, MA) then transferred the filter to induction media 2% agar (BD Biosciences, BD214010) plates for incubation at 26°C for 4 days. We then washed the filters in YPD and plated on YPD 2% agar with 300 µg/ml cefotaxime (Sigma-Aldrich, C7039) and 300 µg/ml carbenicillin (Sigma-Aldrich, C1389) and incubated at 30°C for two days. We scraped these plates to collect transformed *R. toruloides*, recovered the mutant pool in YPD plus cefotaxime and carbenicillin for 24 hr, added glycerol to 15% by volume and stored at −80°C. We repeated this protocol 40 times to recover approximately 2 million transformation events. In some rounds of transformation, we also included 0.05% casamino acids (BD Biosciences, BD223120) or 1% CD lipid concentrate (Thermo Fisher Scientific,11905–031) in the induction media plates to promote recovery of mutants with impaired amino acid or lipid biosynthesis. We then recovered each of these transformation subpools on YPD plus cefotaxime and carbenicillin 12 hr to clear residual *A. tumefaciens* and combined them into one master pool, divided it into 1 ml aliquots in YPD 15% glycerol and stored them at −80°C. Laboratories with an interest in experimenting with this mutant pool should contact the corresponding authors.

## TnSeq library preparation

To isolate high quality genomic DNA we harvested ~10$^8$ cells from a fresh YPD culture of the mutant pool, washed the pellet in water and suspended in 200 µl TSENT buffer (2% Triton X-100 (Sigma-Aldrich, T8787), 1% SDS (Thermo Fisher Scientific, AM9820), 1 mM EDTA (Sigma-Aldrich, ED2SS), 100 mM NaCl (Sigma-Aldrich, S5150), 10 mM Tris-HCl, pH 8.0 (Invitrogen, 15568–025, Carlsbad CA)). We then added the sample to 200 µl 25:24:1 phenol/chloroform/isoamyl alcohol (Invitrogen, 15593–031) in screw-top tubes with glass beads (Sigma-Aldrich, Z763748) on ice and vortexed for 10 min at 4°C. We added 200 µl TE buffer (Thermo Fisher Scientific, AM9858), centrifuged at 21,000 RCF for 20 min at 4°C, removed the aqueous phase to 1 mL 200 Proof ethanol (Koptec, V1016, King of Prussia, PA) and centrifuged at 21,000 RCF for 20 min at 4°C to pellet DNA. DNA was dried and suspended in 200 µl TE, treated with 0.5 µl RNase A (Qiagen, 19101), then purified with a Genomic DNA Clean and Concentrator Kit (Zymo Research, D4064). We checked DNA quality on a 0.8% agarose E-Gel (Thermo Fisher Scientific, G51808) and quantified with a Qubit 3.0 fluorometer using the dsDNA HS reagent (Invitrogen, 1799096).

To sequence sites of genomic insertions we followed the TnSeq protocol of Wetmore et al. (*Wetmore et al., 2015*), using their Nspacer_barseq_universal primer and P7_MOD_TS_index primers for final amplification (*Supplementary file 4*). Because we found a high proportion of non-specific products in our TnSeq mapping and highly variable recovery of the same insertions between technical replicates, we sequenced multiple replicates for each batch of ATMT mutants (around 10,000–100,000 mutants per batch) and used at least two annealing temperatures for the final PCR enrichment for each batch. In total, we sequenced about 900 million reads from 64 independent TnSeq libraries. A full summary of TnSeq libraries used to map the mutant pool is listed in *Supplementary file 4*. Libraries were submitted for single-end 150 bp Illumina sequencing on a HiSeq 2500 platform at the UC Berkeley Vincent J. Coates Genomics Sequencing Laboratory, except for a subset of smaller runs on an Illumina MiSeq platform as indicated in *Supplementary file 4*. Sequence data have been submitted to the NCBI Short Read Archive (SRP116146).

## Mapping insertion locations

We used a similar strategy as Wetmore et al. (*Wetmore et al., 2015*) to map the location of each barcoded T-DNA insertion, with minor alterations (*Coradetti, 2018b*).

MapTnSeq_trimmed.pl processes the TnSeq reads to identify the barcode sequence and is a modified version of MapTnSeq.pl (*Wetmore et al., 2015*), with three minor alterations. We ignore the last 10 bases of the T-DNA sequence, as the length of T-DNA border sequence included in the final insertion is variable. We also allow for barcode sequences of 17–23 base pairs instead of exactly 20. We relaxed this restriction because on manual inspection of our TnSeq data we found that approximately 10% of barcodes appeared to be slightly shorter or longer than 20 base pairs, likely a result of imperfect PAGE purification after oligonucleotide synthesis. We report all TnSeq reads in which sequence past the end of the expected T-DNA insert aligns with other regions of the T-DNA sequence, or with the outside vector as 'past end' reads. These are mappings of junctions between concatemeric T-DNA inserts and unprocessed T-DNA vectors, respectively.

RandomPoolConcatemers.py is a custom script that associates barcode sequences mapped in MapTnSeq_trimmed.pl with genomic locations and then filters those barcodes for insertions at unique, unambiguous locations. First, for all barcodes sequenced, the number of reads mapping to any genomic location and the number of reads mapping to concatemeric junctions are tabulated. Any barcodes that only differ by a single base pair from a barcode with 100 times more reads are removed as likely sequencing errors and reported as 'off by one' barcodes. Any barcode for which there are more than seven times as many 'past end' reads as reads mapping to genomic locations as 'past-end' barcodes. The past-end barcodes are further characterized as 'head-to-tail' concatemers (majority of Tnseq reads map to the left border T-DNA sequence), 'head-to-head' concatemers (majority of the reads map to the right border T-DNA sequence), or 'Run-on' insertions (majority of reads map to pGI2 outside the T-DNA sequence). Any barcodes for which the majority of TnSeq reads map ambiguously to the genome are removed and reported as ambiguous barcodes. Any barcodes for which 20% or more of the TnSeq reads map to a different location than the most commonly observed location are removed and reported as 'multilocus' barcodes. Finally, any barcodes mapped within 10 bases of a more abundant barcode for which there is a Levenshtein edit distance (*Levenshtein, 1966*) less than five are removed as likely sequencing errors and reported as 'off by two' barcodes. The remaining unfiltered barcodes are reported as the mutant pool.

InsertionLocationJGI.py is a custom script to match the genomic locations of barcodes in the mutant pool to the nearest gene in the current JGI *R. toruloides* gene catalog and report whether the insertion is in a 5-prime intergenic region, a 5-prime UTR, an exon, an intron, a 3-prime UTR, or a 3-prime intergenic region of that gene.

InsertBias.py is a custom script to analyze potential biases in T-DNA insertion rates. The script tracks number of insertions versus scaffold length for all scaffolds in the genome, GC content in the local regions of insertion, and insertion rates in promoter regions, 5-prime untranslated mRNA, exons, introns, 3-prime untranslated mRNA, and terminator regions. To assess fine-scale biases in insertion locations, all locations in the genome are apportioned to one of the above feature types, then for each feature type, the same number of insertions as were observed for that feature type in the mutant pool are sampled at random (without replacement) from all the genomic locations assigned to that feature type.

## Barcode sequencing

We isolated genomic DNA with a Fungal/Bacterial DNA MiniPrep kit (Zymo Research, D6005). We used Q5 high-fidelity polymerase with GC-enhancer (NEB, M0491S) to amplify unique barcode sequences flanked by specific priming sites, yielding a 185 bp Illumina-sequencing-ready product (*Figure 1—figure supplement 1*). We used BarSeq primers from Wetmore et al. (*de Hoon et al., 2004*) (*Supplementary file 4*), except we replaced primer P1 with a mix of primers with 2–4 random bases to improve nucleotide balance for optimal sequencing of low-diversity sequences (*Illumina, 2013*). We cleaned PCR products with a DNA clean and concentrator kit (Zymo Research, D4014). We quantified product yield with a Qubit 3.0 fluorometer system and mixed as appropriate for sequencing as multiplexed libraries. We sequenced libraries on an Illumina HiSeq 4000 system at the UC Berkeley Vincent J. Coates Genomics Sequencing Laboratory. If necessary, libraries were purified further with a Pippin Prep system (Sage Biosciences) before

loading with 15% PhiX DNA as a phasing control for low diversity samples (*Illumina, 2013*). We sequenced each biological replicate to a depth of at least 20 million reads. We counted occurrences of T-DNA barcodes in each sample with the script MultiCodes_Variable_Length.pl, a modified version of MultiCodes.pl from Wetmore et al. (*Wetmore et al., 2015*) that allows for barcodes of 17–23 base pairs.

## Fitness analysis

For all BarSeq experiments, we thawed frozen aliquots of the mutant pool on ice and inoculated them into YPD at OD 0.2. Cultures were recovered for about 12 hr until OD 600 was approximately 0.8. Cultures were pelleted at 3,000 RCF for 5 min, washed twice in the appropriate media, and transferred to the condition of interest. Samples were taken from the YPD starter cultures (Time 0) and after 5–7 doublings in the experimental condition. Average fitness scores and T-like statistics as metrics for consistency between individual insertion mutants in each gene were calculated with the scripts combineBarSeq.pl and FEBA.R from Wetmore et al. (*Wetmore et al., 2015*).

Briefly, for each biological replicate and condition, for any barcode with an average of at least three counts in Time 0 samples, a strain fitness score is calculated as $F_{strain} = \log_2(C_{condition} + sqrt(P))$ – $\log_2(C_{Time0} + 1/sqrt(P))$, where C is the raw counts for the barcode and P is a gene-specific 'pseudo-count' added to reduce noise in fitness scores for low-count strains. These strain fitness scores are then normalized such that the median score is 0 to correct for coverage differences between the samples. The strain fitness scores are then assigned a weight proportional to the harmonic mean of counts at Time 0 and in the condition sample. For any one barcode, the weighting mean is capped at 20 reads, which has the effect of limiting the influence of generally more abundant outlier strains (*Wetmore et al., 2015*). T is calculated as the gene fitness divided by the square root of the variance in strain fitness scores. This variance is estimated as the maximum value of a naïve estimate based on Poisson noise or the observed variance (a weighted sum squares of differences in strain fitness versus gene fitness scores plus an estimate of global variance in gene fitness scores calculated by comparing fitness scores in the first and second half of every gene). See the methods subsection 'BarSeq data analysis and calculation of gene fitness' in the original publication by Wetmore et al. (*Wetmore et al., 2015*) for more detail on these algorithms. Wetmore et al. limited their analysis to genes with an average of at least 30 total counts at Time 0, spread across three strains. Because the list of genes satisfying this requirement can change from experiment to experiment, we established a list of genes that met this requirement in any of our experiments and used that list for our analysis. As a result, a minority of genes (649) have fitness scores based on data from one or two barcodes. The number of barcodes used in fitness analysis of each gene is listed in all relevant tables in *Supplementary file 2*. In general, genes with data from only one or two barcodes had smaller T-statistics and thus were filtered out in later analyses.

Because Wetmore et al.'s software does not consider biological replication between independent cultures, we then averaged fitness scores for each condition and combined T-statistics across replicates with the script AverageReplicates.py, treating them as true T-statistics. That is: $T_{condition} = Sum(T_{replicates})/Sqrt(N_{replicates})$. To assess consistency of differences in observed fitness between growth conditions we computed $T_{c1 - c2} = (F_{c1} - F_{c2})/Sqrt((F_{c1}/T_{c1})^2 + ((F_{c2}/T_{c2})^2)$ with the script ResultsSummary.py. We generated K-means clusters of fitness scores using Pearson correlation as the similarity metric using Cluster 3.0 (*de Hoon et al., 2004*). For comparing enrichment in density and FACS separated fractions we computed F and T for each fraction versus the $T_0$ control. The enrichment score E and T between fractions was then calculated as $E = F_{high\ lipid} - F_{low\ lipid}$ and $T_{high\ lipid - low\ lipid} = (F_{high\ lipid} - F_{low\ lipid})/Sqrt((F_{high\ lipid}/T_{high\ lipid})^2 + ((F_{low\ lipid}/T_{low\ lipid})^2)$ with the script ResultsSummary.py. We generated hierarchical clusters of enrichment scores using Pearson correlation as the similarity metric and average linkage as the clustering method. All fitness data are available in *Supplementary file 2* and the fitness browser (http://fungalfit.genomics.lbl.gov/). Custom Python scripts are available at (*Coradetti, 2018b*; copy archived at https://github.com/elifesciences-publications/rb-tdnaseq). Sequence data have been submitted to the NCBI Short Read Archive (SRP116193)

## Transformation of *R. toruloides* by electroporation

We cultured *R. toruloides* overnight in 10 mL YPD on a roller drum to an OD 600 of 2, then pelleted cells at 3,000 RCF for 5 min at 4°C in a benchtop centrifuge (Eppendorf, 5810 R). Cells were kept at 4°C from this point. We transferred the pellets to 1.5 mL tubes and washed them four times with ice cold 0.75 M D-sorbitol (Sigma-Aldrich, S1876), centrifuging each wash 30 s at 8,000 RCF, 4°C (Eppendorf, 5424). After the final wash, we removed excess D-sorbitol and added 35 µl of cell pellet to 10 µl of fresh 0.75 M D-sorbitol and ~1 µg of PCR product in 5 µl water in a chilled 0.1 cm cuvette. We electroporated cells at 1.5 kV, 200 ohms and 25 µF with an ECM 630 (BTX) electroporation system. We then added 1 mL cold 1:1 mixture of YPD and 0.75 M D-sorbitol and transferred to 14 mL round bottom culture tubes for a 3 hr recovery culture at 30°C with shaking at 200 rpm on a platform shaker. We then pelleted the cultures at 8,000 RCF for 30 s, suspended in 200 µl YPD, and then plated on YPD with 100 µg/mL nourseothricin (5.005.000, Werner Bioagents, Germany).

## Gene ontology enrichment

We scored enrichment of gene ontology terms with a custom script that performs a hypergeometric test on the frequency of each term in the genome versus the frequency in given gene set (script GOenrich.py, available at [*Coradetti, 2018b*]). We corrected for multiple hypothesis testing with the Benjamini-Hochberg correction (*Benjamini and Hochberg, 1995*). We extended the GO terms associated with *R. toruloides* genes in the current JGI annotation by collecting terms for orthologous genes in *Arabidopsis thaliana, Aspergillus nidulans, Caenorhabditis elegans, Candida albicans, Homo sapiens, Mus musculus,* and *Saccharomyces cerevisiae*, obtained from the Gene Ontology Consortium (*Ashburner et al., 2000*; *Gene Ontology Consortium, 2015*).

## Total fatty acid quantification with gas chromatography

Cell lysis, extraction of total lipids, and conversion to fatty acid methyl esters (FAMEs) was based on a published protocol (*Browse et al., 1986*). We cultured IFO 0880, a selection of seven targeted deletion strains (see *Supplementary file 6*) and one overexpression strain (RT880-AD, [*Zhang et al., 2016a*]) in low nitrogen medium for 48 or 96 hr. We collected paired 5 mL samples from each in screw-top glass tubes (Corning, 99502–10, Corning, NY) and 15 mL polyethylene tubes (Corning, 352096) for lipid extraction and mass determination, respectively. We pelleted samples by centrifugation at 2,000 RCF for 20 min at 4°C, and washed once in water to remove salts and unused glucose. We then transferred the mass determination sample to a pre-tared 1.5 mL microcentrifuge tube. We froze both samples at −20°C overnight, then lyophilized them 48 hr in a FreeZone freeze dry system (Labconco, 7754042) before weighing/extraction. We added 1 mL methanol spiked with 250 µg methyl tridecanoate to each sample to serve as an internal standard (ISTD). We then resuspended lipid extraction samples (usually about 10–20 mg) by vortexing in 3 mL 3N methanolic HCl (Sigma-Aldrich, 33050-U) and 200 µl chloroform (Sigma-Aldrich, 472476) and incubated at 80°C water bath for 1 hr. Cell lysis and conversion to FAMEs occurs during this incubation. To extract FAMEs we then added 2 mL hexane (Sigma-Aldrich, 650552) and vortexed samples well before centrifugation at 3,000 RCF for 3 min. One µL of the hexane layer was injected in split mode (1:10) onto a SP-2330 capillary column (30 m x 0.25 mm x 0.2 µm, Sigma-Aldrich, 24019). An Agilent 7890A gas chromatograph equipped with a flame ionization detector (FID) was used for analysis with the following settings: Injector temperature 250°C, carrier gas: helium at 1 mL/min, temperature program: 140°C, 3 min isocratic, 10 °C/min to 220°C, 40 °C/min to 240°C, 5 min isocratic. FAME concentrations were calculated by comparing the peak areas in the samples to the peak areas of ten commercially available high-purity standards (C16:0, C16:1, C17:0, C18:0, C18:1, C18:2, C20:0, C20:1, C22:0, C24:0) (Sigma-Aldrich) in known concentration relative to the internal standard, respectively.

## Relative TAG measurement with BODIPY and flow cytometry

We inoculated deletion mutants and the *YKU70Δ* parental strain at OD 0.1 in low nitrogen medium and cultured for 40 hr. We fixed samples by adding 180 µl cell culture to 20 µl 37% formaldehyde (Electron Microscopy Sciences, Hatfield, PA) and incubating for 15 min at room temperature. We then diluted fixed cells 1:100 in 200 µl PBS (from 10X concentrate, Thermo Fisher Scientific, 70011–44) with 0.5 M KI and 0.25 µg/mL BODIPY 493/503 (Thermo Fisher Scientific, D-3922), then incubated 30 min at room temperature. We quantified BODIPY signal for 10,000 cells per sample on a

Guava HT easyCyte system (EMD Millipore) in the green channel (excitation 488 nm, emission 525 nm) using InCyte software (EMD Millipore). Due to logistical constraints, samples were processed in batches of at most 30 cultures at a time. Each batch included three biological replicates of the YKU70Δ parental strain as an internal reference. Distribution of mutant strains into these batches was not explicitly randomized, but each batch included both strains expected to accumulate more lipid and strains expected to accumulate less lipid than the parent. Each mutant was processed in at least two different batches.

## Population enrichment with FACS

We cultured the barcoded mutant pool in low nitrogen medium for 40 hr. We then diluted unfixed cells 1:100 in 10 ml PBS with 0.5 M KI and 0.25 μg/mL BODIPY 493/503, then incubated 30 min at 30°C with shaking. We then sorted the population on a Sony SH800 cell sorter with a 70 μM fluidic chip, sorting in semi-purity mode. We first applied a gate for single cell events with forward scatter height within 15% of forward scatter area. We sorted a sample of 10 million cells with the scattering gate alone as a control population, to account for effects of growth, sorting, and collection that are independent of lipid accumulation. Then we collected the 10% of the size-filtered population with the highest and lowest signals in the FITC channel. We collected 10 million cells each for the high and low signal populations. We collected all sorted cells in YPD with 300 μg/ml cefotaxime (Sigma-Aldrich, C7039) and 300 μg/ml carbenicillin (Sigma-Aldrich, C1389), then grew them to saturation in our standard culture conditions and pelleted 1 mL sample, and then stored at −20°C for BarSeq analysis.

## Population enrichment with sucrose density gradients

We prepared linear sucrose gradients with the method of Luthe et al. (*Luthe, 1983*). For example, to prepare a 65–35% sucrose gradient; we prepared four solutions of sucrose (Sigma-Aldrich, G7528) at 65, 55, 45, and 35 grams per 100 mL in PBS, then successively froze 10 mL layers of each concentration in a 50 mL conical tube (Corning, 430829) on dry ice and stored the gradient at −20°C. We selected appropriate gradients to maximize the physical separation of the cell population by running trial experiments with wild type IFO 0880 cultures on a number of sucrose gradients. The gradients used in each experiment are described in *Table 3*. Approximately 24 hr before performing density separation on cell population, the appropriate step gradient was moved to 4°C to thaw, yielding a linear gradient (*Luthe, 1983*).

To perform the separation, we centrifuged 50 mL of culture at 6,000 RCF at 4°C for 20 min. We then suspended the pellet in 5 ml PBS at 4°C and carefully loaded it onto a sucrose gradient. We centrifuged the gradients for 1 hr at 5,000 RCF at 4°C with slow acceleration and no brake for deceleration in an Avanti J-26 XP centrifuge with a JS5.3 swinging bucket rotor (Beckman Coulter, Brea, CA). To collect fractions, we pierced the bottom of each tube with the tip of a 16 gauge needle (BD Biosciences, 305197), to slowly drain the gradient from the bottom, at 1 drop every 1–5 s. We collected 2 mL fractions, estimated average fraction density by weighing a 100 μl sample and measured the distribution of the cell population across the sample by optical density. The appropriate fractions

**Table 3.** Sucrose density gradients used in this study

| Media | Time | Sucrose range (Density)* | Average density ±StDev | High buoyancy fractions (Density) | Median buoyancy fractions (Density) | Low buoyancy fractions (Density) |
|---|---|---|---|---|---|---|
| Low Nitrogen | 40 hr | 50–20% (1.22–1.10) | 1.177 ±0.003 | 17–20 (<1.11) | 6–7 (1.18–1.19) | 1–2 (>1.21) |
| YPD | 40 hr | 80–50% (1.29–1.16) | 1.234 ±0.012 | 19–22[†] (<1.14) | 4–8[†] (1.24–1.27) | 1 (>1.28) |

All density measurements in g/mL

*Highest and lowest specific density measured in any collected fraction in the linear portion of the gradient.

[†]Some biological replicates differ in exact fractions collected. Fractions were collected within this range such that the high buoyancy fraction constituted the most buoyant 5–10% of the population, the median buoyancy fraction constituted the median 30–50% of the population and the low buoyancy fraction constituted the least buoyant 5–10% of the population.

DOI: https://doi.org/10.7554/eLife.32110.026

were then combined to sample the least buoyant (highest density) 5–10%, median buoyancy 30–50%, and most buoyant (lowest density) 5–10% of the population. For each biological replicate, we also collected a 1 mL sample from the culture before separation to monitor growth in the experimental condition.

## Microscopy

Cover slips were submerged in 0.1% v/v polylysine (Sigma-Aldrich, P8920) for 15 min. Cover slips were removed from polylysine and blotted dry from the bottom of vertically-held slips. Slips were then washed several times with ddH$_2$O and rapidly dried with compressed air. Directly prior to imaging, slips were visually inspected for streaks and dust and softly cleaned with lens paper. Cells were grown 40 hr in low nitrogen medium, 1 mL of culture was transferred to 2 mL microcentrifuge tubes with 1 mL of PBS, and tubes were mixed briefly by vortexing. Cells were pelleted at 9,000 RCF for 1 min in a microcentrifuge, and then resuspended in 100 µl of fluorescent staining solution (PBS with 0.5 M KI and 0.25 µg/mL BODIPY 493/503) to visualize intracellular lipid droplets. Four µl of stained cells were pipetted up and down and transferred to the clean slides. Polylysine-coated cover slips were carefully placed on the 4 µl drop to ensure even spreading of liquid. Cells were observed on an Axio Observer microscope (Carl Zeiss Microscopy, Thornwood, NY) with a plan-apochromat 100x DIC objective (Carl Zeiss Microscopy, 440782-9902-000), ORCA-Flash 4.0 camera (Hamamatsu, C11440-22CU, Japan), and ZenPro 2012 (blue edition) software. For BODIPY imaging cells were illuminated with an X-cite Series 120 arc-lamp (EXFO Photonics Solutions, Canada) and 38HE filter set, 450–490 excitation, 500–550 emission (Carl Zeiss Microscopy, 489038-9901-000). Zvi files were converted to 16 bit TIFF images and representative fields of view were cropped and channels merged using FIJI image processing software (*Schindelin et al., 2012*).

## Acknowledgements

We thank Christopher Rao and Shuyan Zhang for initial advice and protocols for ATMT. We thank Kelly Wetmore and Adam Deutschbauer for their guidance on technical aspects of TnSeq and Bar-Seq experiments. We thank Morgan Price for his assistance and advice on TnSeq and BarSeq analysis, as well as for hosting our data on the fitness browser. This material is based upon work supported by the U.S. Department of Energy, Office of Science, Office of Biological and Environmental Research program under Award Number DE-SC-0012527. Preliminary work establishing genetic, culturing, and assay protocols with *R. toruloides* was funded by grants OO1605 and OO6J01 from the Energy Biosciences Institute at the University of California Berkeley. Work performed at the DOE Joint BioEnergy Institute (http:// www.jbei.org) is supported by the U.S. Department of Energy, Office of Science, Office of Biological and Environmental Research, through Contract No. DE-AC02-05CH11231 between Lawrence Berkeley National Laboratory and the U.S. Department of Energy. The work conducted by the U.S. Department of Energy Joint Genome Institute (http://jgi.doe.gov/), a DOE Office of Science User Facility, is supported by the Office of Science of the U.S. Department of Energy under Contract No. DE-AC02-05CH11231 between Lawrence Berkeley National Laboratory and the U.S. Department of Energy. This work used the Vincent J Coates Genomics Sequencing Laboratory at UC Berkeley, supported by NIH S10 Instrumentation Grants S10RR029668, S10RR027303, and S10OD018174.

# Additional information

## Funding

| Funder | Grant reference number | Author |
| --- | --- | --- |
| U.S. Department of Energy | Office of Biological and Environmental Research, DE-SC-0012527 | Samuel T Coradetti<br>Dominic Pinel<br>Gina M Geiselman<br>Masakazu Ito<br>Ya-Fang Cheng<br>Stefan Bauer<br>Rachel B Brem<br>Adam P Arkin<br>Jeffrey M Skerker |
| University of California Berkeley | OO1605 | Dominic Pinel<br>Adam P Arkin<br>Jeffrey M Skerker |
| University of California Berkeley | OO6J01 | Dominic Pinel<br>Adam P Arkin<br>Jeffrey M Skerker |
| U.S. Department of Energy | Office of Biological and Environmental Research, DE-AC02-05CH11231 | Stephen J Mondo<br>Igor V Grigoriev<br>John M Gladden<br>Blake A Simmons |

The funders had no role in study design, data collection and interpretation, or the decision to submit the work for publication.

## Author contributions

Samuel T Coradetti, Conceptualization, Data curation, Software, Formal analysis, Validation, Investigation, Methodology, Writing—original draft, Writing—review and editing; Dominic Pinel, Conceptualization, Data curation, Software, Formal analysis, Validation, Investigation, Visualization, Methodology, Writing—original draft, Writing—review and editing; Gina M Geiselman, Validation, Investigation, Writing—review and editing; Masakazu Ito, Resources, Methodology; Stephen J Mondo, Data curation, Software, Formal analysis; Morgann C Reilly, Investigation, Writing—review and editing; Ya-Fang Cheng, Validation, Investigation, Visualization, Methodology, Writing—review and editing; Stefan Bauer, Methodology, Writing—review and editing; Igor V Grigoriev, Blake A Simmons, Resources, Funding acquisition, Writing—review and editing; John M Gladden, Resources, Funding acquisition, Project administration, Writing—review and editing; Rachel B Brem, Conceptualization, Resources, Supervision, Funding acquisition, Methodology, Writing—review and editing; Adam P Arkin, Conceptualization, Resources, Supervision, Funding acquisition, Methodology, Writing—original draft, Writing—review and editing; Jeffrey M Skerker, Conceptualization, Resources, Formal analysis, Supervision, Funding acquisition, Investigation, Visualization, Methodology, Writing—original draft, Project administration, Writing—review and editing

## Author ORCIDs

Samuel T Coradetti (iD) http://orcid.org/0000-0003-0173-0403
Stephen J Mondo (iD) https://orcid.org/0000-0001-5797-0647
John M Gladden (iD) http://orcid.org/0000-0002-6985-2485
Adam P Arkin (iD) http://orcid.org/0000-0002-4999-2931
Jeffrey M Skerker (iD) http://orcid.org/0000-0003-2653-1566

## Decision letter and Author response

Decision letter https://doi.org/10.7554/eLife.32110.040
Author response https://doi.org/10.7554/eLife.32110.041

## Additional files

### Supplementary files

• Supplementary file 1. *R. toruloides* Genome Summary. Overview of the revised genome assembly of *R. toruloides* IFO 0880 with summary information on gene models, predicted functions, orthologous genes, manually curated annotations for genes with lipid utilization and accumulation phenotypes observed in this study, and likely essential genes identified by recalcitrance to T-DNA insertion.
DOI: https://doi.org/10.7554/eLife.32110.027

• Supplementary file 2. BarSeq and Mutant Data. Fitness and enrichment scores from BarSeq data in all conditions/fractions tested in this study; fitness/enrichment scores used in clustering analysis for *Figures 2*, *3* and *5*; raw data for growth of validation deletion mutant strains on fatty acids; and summary statistics for all validation experiments with deletion mutants, including flow cytometry on lipid accumulation mutants.
DOI: https://doi.org/10.7554/eLife.32110.028

• Supplementary file 3. GO Enrichments. GO term enrichments for genes clustered in *Figure 3—figure supplement 1* and *Figure 5A*.
DOI: https://doi.org/10.7554/eLife.32110.029

• Supplementary file 4. Strains, Primers, and Libraries. High throughput sequencing libraries used in this study, strains constructed in this study, and primers used for strain construction on RB-TDNAseq analysis.
DOI: https://doi.org/10.7554/eLife.32110.030

• Supplementary file 5. Functional Lipid Surveys. Summary of genes identified in 35 systems-level studies on lipid accumulation, lipid catabolism, and lipid droplet biology in fungi and other eukaryotes with functional genomics screens and proteomics.
DOI: https://doi.org/10.7554/eLife.32110.031

• Supplementary file 6. FAME versus BODIPY. Raw data for *Figure 4—figure supplement 1* comparing total lipid from gas chromatography of fatty acid methyl esters to average BODIPY intensity in samples of wild type and mutant strains of *R. toruloides*.
DOI: https://doi.org/10.7554/eLife.32110.032

• Transparent reporting form
DOI: https://doi.org/10.7554/eLife.32110.033

### Major datasets

The following datasets were generated:

| Author(s) | Year | Dataset title | Dataset URL | Database, license, and accessibility information |
|---|---|---|---|---|
| Coradetti ST, Pinel D, Geiselman GM, Ito M, Mondo SJ, Reilly MC, Cheng YF, Bauer S, Grigoriev IV, Gladden JM, Simmons BA, Brem RB, Arkin AP, Skerker JM | 2017 | BarSeq on *Rhodosporidium toruloides* | http://www.ncbi.nlm.nih.gov/bioproject/?term=PRJNA400215 | Publicly available at NCBI BioProject (accession no: PRJNA400215) |
| Coradetti ST, Pinel D, Geiselman GM, Ito M, Mondo SJ, Reilly MC, Cheng YF, Bauer S, Grigoriev IV, Gladden JM, Simmons BA, Brem RB, Arkin AP, Skerker JM | 2017 | RB-TDNAseq on *Rhodosporidium toruloides* | http://www.ncbi.nlm.nih.gov/bioproject/?term=PRJNA400086 | Publicly available at NCBI BioProject (accession no: PRJNA400086) |

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

## Appendix 1

DOI: https://doi.org/10.7554/eLife.32110.034

# Refining the *R. toruloides* IFO 0880 genome sequence and annotation

An effective functional genomics approach requires high quality genomic sequence and reliable gene models. To improve assembly, we added long-read sequencing from Pacific Biosciences to our previously published data from Illumina sequencing (*Zhang et al., 2016a*). The refined gapless assembly is high quality, consisting of 21 megabases on 30 scaffolds (N50 = 6, L50 = 1.4 Mb) and a complete 112 Kb mitochondrial genome. Seven de novo scaffolds have telomeric repeats (*Ramirez et al., 2011*) at both ends, suggesting they represent complete chromosomes, and seven scaffolds have a telomeric repeat at one end (*Supplementary file 1*). For comparison, electrophoretic karyotyping of *R. toruloides* NP11 indicated 16 total chromosomes (*Zhu et al., 2012*). We also used 100 bp paired-end Illumina sequencing of mRNA to improve gene model prediction. The revised genome (*Rhodosporidium toruloides* IFO 0880 v4.0) encoding 8,490 predicted proteins is available at the Joint Genome Institute's Mycocosm genome portal (*Nordberg et al., 2014*) and Genbank accession LCTV02000000. While the bulk of the gene models were predicted with the JGI's automated protocols, erroneous fusion of neighboring genes was a significant issue. We have manually corrected several hundred fused models not supported by RNAseq data and encourage the *R. toruloides* research community to continue annotation refinement through the JGI portal. Summary tables of gene IDs, predicted functions, and probable orthologs in other systems are included in *Supplementary file 1*.

# Additional detail on mapping insertion locations with RB-TDNAseq

We adapted a high-throughput phenotyping strategy previously demonstrated in bacteria (*Wetmore et al., 2015*) by employing *Agrobacterium tumefaciens*-mediated transformation (ATMT) (*Figure 1A*). Briefly, we created a large barcoded mutant pool in which *A. tumefaciens* transfer DNAs (T-DNA) bearing an antibiotic resistance cassette and a 20 base-pair random sequence (barcode) were inserted randomly throughout the genome. We then mapped the location of each insertion and its associated barcode with RB-TDNAseq, a variant of RB-TnSeq (a high-throughput method to enrich and sequence a diverse pool of transposon/genome junctions [*Wetmore et al., 2015*]), applied to T-DNA inserts. A more detailed view of the junction sequence and primers used for RB-TDNAseq and BarSeq are shown in *Figure 1—figure supplement 1*.

From a mutant pool of approximately two million *R. toruloides* colonies, we sequenced 1,391,040 unique barcoded insertions with RB-TDNAseq. We successfully mapped 293,613 barcodes (21%) to T-DNA insertions at unique, unambiguous locations in the *R. toruloides* genome. The remainder of sequenced barcodes could not be mapped for several reasons (*Figure 1—figure supplement 2A*). T-DNA is often inserted in concatemeric repeats (*Kunitake et al., 2011*; *Sullivan et al., 2002*; *Rolloos et al., 2014*), in which case only RB-TDNAseq reads from the terminal repeat provides mapping information. If the terminal repeat is truncated (*Bundock and Hooykaas, 1996*), or if it abuts genomic sequence that is recalcitrant to sequencing for any reason, then we are able to detect the barcode at junctions between T-DNA repeats, but not at junctions with the genome (47% of sequenced barcodes were not mappable for this reason). Likewise, if the terminal T-DNA is inserted in an inverted orientation, the result is an unmappable convergent concatemer (5% of barcodes). About 16% of barcodes were associated with vector sequence outside the T-DNA sequence, indicating integration of unprocessed plasmid into the genome. Approximately 1% of RB-TDNAseq reads mapped equally well to two or more highly similar sequences and thus we could not determine which locus is the true site of insertion. Finally, another 1% of barcodes appeared in

distinct RB-TDNAseq reads mapping to two or more sequences, suggesting two or more different mutant strains have received the same barcode, rendering those strains indistinguishable in BarSeq data.

T-DNA can integrate into multiple locations in the same genome, giving rise to confounding phenotypes between different mutations. Rates of multi-locus insertion range widely (5% to 45%) depending on transformation conditions and the targeted cell type (*Kunitake et al., 2011*; *Ondřej et al., 1999*; *De Neve et al., 1997*; *Choi et al., 2007*; *Głowacka et al., 2016*). Multi-locus insertions can be derived from multiple copies of T-DNA from a single transformation event, or from co-transformation of distinct T-DNAs. Since only 1% of barcodes mapped to multiple locations, we inferred the former scenario was rare. To estimate the frequency of multiple insertion events from co-transformation, we isolated single colonies and then sequenced their barcodes using PCR amplification with Sanger sequencing. Of 58 colonies with unambiguous sequence of the common sequence preceding the random barcodes, 41 colonies (71%) had a single, unique sequence in the barcode region, suggesting a single barcode was present, and 17 colonies (29%) had mixed signals in the barcode region suggesting T-DNAs with multiple barcodes were present (example traces in *Figure 2—figure supplement 2B*). This estimate may be biased by sequence artifacts and should be taken as an upper bound. Furthermore, co-transformed T-DNAs are often integrated into a single concatemeric repeat (*De Neve et al., 1997*; *De Buck et al., 2009*; *De Block and Debrouwer, 1991*). Thus, far fewer than 29% of strains may actually harbor T-DNA insertions at multiple loci. Conversely, T-DNA insertions have been shown to cause other local mutations (6% of insertions were associated with deletions of more than 100 bp and 0.7% with local inversions in *A. thaliana* [*Kleinboelting et al., 2015*]). These combined sources of confounding phenotypes highlight the importance of integrating data from multiple T-DNA insertions in any fitness analysis. As such, our main concern in constructing our mutant pool was to effectively probe the entire genome with multiple inserts per gene.

## Fine-scale biases in T-DNA insertion sites

On a genome level, there was no significant bias in rates of T-DNA insertion, with insertion number proportional to scaffold length (*Figure 1—figure supplement 3A*) and no apparent bias in insertion rates with respect to local GC content (*Figure 1—figure supplement 3B*). We did observe some bias in T-DNA insertion sites at the kilobase scale, however. T-DNAs were mapped within intergenic regions at a higher rate than expected given the composition of the genome (*Figure 1—figure supplement 3C*). For instance, 20% of T-DNA inserts were mapped in promoter regions, even though these regions only constitute 8% of the genome. This bias towards promoter regions is consistent with observations in *Cryptococcus neoformans* (*Walton et al., 2005*), in *Magnaporthe oryzae* (*Choi et al., 2007*), and with the fact that 41% of T-DNA insertions in *S. cerevisiae* mapped in intergenic regions (*Bundock et al., 2002*) though only 27% of the *S. cerevisiae* genome is intergenic (*Alexander et al., 2010*). We also observed further fine-scale variation in the density of mapped insertions, with dozens of T-DNA 'hotspots' on each scaffold with a higher local density of T-DNA insertion that cannot be explained by a simulated random integration with the observed biases towards promoters, terminators and five-prime UTRs (*Figure 1—figure supplement 3D*). We have not explored the mechanism of these fine scale biases, though microhomology to T-DNA borders and local DNA bendability have been suggested as influencing T-DNA insertion into eukaryotic genomes (*Choi et al., 2007*; *Zhang et al., 2007a*).

## Additional information on strain abundance and sequencing depth

We observed a wide range in relative abundance of individual mutant strains (i.e. relative counts for different barcodes in BarSeq data). In a typical fitness experiment, we sequenced each sample to a depth of 20 million reads (as opposed to 900 million reads to map insertion locations by RB-TDNAseq). At this depth, approximately 40,000 mapped barcodes (14%) were too rare to count. Countable barcodes ranged from 1 to 1,000 counts per sample with a mode

around 10 (*Figure 2—figure supplement 1A*). Further, for estimating strain abundance in fitness experiments we considered only insertions in the central 80% of the coding region to avoid confounding data from incorrectly predicted gene boundaries, functional truncated proteins, and altered expression of neighboring genes. Within these constraints, we were able to measure fitness for 6,558 genes (92% of non-essential genes) by tracking abundance of 68,021 insertions in coding regions with a median of 7 insertions per gene (*Figure 2—figure supplement 1B*). This distribution of countable barcodes and sequencing depth translated to approximately 100–250 total usable reads per sample for estimating fitness of mutants in each gene (*Figure 2—figure supplement 1C*). This depth of usable sequence on a per-gene basis is comparable to the point at which Wetmore et al. found that additional sequence depth gave diminishing returns in terms of reduced variance in fitness scores (*Wetmore et al., 2015*).

## Contributions of individual strains to gene-level fitness scores

Examples of how fitness scores from individual barcoded insertion strains are combined to calculate gene-level fitness scores are shown in *Figure 2—figure supplement 2*. *ARG5* ortholog *RTO4_9377* is an example of a gene with a strong, unambiguous conditional fitness score. We mapped 47 barcoded T-DNA insertions between the start and stop codon of *ARG5*, of which 21 were located in the central 80% of this coding region and were abundant enough to be counted in BarSeq experiments. Though raw counts varied over two orders of magnitude for these strains (*Figure 2—figure supplement 2A*), strain fitness scores in non-supplemented media were similar across the entire length of the gene model, including introns, with a sharp change in fitness scores at the gene boundaries (*Figure 2—figure supplement 2B*). In supplemented media (DOC), counts for all insertions are similar to the Time 0 samples (*Figure 2—figure supplement 2C*). Thus fitness scores average around zero and few barcodes consistently have fitness scores with a magnitude >1 across replicates (*Figure 2—figure supplement 2D*)

To calculate F and T for the gene, first fitness scores are averaged between insertions, with more abundant insertions weighted more than less abundant ones (relative weights in this average are indicated by shading of the fitness scores in panel B). This weighting serves three purposes: (1) to privilege fitness scores with less error due to sampling and sequencing noise, (2) to bias the average score towards that of strains with counts in both conditions, and (3) to limit the influence of rare, high-abundance outlier strains. Intuitively, strains with more counts will tend to have scores less affected by sequencing noise or stochastic representation of cells in small samples from a given culture. Because the weight is based on the harmonic mean of counts in Time 0 and the condition sample, strains with very low counts in one condition (and thus more noisy $\log_2$ ratios) will be weighted lower than those with $\log_2$ ratios with less noise in both the numerator and denominator. The effect of rare, high-abundance strains, which tend to have discordant fitness scores (*Wetmore et al., 2015*), is reduced by capping the weight assigned to any one insert to that given a strain with an average of 20 counts. T is calculated as a moderated T-statistic: the gene fitness (F) is divided by the square root of a conservative estimate of variance in F amongst strains. These statistics are first computed for each biological replicate separately, then F is averaged across replicates and T is combined as a true T-statistic between independent experiments, by summing T and dividing by the square root of the number of replicates.

*MET16* ortholog *RTO4_11741* is an example of a gene with more ambiguous data (*Figure 2—figure supplement 2E–H*). We mapped 15 barcoded T-DNA insertions between the start and stop codon of *MET16*, of which seven were located in the central 80% of this coding region and were abundant enough to be counted in BarSeq experiments. Visual inspection of the raw reads for these barcodes at Time 0 and after growth on non-supplemented media suggests that mutants in *MET16* are deficient for growth in non-supplemented conditions, but two of the barcodes behave differently than the others; they show no significant reduction in counts in the non-supplemented conditions. In particular, a single barcode that mapped near the three prime end of the gene has relatively high counts

and fitness scores around zero, weakening the overall fitness score and yielding a marginal T-statistic. In this particular case, restricting the region of the gene used in the analysis would improve F and T and better capture the true phenotype of the mutants in this gene. However, any further global restriction on the barcodes analyzed per gene would compromise the data for other genes with fewer insertions.

In most cases, meaningful interpretation of fitness scores will require comparisons amongst several conditions. If the conditions of interest are tested in the same experiment with replicates inoculated from the same Time 0 samples, then direct comparison of the BarSeq counts between conditions would be the most straightforward and statistically powerful approach to make binary comparisons between conditions. That approach is difficult to implement across a large panel of conditions, however, and it is not amenable to an incremental inclusion of new experiments into an existing database. Therefore to compare fitness between conditions we first calculate F and T versus Time 0 in all conditions, and then compare F and T directly between conditions. We calculate relative fitness by taking the difference in fitness scores in the two conditions. We then calculate T between conditions by estimating the variance we would observe in the direct comparison by adding the variance observed in both original comparisons to Time 0. As each of those variances was a conservative estimation, this approach tends to inflate variance and conservatively bias the relative T-statistic. Thus working directly with F and T versus Time 0 to compare conditions is the most flexible, but often not the most sensitive approach. In the case of *MET16* illustrated in **Figure 2—figure supplement 2**, we calculate the relative F and T between supplemented and non-supplemented media as F = −1.3 and T = −0.8. Because these respective samples were seeded from the same Time 0 culture, however, we can also compare counts directly between the two conditions and find F = −1.6 and T = −3.8. This reduced sensitivity is less consequential for *ARG5*, for which relative F and T are −4.3 and −16.0, respectively, but recalculating directly from the actual counts we find F = −4.5 and T = −21.4. Regardless of how a direct comparison is calculated, it is only interpretable if cultures are grown the same number of generations in both conditions. If samples have been grown a different number of generations, the best approach to synthesize data across multiple conditions is to subject fitness scores to clustering analysis as in **Figure 2C**.

## T-statistics assume a normal distribution

An important assumption in our analysis is that T, our metric of consistency, should assume the standard normal distribution when samples with no true abundance differences are compared. We analyzed the Time 0 replicates from our initial auxotrophy experiment treating two replicates as mock samples from an experimental condition and found that the resulting distribution of T-statistics was indeed very similar to the normal distribution, albeit with a small number of outliers at the tails (**Figure 2—figure supplement 3A**). This minor deviation was largely eliminated when we analyzed a mock experiment with biological replication by shuffling Time 0 samples between three independent experiments (**Figure 2—figure supplement 3B**).

## T-statistics have a small bias towards longer, GC rich genes

Genes with |T| > 3 for any condition versus Time 0 had a length distribution similar to that of the total genome, with a slight bias towards longer genes (**Figure 2—figure supplement 3C**). This bias is reflective of the tendency of smaller genes to receive fewer insertions in a randomly mutagenized mutant pool. There is a stronger bias against large T-statistics for genes with fewer than four barcodes contributing to fitness scores (**Figure 2—figure supplement 3D**), and genes with fewer than ~30 counts total across all barcodes in a typical sample (**Figure 2—figure supplement 3E**). There was also a slight over-representation of genes with greater than average GC content among genes with |T| > 3 (**Figure 2—figure supplement 3F**). It is unclear if this bias towards higher GC content is the result of a technical artifact, a reflection of lower GC content of pseudogenes and other mis-annotated features, or

evidence that genes with less impact on fitness in most conditions are under less stringent selection for high GC content in *R. toruloides*. Fitness scores are less sensitive to gene length than T-statistics (*Figure 2—figure supplement 3G*). Given that a gene meets our threshold of |T| > 3, the magnitude of the fitness score is the best measure of biological importance in a given condition.

## Methionine and Arginine biosynthesis in *R. toruloides*

Our fitness data were consistent with established models of arginine biosynthesis in *S. cerevisiae*, as well as cysteine and methionine synthesis in *A. nidulans* (*Figure 2D–E*). Out of 13 genes required to produce methionine from sulfate, one gene (*MET7*) was essential in mutant construction conditions, 10 genes (*MET1, MET2, MET3, MET5, MET6, MET10, MET12, MET13, MET14, and GDH1*) had fitness scores less than −1 in non-supplemented conditions and relative T-statistics less than −3 between supplemented and non-supplemented conditions. Eight of nine genes expected to be required for arginine biosynthesis (*ARG1-7, CPA1,* and *CPA2*) also fit these criteria. Mutants for three genes in these pathways (*ARG8, MET8,* and *MET16*) had fitness scores consistent with auxotrophy, but their relative T-statistics between supplemented and non-supplemented conditions were −2.9, −2.7, and −0.8 respectively. Discordant fitness scores between individual mutant strains drove the relatively weak T-statistics for *ARG8* and *MET16* (see *Figure 2—figure supplement 2* for a full breakdown of strain fitness scores contributing to the *MET16* scores). Both *ARG8* and *MET16* had T < −3 in non-supplemented conditions versus Time 0 and fitness scores that clustered with the other genes their respective pathways. We only mapped two insertions in *MET8* with sufficient abundance for BarSeq analysis. Genes with very few insertions tend to have greater estimated variance in fitness scores and thus have T-statistics with smaller absolute values. The fitness scores for these two insertions in *MET8* were not inconsistent with its expected function in methionine synthesis, however. We also noted that though the transulfuration pathway and *MET17* were dispensable on non-supplemented media, *RTO4_15248* and *RTO4_12031* (orthologs of *A. nidulans cysA* and *cysB*) were required for robust growth, suggesting sulfur uptake occurs primarily through cysteine. The mitochondrial ornithine transporter *ORT1* was also required for arginine prototrophy, but *AGC1* was not, suggesting alternative routes for glutamate transport.

## K-means clusters of fitness scores on fatty acids

Cluster FA1 consists of 21 genes for which mutants had consistent growth defects across all three fatty acids. These genes included three mitochondrial beta-oxidation enzymes; the acyl-CoA dehydrogenase *RTO4_14070* (ortholog of *Homo sapiens ACADSB*), the enoyl-CoA hydratase *RTO4_14805* (ortholog of *H. sapiens ECHS1*), and the hydroxyacyl-CoA dehydrogenase *RTO4_11203* (ortholog of *H. sapiens HADH*). Also included were the electron transfer flavoprotein subunit *AIM45* and the electron transfer flavoprotein dehydrogenase *CIR2*, likely reflective of their known roles as an electron acceptor for acyl-CoA dehydrogenases (*Izai et al., 1992*). The carnitine O-acetyltransferase *CAT2* (involved in fatty-acyl-CoA transfer in the mitochondria [*Strijbis et al., 2010*]) and *PEX11* (involved in peroxisome division and possibly interaction between peroxisomes and mitochondria [*Mattiazzi Ušaj et al., 2015*]) were also in cluster FA1. Rounding out cluster FA1 were nine genes with likely roles in gluconeogenesis, glucose homoeostasis and/or growth on non-preferred carbon sources (*FBP1, RTO4_14162* (ortholog of *ICL1*), *MLS1, GLG1, MRK1, SNF1, SNF3, SNF4, RTO4_11412* (similar to *SWI1*); two genes involved in mitochondrial amino acid metabolism (*PUT2* and *AGC1*); the peroxidase *RTO4_10811* (ortholog of *CCP1*); and *RTO4_12955*, a LYRM domain-containing protein with likely roles in mitochondrial electron transport (*Angerer, 2013*).

Clusters FA2 through FA7 were comprised of 108 genes for which mutants had stronger fitness defects on one or two fatty acids, primarily genes with stronger defects on methylricinoleic acid and ricinoleic acids (FA2 and FA3, 55 genes) or on ricinoleic acid only (FA4 and FA5, 30 genes). These clusters were comprised of genes with predicted roles in various aspects of cellular homeostasis including amino acid metabolism, glycogen

metabolism, phospholipid metabolism, protein glycosylation, the mitochondrial electron transport chain, and 17 genes with no well-characterized homologs. See *Supplementary file 2* for a complete list. Clusters FA2 and FA7 also included 10 genes predicted to play direct roles in peroxisomal beta-oxidation, however. Cluster FA2 (stronger defect on methylricinoleic and ricinoleic acid) included *RTO4_10408* (ortholog of *H. sapiens ACAD11*), *RTO4_14567* (similar to *H. sapiens ACAD11*), acyl-CoA oxidase *RTO4_12742* (ortholog of *POX1*), and *RTO4_8673* (similar to *PEX11*). Cluster FA7 (stronger defect on oleic acid) included 3-ketoacyl-CoA thiolase *RTO4_13813* (ortholog of *POT1*), enoyl-CoA hydratase *RTO4_11907* (ortholog of *H. sapiens ECH1*), 3-hydroxyacyl-CoA dehydrogenase/enoyl-CoA hydratase *FOX2*, predicted acyl-CoA dehydrogenase *RTO4_8963*, and peroxisomal signal receptors *PEX7 and RTO4_13505* (similar to *PEX5*).

## BODIPY 493/503 and buoyancy as measures of lipid content in *R. toruloides*

Under carbon-replete growth conditions in which nitrogen, sulfur, or phosphorus are limiting, *R. toruloides* accumulates up to 70% of its dry weight in neutral lipids (*Wu et al., 2011*; *Wu et al., 2010*; *Wiebe et al., 2012*; *Li et al., 2007*). These lipids are stored as triacylglycerides (TAG) in specialized organelles called lipid droplets (reviewed in [*Walther and Farese, 2012*; *Farese and Walther, 2009*; *Fujimoto and Parton, 2011*]) (*R. toruloides* lipid droplets visualized in *Figure 4A*). BODIPY staining has been used extensively to label lipids and we find that in *R. toruloides* cultures, average cellular BODIPY signal correlates well with total fatty acid methyl ester content as quantified using gas chromatography with flame ionization detection (*Figure 4—figure supplement 1*). Because lipid droplets have lower density than most cell components, as cells accumulate large lipid droplets, they become more buoyant (*Figure 4—figure supplement 2*).

## Normalization and statistical treatment of BODIPY measurements on targeted deletion strains

As we had no a priori estimate of how enrichment scores in FACS and buoyancy experiments would transfer to effect sizes on BODIPY signal in pure culture, we took a pragmatic approach to our validation experiments. We found that average BODIPY signal for a given strain could vary by as much as 2-fold between experiments on different days. Variation was typically lower between biological replicates performed on the same day, however, with standard deviation typically around 10–20% of the average signal (see *Supplementary file 2* for raw BODIPY averages for all samples). To minimize these batch effects, we normalized each culture's average BODIPY signal to the BODIPY signal averaged across three biological replicates of the parental *YKU70Δ* strain grown and processed at the same time as the given culture of a targeted deletion mutant. For each mutant we processed at least six biological replicates cultured in at least two different experiments (cultured and processed on different days). A post-hoc power analysis of this strategy using published power analysis software (*Faul et al., 2007*) calculated that for a six-sample comparison with our observed variance, a one-sided T-test with a significance threshold of p=0.05 should have a 95% confidence of detecting a 1.6-fold difference in BODIPY signal. Increasing sample size to 9 or 12 replicates would lower this detection threshold to 1.5-fold or 1.4-fold, respectively see *Supplementary file 2*). While the considerable variation we observed between cultures limits the sensitivity of our analysis, our candidate strains had sufficiently strong phenotypes such that 18 of our 21 strains from our high confidence clusters of enrichment scores nonetheless had significantly altered BODIPY signal.

# Detailed summary of mutants with altered lipid accumulation in *R. toruloides*

## Mutants with increased lipid accumulation

Mutants in several homologs of known signaling genes had increased lipid accumulation, depicted in *Figure 6* under 'G Protein Switches', 'Kinases and Phosphatases', and 'Gene Expression'. Three GTPases, a GTPase-activating protein (GAP) and two guanine nucleotide exchange factors (GEFs) were in cluster LA1, along with two orthologs of *BMH1*. BMH1 is a 14-3-3 family protein, involved in G protein signaling, the RAS/MAPK signaling cascade, and many other processes (*Roberts et al., 1997*; *Gelperin et al., 1995*). The genes encoding calcineurin complex were also in this cluster as was another protein phosphatase and two protein kinases. Four genes with predicted roles in histone modification were included in cluster LA1 along with three transcription factors and the RNA splicing factor *CBC2*, which is involved in mRNA processing and degradation (*Das et al., 2000*).

Mutants in ten genes with likely roles in protein modification, protein trafficking or other processes in the ER and Golgi led to increased lipid accumulation (*Figure 6*). These genes included three cargo adapter proteins, GPI anchor modifying protein *BST1*, the GTPase *RAS1* (which has been implicated in regulation of vesicular trafficking), and three probable glycosyltransferases. These results show that protein trafficking plays an important role in lipid accumulation in *R. toruloides*, as has been shown in other systems (*Gao and Goodman, 2015*; *Beller et al., 2008*; *Zappa et al., 2017*), though different ensembles of trafficking proteins may be involved in different species.

Disruption of sulfur assimilation also increased lipid accumulation, with five genes involved in sulfate conversion to sulfide clustering in LA1. The cysteine synthase *cysB* was also in this cluster, though *cysB*Δ mutants did not have significantly increased lipid accumulation in our flow cytometry assay. A *MET14*Δ mutant had significantly increased lipid content as expected (*Figure 5B*). In general, the sulfate assimilation mutants had reduced growth in low nitrogen conditions, as indicated by negative fitness scores for pre-enrichment control samples (*Supplementary file 2*). As expected, the auxotrophic mutants identified in our supplementation experiments also had compromised growth in low nitrogen conditions, though the phenotype was generally less severe, likely reflective of slower growth of the population generally. However, slower growth due to auxotrophy was not predictive of higher enrichment scores even for *MET2, MET6, MET12, and MET13*, which are required for methionine synthesis but not sulfate incorporation through cysteine (*Figure 2—figure supplement 2A*). These data suggest that cysteine or intermediate sulfur compounds in the assimilation of sulfate to sulfide may be involved in regulation of lipid accumulation.

## Mutants with decreased lipid accumulation

We found evidence that tRNA thiolation plays a role in lipid accumulation in *R. toruloides*. Enrichment scores for six genes known to be important in the thiolation of tRNA wobble residues (*Huang et al., 2008*) clustered together in LA7. Though these mutants also had apparent buoyancy phenotypes on YPD, two deletion strains (*NCS6*Δ and *NCS2*Δ) had reduced lipid content in pure culture (*Figure 5C*). Furthermore, we observed that for orthologs of *S. cerevisiae* genes with measured tRNA thiolation levels (*Huang et al., 2008*), a decrease in tRNA thiolation corresponded to a lower enrichment score (*Figure 5—figure supplement 1*). Modification of tRNA wobble positions has been implicated in regulation of gene expression in response to heat shock (*Damon et al., 2015*) and sulfur availability (*Laxman et al., 2013*). Our observations suggest that in *R. toruloides* the refactoring of the proteome for efficient lipid accumulation requires fully functional tRNA thiolation. The role that tRNA thiolation plays in this metabolic transition is unclear and deserves more detailed study.

Efficient lipid accumulation also required the regulatory action of orthologs to the *H. sapiens* GTPase *Rab6* and the guanine nucleotide exchange factor *RGP1*, nine protein kinases, three phosphatases or their binding partners. These genes are likely involved in signaling

pathways mediating nutrient state. They include four genes with orthologs implicated in the regulation of glucose and glycogen metabolism (*VHS1*, *HRK1*, *GLC7* and *KIN1*) and four genes with orthologs involved in regulation of nitrogen catabolism (*PPH3*, *PSY2*, *SCH9*, and *ATG1*).

Mutants in nine core components of autophagy were deficient for lipid accumulation. The vacuolar protease *PRB1* and *SIS1* (chaperone mediating protein delivery to the proteasome) were also required for efficient lipid accumulation, as were six genes implicated in protein ubiquitination (*Table 2*). Ubiquitination can affect many aspects of gene function, but likely most of these genes participate in regulation of proteolysis. These results show that autophagy and recycling of cellular components are important for efficient lipid accumulation in *R. toruloides* and provide direct genetic evidence for a previous observation that chemical inhibition of autophagy using 3-methyladenine reduced lipid accumulation in the oleaginous yeast *Y. lipolytica* (*Qiao et al., 2015*).

While most genes encoding enzymatic steps in fatty acid and TAG biosynthesis had too few insertions to calculate reliable enrichment scores (many are probable essential genes, see *Supplementary file 1*), mutants in six genes with predicted function in TAG synthesis resulted in lower lipid accumulation (see *Figure 6—figure supplement 1*). Three of these genes directly mediated reactions in TAG synthesis: *RTO4_12154*, *RTO4_11043*, and *DGA1*. *RTO4_12154* is one of two *R. toruloides GPD1* orthologs predicted to convert dihydroxyacetone phosphate (DHAP) into glycerol-3-phosphate (G3P) (*Overkamp et al., 2000*). *RTO4_11043* is a distant homolog of *H. sapiens BSCL2* (seipin), which modulates the activity of G3P acyltransferase in nascent lipid droplets (*Pagac et al., 2016*). *DGA1* catalyzes conversion of diacylglyceride into TAG (*Sorger and Daum, 2002*). Three more genes were more peripherally involved in TAG biosynthesis: *ACS1*, *YEF1*, and *GUT2*. *ACS1*, which encodes acetyl-CoA synthetase (*De Virgilio et al., 1992*), may supplement production of cytosolic acetyl-CoA from acetate. *YEF1*, encodes an NADH kinase that converts cytosolic NADH to NADPH (*Shi et al., 2005*). *GUT2* converts G3P to DHAP and participates in the G3P shuttle for transfer of electrons from cytosolic NADH to mitochondrial NADH (*Overkamp et al., 2000*). Conversely, mutations in *NDE1* (encoding an alternative enzyme for cytosol/mitochondrial NADH exchange and known to affect activity of Gut2 [*Påhlman et al., 2002*]) had an apparent increase in lipid accumulation. In sum, our fitness data are consistent with the known importance of the precursors acetyl-CoA, G3P, and NADPH for TAG biosynthesis. However, the interactions of NADH transfer and glycerol metabolism in *R. toruloides* deserve more detailed study, as our results stand in contrast to observations in *Y. lipolytica* that *GUT2* mutants had increased lipid accumulation (*Dulermo and Nicaud, 2011*).

*RTO4_16381*, a distant homolog of *H. sapiens PLIN1* (perilipin), was also essential for high lipid accumulation, consistent with its homolog's known roles in lipid body maintenance and regulation of triglyceride hydrolysis (*Bickel et al., 2009*). Our data are in accordance with previous observations that protein RTO4_16381 (previously named Lpd1) localized to lipid droplets in *R. toruloides* and that a GFP fusion construct localized to lipid droplets when heterologously expressed in *S. cerevisiae* (*Zhu et al., 2015*). RTO4_16381 is depicted as localized to the lipid droplet in *Figure 6*, along with 11 other lipid droplet-associated proteins with high confidence lipid accumulation phenotypes or consistent fitness defects on fatty acids. The products of these genes were observed in proteomic analysis of *R. toruloides* lipid droplets by Zhu et al. (*Zhu et al., 2015*), except for RTO4_11043 (similar to human *BSCL2*) and DGA1 which have been localized to the lipid droplet in many other species (*Pagac et al., 2016*; *Salo et al., 2016*; *Grillitsch et al., 2011*; *Athenstaedt et al., 2006*).

## Genes affecting cytosolic NADPH concentrations

*Rhodosporidium toruloides* has two predicted malic enzymes, *RTO4_12761* and *RTO4_13917*, which could theoretically provide NADPH for fatty acid synthesis. Their specificities for NAD+ versus NADP+, are unknown but *RTO4_12761* is more closely related to the NADP-specific malic enzyme from *Mucor circinelloides* (*Zhang et al., 2007b*) and Zhu et al. measured increased protein for *RTO4_12761* in nitrogen-limited conditions (*Zhu et al., 2012*). Neither gene had high confidence enrichment scores in our lipid accumulation assays. We mapped

very low insertion density in the major enzymes of the pentose phosphate pathway (the primary source for NADPH in *Y. lipolytica* [*Wasylenko et al., 2015*]) in our pool, suggesting it was essential in our library construction conditions. As such, the primary source of NADPH in *R. toruloides* remains unconfirmed. Our data are consistent with recent predictions from a simplified metabolic model for *R. toruloides* that during lipid production from glucose, the pentose phosphate pathway should account for greater metabolic flux and NADPH production than malic enzyme (*Bommareddy et al., 2015*).

*YEF1* may also increase the supply of NADPH by phosphorylation of NADH, but presumably this reaction could only play a significant role in fatty acid synthesis if NADP+ is efficiently converted to NAD+ for reduction by NAD(+)-dependent enzymes. NADPH phosphatase activity has been observed for inositol monophosphatases of archaea (*Fukuda et al., 2007*), but these activities have not been well explored in fungal species. Alternatively, *YEF1* may be required for efficient lipid accumulation simply because in its absence the total cytosolic NADPH concentration is too low for efficient fatty acid synthesis, regardless of the balance between NADP+ and NADPH.

## Detailed discussion of processes affecting lipid accumulation in *R. toruloides*

### Organelle interactions and protein localization

Long regarded as essentially inert spheres of lipid, eukaryotic lipid droplets have of late come to be recognized as complex, dynamic, organelles with unique proteomic content and regulated interaction with other organelles (*Walther and Farese, 2012*; *Farese and Walther, 2009*; *Gao and Goodman, 2015*). In animal cells, seipin (*H. sapiens* BSCL2) is thought to mediate lipid droplet nucleation from the ER (*Wang et al., 2016*; *Szymanski et al., 2007*). The *BSCL2* homolog *SEI1* was found to have conserved function in *S. cerevisiae* and *H. sapiens* BSCL2 functionally complemented a *SEI1Δ* mutant. Cells with abnormally small and abnormally large lipid droplets were also reported in an *SEI1Δ* mutant (*Salo et al., 2016*). We found evidence that the closest *R. toruloides* homolog for *BSCL2* (*RTO4_11043*) has conserved function, as deletion mutants had quantitatively lower TAG content (as measured by flow cytometry) and qualitatively more cell-to-cell variation in lipid droplet sizes (by microscopy) than control strains. Perilipins (*H. sapiens* PLIN1-5) act as gatekeepers to the lipid droplet, regulating access by lipases (*Bickel et al., 2009*) and possibly mediating interaction with mitochondria (*Mason and Watt, 2015*). Accordingly, we found that mutants for an *R. toruloides* perilipin homolog had reduced lipid accumulation. Protein trafficking between the ER and Golgi has been implicated in lipid droplet accumulation in *D. melanogaster*, specifically COPI retrograde transport is necessary to limit storage in lipid droplets (*Beller et al., 2008*). We found that disruption of *ERP1, ERP2, EMP24*, or *BST1* (implicated in ER to Golgi transport in COPII vesicles [*Castillon et al., 2011*; *Tanaka et al., 2004*]) led to increased lipid accumulation. It is unclear at this time if COPI and COPII have different functions in lipid body formation across different eukaryotes, or if differing components of ER to Golgi trafficking are more critical when lipids are synthesized de novo from glucose or incorporated from exogenous fatty acids (Beller et al. [*Beller et al., 2008*] cultured *D. melanogaster* cells on oleic acid to maximize lipid droplet size). Increased lipid accumulation in mutants with defective COPII trafficking might also be a function of impaired protein quality control (*Fujita et al., 2006*). *H. sapiens* DNAJC3 is implicated in regulation of the unfolded protein response by controlling elongation factor two phosphorylation (*van Huizen et al., 2003*). The *DNAJC3* ortholog *RTO4_14088* was a high confidence candidate for decreased accumulation as well. These data are consistent with a hypothesis that interaction between protein sorting, quality control and the unfolded protein response play a role in regulating lipid accumulation through modulation of protein translation. Alternatively, delivery of specific proteins to the lipid droplet via the vesicular trafficking system may be critical to lipid droplet growth and maintenance, or the effects of mutations in the endomembrane network on the lipid droplet may arise from redirection of carbon flux through membrane lipids.

## G protein and kinase signaling cascades

We identified 28 genes with high-confidence roles in lipid metabolism that are homologous to genes implicated in G protein–coupled kinase signaling cascades, including Rac, Ras and Rab family G proteins. Rab GTPases are implicated in several aspects of vesicular traffic (*Hutagalung and Novick, 2011*) and are also thought to mediate droplet fusion and interaction with endosomes (*Gao and Goodman, 2015*). Several Rab family members have been identified in lipid droplets in *R. toruloides, S. cerevisiae, D. melanogaster*, and mammals, though their functional roles there remain unclear. 14-3-3 family proteins are known to affect several cellular processes (*Wilson et al., 2016*) including protein trafficking (*Bajaj Pahuja et al., 2015*) and modulate activity of both G proteins (*Riou et al., 2013*) and kinases (*Roberts et al., 1997*). Rac and Ras G proteins have diverse roles in regulating the actin cytoskeleton, cell proliferation, cell cycle progression and polarity (*Goitre et al., 2014*) and tend to localize to cell membranes, interacting with lipid kinases and transmembrane receptors (*Campa et al., 2015*). Likely both Rac1 and Ras1 interact directly with the lipid body, as Rac1 was detected in *R. toruloides* lipid droplets during nitrogen starvation (*Zhu et al., 2015*) and the Ras1 ortholog Ras85D was detected in *D. melanogaster* lipid droplets (*Krahmer et al., 2013b*). We were unable to quantify fitness scores for *RHO1*, but that G protein was also found associated with lipid droplets in *R. toruloides* (*Zhu et al., 2015*) and *S. cerevisiae* (*Bouchez et al., 2015*). Undoubtedly these G proteins and downstream kinases function in a complex network of specific interactions, likely with considerable rearrangement of interactions from those observed in other species (*Choi et al., 2015*; *Nikolaou et al., 2009*; *Hagiwara et al., 2016*). Mapping these signaling networks in *R. toruloides* will require significant effort, but deep regulatory understanding will likely be required to truly optimize engineered pathways in any oleaginous yeast.

## Autophagy and protein turnover

In mammalian and fungal cells, inhibition of autophagy has been reported to both decrease (*Rambold et al., 2015*; *Singh et al., 2009a*; *Shibata et al., 2009*; *Shibata et al., 2010*) and increase (*Singh et al., 2009b*; *Ouimet et al., 2011*) lipid content. These discrepancies may be reflective of competing roles in fatty acid mobilization from lipid droplets and lipid droplet biogenesis, with different processes dominating in different cell types and under different conditions. Mechanisms of fatty acid mobilization have been proposed involving a macroautophagy-like process called lipophagy (*Singh et al., 2009b*; *Ouimet et al., 2011*), a microautophagy-like process (microlipophagy) (*van Zutphen et al., 2014*; *Seo et al., 2017*), and autophagy-independent lipolysis (*Rambold et al., 2015*; *Dupont et al., 2014*; *Maeda et al., 2017*). Why autophagy might be necessary for lipid droplet biogenesis is less clear, but autophagy-dependent recycling of membrane lipids to the lipid body has been demonstrated in mouse hepatocytes (*Rambold et al., 2015*). Conversely, autophagy was also inhibited when TAG hydrolysis was impaired in HeLa cells (*Dupont et al., 2014*) and when TAG synthesis or hydrolysis was blocked in *S. cerevisiae* (*Shpilka et al., 2015*) suggesting that these processes influence each other in a bi-directional manner. In both *Y. lipolytica* and *R. toruloides* several autophagy genes were transcriptionally induced under nitrogen starvation, coincident with lipid accumulation (*Zhu et al., 2012*; *Qiao et al., 2015*). Further, in *Y. lipolytica*, chemical inhibition of autophagy strongly reduced lipid accumulation (*Qiao et al., 2015*). In *S. cerevisiae* deletion of *ATG8* reduced lipid content, but that effect was lipolysis-dependent and *ATG3, ATG4,* and *ATG7* mutants were unchanged in lipid content (*Maeda et al., 2017*).

Our findings demonstrated that autophagy was required for robust lipid accumulation in *R. toruloides*. While we cannot rule out a more direct role in lipid droplet growth and maintenance, a simple theory for this requirement is that autophagy is required for extensive recycling of cellular resources during lipid accumulation. Not only were several core components of autophagy necessary, but also the vacuolar proteases, and several proteins with predicted function in ubiquitination of proteins for proteosomal degradation. The

methylcitrate cycle was required for robust lipid accumulation, which may be reflective of its proposed role in threonine recycling (*Luttik et al., 2000*) or metabolism of propionyl-CoA from released odd-chained fatty acids (*Tabuchi and Serizawa, 2014*). How and why the role of autophagy in lipid droplet development varies by species and condition remains an open question, but *R. toruloides* is an attractive species in which to explore and answer those questions.

## Amino acid biosynthesis and lipid accumulation

We also noted that disruption of several amino acid biosynthesis genes, particularly genes involved in sulfate assimilation into cysteine led to increased lipid production. These data are consistent with the repression of amino acid biosynthesis genes observed in *R. toruloides* (*Zhu et al., 2012*) and other oleaginous fungi (*Kerkhoven et al., 2016*) in nutrient limited conditions. Notably, mutants for genes involved in methionine biosynthesis but not required for sulfate assimilation did not have enrichment scores reflective of increased lipid accumulation, nor did several arginine biosynthesis genes, or other auxotrophic mutants such as insertions in *PHA2* or *ADE5*. Mutants for *ARG1* had higher lipid content, but other mutants in the arginine pathway either had mixed results between the buoyancy and FACS assays (*ARG5* and *ARG7*), T-statistics below our thresholds (*ARG2, ARG7,* and *ARG8*) or showed no sign of increased lipid content (*CPA1, CPA2,* and *ARG3*). These discrepancies suggest that the increased lipid accumulation observed for some mutants may not be simply attributable to redirection of carbon flux from amino acid biosynthesis, but might be the result of active regulation in response to specific amino acids or metabolic intermediates. The transcriptional and proteomic response during nitrogen limitation in these mutants warrants deeper study.

## tRNA thiolation, protein expression and carbon flux in nutrient limited conditions

Post-translational modification of tRNAs has long been known to be critical for efficient protein translation in general (*Agris et al., 2007*), but in recent years thiolation of the U34 base on tRNAs for lysine (UUU), glycine (UUG), and glutamate (UUC) has been recognized to play an important role in fungal metabolic regulation generally (*Zinshteyn and Gilbert, 2013*) and particularly in response to stress such as nutrient limitation (*Laxman et al., 2013*; *Hopper and Phizicky, 2003*) and heat shock (*Damon et al., 2015*). In *S. cerevisiae*, defects in tRNA thiolation significantly alter protein expression for a large number of genes, but the mechanism of that change is disputed. Both transcriptional (*Zinshteyn and Gilbert, 2013*) and translational (*Laxman et al., 2013*) mechanisms have been proposed. A commonality in these studies, however, is the altered expression of genes related to amino acid biosynthesis, protein expression and carbon metabolism. We found that any disruption in the URM1/ elongator complex or tRNA thiolation process reduced lipid accumulation in our experimental conditions. The dramatic metabolic changes entailed in lipid accumulation under nutrient limitation may make for an informative framework in which to explore the mechanisms by which tRNA thiolation interacts with cellular metabolism.

## Cell-to-cell variation in lipid accumulation

We noted extreme cell-to-cell variation in total lipid content in wild-type and mutant strains. This variation was evident in BODIPY fluorescence intensities that varied over at least an order of magnitude within any given sample (*Figure 4—figure supplement 2*) and a wide range of lipid droplet sizes visible in microscopy images (*Figure 7*). Extreme variation in lipid accumulation is typical across eukaryotes, and has emerged as a useful paradigm to explore phenotypic diversity within isogenic populations (*Gocze and Freeman, 1994*; *Herms et al., 2013*; *Krismer et al., 2017*; *Vasdekis et al., 2015*; *Vasdekis et al., 2017*). Our results indicate that *R. toruloides* may make a convenient system to dissect the genetic basis of single-cell phenotypic variation.

