## [Decision Letter]

Thank you for submitting your article "Functional genomics of lipid metabolism in the oleaginous yeast *Rhodosporidium toruloides*" for consideration by *eLife*. Your article has been reviewed by two peer reviewers, and the evaluation has been overseen by a Reviewing Editor and Naama Barkai as the Senior Editor. The reviewers have opted to remain anonymous.

Both reviewers were generally positive about the paper. They have also discussed the reviews with one another and agreed that the paper requires some substantial revisions as detailed in their reports.

*Reviewer #1:*

In this manuscript the authors adapted random barcoded random transposon insertion methods to the oleaginous yeast *Rhodosporidium toruloides*, enabling the functional inference into 100s - 1000s of genes. Insert density was high (1 insert per 100bp) and fairly even across the genome, though there were some biases (regions of low insertion rates). The authors were convincing in the power of this method, and inferred the function of 1337 putatively essential genes, including 150 genes affecting lipid accumulation, of which 35 were validated through targeted deletion assays. The research appears to have been conducted with rigor and the manuscript is well written, figures are appropriate, etc.

I found this research quite exciting and do not see any flaws or major issues with this research. The only suggestions I have would be to shorten the text (Discussion is six pages, for instance), as it is quite long and in some places repetitive. Also, there are many advantages to working with yeast. The authors should discuss how well this approach would transfer to non-yeast fungi, which account for the vast diversity of Fungi and differ in biology, genome size and intron distribution. While the authors were able to dissect the lipid metabolism and importance of different processes (e.g. autophagy) and pathways for accumulation, I would have liked to have seen the use of this information to increase the size and abundance of lipid droplets, and yields in *Rhodosporidium*, obvious next steps.

*Reviewer #2:*

The paper entitled "Functional genomics lipid metabolism in the oleaginous yeast" represents a comprehensive random insertion mutagenesis effort in a relatively uncharacterized basidiomycete yeast. The successful construct of this library allows high resolution parallel genomic fitness analysis, whereby relative changes in mutant strain abundance in response to perturbation provide insight into gene function. Oleaginous yeast are of high interest as they are an attractive host for the production of sustainable chemicals and diesel-like fuels. In addition, understanding the unique accumulation of triacylglycerides (TAG) observed in this organism may be of relevance to lipid storage disorders in human. Here, re-annotation of the *R. toruloides* genome, combined with a molecular twist on the now classic insertional mutagenesis approach pioneered in *Saccharomyces cerevisiae* is deployed to address two important biological questions; the identification of genes that are i) essential for growth (in the conditions tested) and ii) involved in the unusual lipid metabolism of this organism.

Before embarking on creating a random insertional mutation library, the authors first greatly improved the existing genome assembly using standard approaches including long read scaffolds and paired-end sequencing of messenger RNA to guide gene annotation. By modifying bar-coded transposon sequencing methodologies, a library of insertional mutations was constructed by taking advantage of established methods of *Agrobacterium tumefasciens* T-DNA mediated transformation to overcome the low transformation efficiency of *R. toruloides*. All random mutagenesis techniques used for library construction suffer from insertional biases that can confound interpretation. T-DNA insertions exhibit a preference for intergenic regions, can include multiple insertion events, and have been shown to cause local mutations and inversions among others. Despite these issues, through careful tracking and detailed characterization of T-DNA biases, the authors successfully mapped 293,163 barcodes (from a total of ~2 million) to useable T-DNA insertions that were well-dispersed throughout the genome. These mutants represented >90% of nuclear encoded genes for use in parallel downstream fitness assays. In practice however, barcodes were sequenced on average to a depth of ~20 million counts per fitness assay. This precluded detection of ~40,000 mapped barcodes due to low counts and combined with constraining fitness measurements only to insertions that landed in central regions of the coding sequence. These constraints reduced the number of insertions that could be accurately assessed for fitness to 68,021; representing 6,588 genes and ~20% of the original total mapped barcodes. Counts per barcode ranged from 1 to 1000, with a mode of ~10.

The authors then turned to applying their insertion library to identify genes required for fitness in various conditions. As a proof of principle, synthetic minimal media (YNB) was supplemented with arginine and, in a different experiment, methionine was used and at least in one case compared to YNB alone. This is an ideal feasibility test, as any gene required for biosynthesis of these amino acids cannot grow in YNB alone, and therefore exhibit a maximum decrease in fitness. Though the authors did not address this issue, this experiment would be useful in defining the dynamic range and establishing the sensitivity of the assay. When fitness differences were striking when YNB was compared to YNB + the required amino acid the results were striking. Nearly all members required for arginine and methionine were identified, although there were a handful of exceptions. It seems that further study of the specific reason for these false negatives was a missed opportunity to learn about unforeseen issues inherent to the methodology. It is also noteworthy that the results were much less striking when the different fitness conditions were compared to the standard T_0_ control, the metric of fitness that the authors argued as being the preferred method for accurate quantification of changes in fitness. For this reason, why this analysis method was chosen over the other was not specifically addressed, a general finding echoed in other fitness experiments is throughout the manuscript. This concern is addressed more fully in the specific comments.

Additional fitness tests included growth on three fatty acids as the sole carbon source; 129 genes were identified with significant fitness defect scores that included genes involved in fatty acid oxidation, gluconeogenesis and mitochondrial amino acid metabolism. Many of these genes were consistent with those known to be required for fatty acid oxidation in other species. Importantly, these findings were validated by constructing targeted deletion mutants in ~10 of these genes and measuring growth on the various carbon sources.

To identify genes involved in lipid accumulation, the library was fractionated using two measures of lipid content – buoyancy separation and neutral-lipid staining. In this case, instead of using the initial library as the reference, genes involved in lipid accumulation were identified by comparing the abundance of each mutant in high and low lipid fraction and looking for strains that had the biggest differences measured in the two conditions compared to the control. Subsequent clustering of the data reveal enrichment in biological functions. Select strains were again validated; in this case finding a greater number of inconsistencies between the two assays as well as a greater number of false positives. The authors therefore defined more stringent criteria to correct this; requiring gene mutations identified as exhibiting increased or decreased lipid content to be consistent between both the buoyancy and staining lipid content assays. The rationale behind using their genomic toolset to understand lipid metabolism and biogenesis in this particular fungus is clear, but the results need to be put in context of what has been observed in similar assays in the published literature i.e. the large amount of information regarding particular pathways available from yeasts such as *Saccharomyces, Schizosaccharomyces, Candida* and *Coccidioides*. For example, the role of Sulphur metabolism in lipid biosynthesis and biogenesis would seem to be well served by a more extensive comparison to the data that has been collected and published with these other model systems. Indeed, it is unclear what, if anything, was specifically novel to *R. toruloides*.

Nonetheless, the morphological phenotypes observed were fascinating and would benefit from a more thorough treatment and a more in-depth comparison between these observations. Indeed, one of the key contributions represented by this thorough study is the power that it provides to compare and contrast these findings with findings observed in these other systems or by other methods.

In summary, the resource and library provided by this study will be extremely useful in further defining and characterizing the genetics and metabolic pathways unique to *R. toruloides*. Importantly, the study supports other work suggesting that existence mitochondrial beta-oxidation is widespread in fungi. The involvement of both peroxisomal and mitochondrial beta-oxidation has been shown to be important in gluconeogenesis in mammalian cells and has also been linked to altered metabolism in cancer.

Specific comments:

Overall paper is well written and represents a significant contribution to the genomic analysis of another emerging model fungus. However, there were several issues with the statistical analysis that must be addressed.

First, the issue of sufficient coverage in counts per gene needs to be included and discussed as necessary. As mentioned, the author's state fitness was measured fore 68,021 representing 6558 genes. After accounting for constraints including low reads, ~4.6 million counts were available to measure these ~70k mutations, leaving 50 counts per gene. If the counts were evenly distributed, this should be sufficient for fitness measures. We know that this is clearly not the case for sequencing reads as they are typically modeled using a noisy Poisson distribution or by a negative binomial. In the manuscript presented, neither the initial counts nor the final counts per gene are reported, not even including an example. This is a glaring omission: the authors state that the range of counts per gene varied broadly, with a mode of 10. Naively, this would imply that many mutations with significant fitness effects would be identified by ~10 counts. This seems too low for adequate gene coverage, particularly as the variance increases dramatically in this low count range, making significant fitness changes difficult to detect and accurately quantify.

Though the methodology used for the analysis was referenced (PMID: 25968644), the assumptions made in this analysis were not discussed here. Because the authors suggest that the use of the T_0_ sample strengthens quantitative fitness measurements, yet the results in the supplementary files do not seem to reflect this, the analysis section needs to include this discussion.

Overall, I found the statistical analysis difficult to follow and thinly presented – including details such as how the cells were grown. As manuscripts become increasingly packed with massive amounts of data, these sections of the manuscript are critical in order for the reviewer to evaluate the data quality and robustness.

In the supplementary text, fitness scores are described as:

"For each barcoded T-DNA insertion, we calculate the log_2_ ratio of abundance before and after competitive growth in the experimental condition. F is the average of those ratios (weighted by sequence depth) for all the insertions disrupting a given gene. T is a modified student's T-statistic, a measure of statistical significance of F that incorporates consistency between individual insertions across biological replicate cultures."

The authors need to include at least one example of how a gene is modeled by averaging the log_2_(T_0_ /T_after_) from different insertional mutants and include a figure that demonstrates the consistency across biological replicates.

The supplementary data mentions several different metrics and it is difficult to know which is being used in the main text, or why they are all included to begin with. For example in the auxotrophy experiments the results are presented in several different ways, the columns described by:

1) Fitness Scores (averaged between replicates), included 5 comparisons to T_0_

YPD

YNB + DOC

YNB + Arginine

YNB + Methionine

YNB + No Supplement

2) T-like Statistics versus T_0_; T-like test statistics for fitness/enrichment scores above

YPD

YNB + DOC

YNB + Arginine

YNB + Methionine

YNB + No Supplement

3) Fitness differences vs. Control Conditions (averaged between replicates)

DOC vs. No Supplement

Methionine vs. No Supplement

Arginine vs. No Supplement

YPD vs. No Supplement

3) T-like Statistics vs. Control Conditions

DOC vs. No Supplement

Methionine vs. No Supplement

Arginine vs. No Supplement

YPD vs. No Supplement

4) Wilcoxon Signed Rank Tests Multiple Hypothesis Adjusted; Wilcoxon signed rank test between condition and T_0_

No Supplement vs. T_0_

DOC vs. T_0_

Methionine vs. T_0_

Arginine vs, T_0_

DOC vs. No Supplement

Methionine vs. No Supplement

Arginine vs. No Supplement

YPD vs. T_0_

YPD vs. No Supplement

It is not clear which method of analysis was used on which dataset – every experiment should be annotated as such.

This presentation of the data is especially confusing because fitness scores are weighted averages by sequencing depth across all insertions and then averaged to obtain a single score. I can imagine all kinds of scenarios where this could be problematic. For example, for a single gene averaging the log_2_(T_0_ /T_after_) would seem to be vulnerable to over or underestimating the actual fitness – for example when different insertions for the same gene are conflicting in magnitude or even sign. Problems when measuring the relative importance of different genes to each other may also arise for example, if shorter genes are penalized due to having fewer insertions, or may introduce bias due to unequal numbers of insertions associated with each gene and possibly influenced by variance as well. To avoid these issues it would seem necessary to use a metric that corrects or normalizes for these issues.

An additional concern as has already been mentioned is the higher confidence in measuring fitness relative to the T_0_ condition; presumably due to the depth of sequencing and the ability to obtain associated 't-like' statistics. However, there seems to be some logic missing here. For example, in the condition YNB + Arginine, a decrease in fitness compared to T_0_ may be due to 1) slow growth of the strain in any condition 2) slow growth only in YNB 3) slow growth only in arginine.

Although there is an accompanying website, it is clearly in its early stages and no key is provided for explanation of the metrics used in these files.

In Supplementary file 2 multiple scores reported for each gene in each experiment. However, each gene described as a weighted average of all of the insertions for a given gene. The number of independent insertions is not given except to say that there were around 10 inserts/kb for most genes. This raises the possibility that the assumption that genes with fewer than 2 inserts are essential – a key conclusion only briefly discussed – maybe misleading. This section needs to mention the logic and statistics behind this as well as to acknowledge previous work (e.g. PMID: 28481201 which used saturation transposition to identify essential genes). Another point that requires attention is why the variance and confidence in the counts is not addressed (as it is in RNA seq) prior to scoring log ratios by the average weighted counts for the insertions associated with each gene.

This paper relies on an advance (more barcodes) on the well-established barcoded Tn-Seq methodology pioneered by Adam Deutschbauer. It seems unnecessary and not helpful to rebrand the technique with another acronym/abbreviation. The authors need to acknowledge the many Tn-Seq papers in many other model systems that have been successful in characterization of new genomes. Along these same lines, the authors mention that co-fitness analysis will accelerate annotation of new genomes. This is likely correct, but the previously published concept of co-fitness should be elaborated upon and cited.

Returning to the insertional mutagenesis technique and unanswered questions:

Overall, the main difficulties in insertional mutagenesis in comparison to targeted gene deletions are that saturation is difficult as some regions are hotspots, while others are immune. Other problems include the fact, as in this study, that each gene is covered ~10x, yet the nature of the mutation is unknown. This can be partially controlled by limiting insertion sites to those that interrupt coding regions.

If a gene-specific model is the intent for measuring fitness defects by relying on deep sequencing of the time zero sample, several specific issues need to be explained:

How many sequence reads are required for time zero? Shouldn't replicates of time zero be included? What is the supporting statistical test? What is the range of ratios and variance for F?

To end on a more positive note, I do not doubt that the fundamental conclusions of the manuscript are correct. Many if not most of the issues can be addressed by providing all of the data to the reader, preferably in a user-friendly format. However, the conclusions themselves did not obviously follow from presentation of the results, neither for the methodology or the biology. If the intent was instead to focus on novel biological findings using the technology, the biological findings seemed to rely heavily on homology and the presentation of complex model pathways was inappropriate without making clear in the model what is and is not known in other model organisms, or distinguishing what findings were novel and unique to *R. toruloides*.

---

## [Author Response]

Reviewer #1:[…] I found this research quite exciting and do not see any flaws or major issues with this research. The only suggestions I have would be to shorten the text (Discussion is six pages, for instance), as it is quite long and in some places repetitive.

We have condensed the main text considerably in the revised draft. The Results section and Discussion section in particular are now approximately 80% and 50% of their original lengths, though the Materials and methods and Appendix are longer.

Also, there are many advantages to working with yeast. The authors should discuss how well this approach would transfer to non-yeast fungi, which account for the vast diversity of Fungi and differ in biology, genome size and intron distribution.

We added a brief discussion of these issues to the Discussion section.

“In some respects *R. toluroides* was an ideal species to develop these methods. […] RB-TDNAseq can be applied to study the germination of these spores and their growth into nascent, isogenic colonies prior to their fusion into more physiologically and genetically complex networks of mycelia and fruiting bodies.”

While the authors were able to dissect the lipid metabolism and importance of different processes (e.g. autophagy) and pathways for accumulation, I would have liked to have seen the use of this information to increase the size and abundance of lipid droplets, and yields in Rhodosporidium, obvious next steps.

We have strong interest in leveraging these data to engineer greater titer, rate, and yield of lipidbased bioproducts. Such projects are beyond the scope of this work, which also has much broader applications and implications beyond metabolic engineering.

Reviewer #2:[…] Though the authors did not address this issue, this experiment would be useful in defining the dynamic range and establishing the sensitivity of the assay. When fitness differences were striking when YNB was compared to YNB + the required amino acid the results were striking. Nearly all members required for arginine and methionine were identified, although there were a handful of exceptions. It seems that further study of the specific reason for these false negatives was a missed opportunity to learn about unforeseen issues inherent to the methodology.

Our purpose for these experiments was to establish the soundness of the mutant pool, sequencing protocols and analysis methods, and to establish useful statistical cutoffs to focus downstream analysis on high confidence sets of genes with minimal type I error. We expect most readers with an interest in systems biology will be well aware of the inherent trade-off between sensitivity and specificity. To illustrate the various factors adding uncertainty to our fitness scores, we have added the following text to the Appendix detailing why three false negatives in this experiment had T-statistics that fell short of our consistency thresholds (either too few barcodes, or barcodes with discordant data).

“Mutants for three genes in these pathways (*ARG8, MET8*, and *MET16*) had fitness scores consistent with auxotrophy, but their relative T-statistics between supplemented and nonsupplemented conditions were -2.9, -2.7, and -0.8 respectively. […]The fitness scores for these two insertions in *MET8* were not inconsistent with its expected function in methionine synthesis, however.”

We also added a clustering analysis of fitness data across supplementation conditions in Figure 2C, and moved figures summarizing fitness scores for genes in the predicted pathways from a supplemental figure to Figure 2D and E to more clearly show which genes in these pathways are identified with the two approaches. The new figure is described in the Results:

“Alternatively, when we hierarchically clustered the fitness scores for genes with F < -1 and T < -3 versus Time 0 in any supplementation condition (Figure 2C), the resulting clusters included twelve and nine mutants rescued by methionine and arginine respectively; this was a nearly complete recovery of genes with predicted function in this pathway (shown in Figure 2D-E with additional discussion in the appendix).”

*It is also noteworthy that the results were much less striking when the different fitness conditions were compared to the standard* T_0_*control, the metric of fitness that the authors argued as being the preferred method for accurate quantification of changes in fitness. For this reason, why this analysis method was chosen over the other was not specifically addressed, a general finding echoed in other fitness experiments is throughout the manuscript. This concern is addressed more fully in the specific comments.*

*Because the authors suggest that the use of the* T_0_*sample strengthens quantitative fitness measurements, yet the results in the supplementary files do not seem to reflect this, the analysis section needs to include this discussion.*

*An additional concern as has already been mentioned is the higher confidence in measuring fitness relative to the* T_0_*condition; presumably due to the depth of sequencing and the ability to obtain associated 't-like' statistics. However, there seems to be some logic missing here. For example, in the condition YNB + Arginine, a decrease in fitness compared to* T_0_*may be due to 1) slow growth of the strain in any condition 2) slow growth only in YNB 3) slow growth only in arginine.*

It was not our intention to imply that comparing all samples to Time 0 first and then manipulating the resulting fitness scores and T-statistics is necessarily the most accurate and sensitive methodology. We, in fact, adopted this strategy out of a combination of practical considerations in experiment design and the intention to establish an analysis pipeline amenable to incrementally building large databases of fitness data. The flexibility that this approach affords comes with a price in the form of reduced sensitivity in some instances, as the reviewer correctly noted. We have added some discussion of this tradeoff in the main text and the appendix.

Added to the main text:

“Different aliquots of the mutant pool have subtly different starting compositions and experience stochastic variations in the length of lag phase as they recover from frozen stocks. […] Given F and T in two different conditions (F_C1_, T_C1_ and F_C2_, T_C2_), we calculate relative fitness F_C1-C2_ = F_C1_-F_C2_ and relative T-statistics T_C1-C2_ = (F_C1_F_C2_)/sqrt(var(F_C1_) + var(F_C2_)).”

Added to the appendix:

“In most cases, meaningful interpretation of fitness scores will require comparisons amongst several conditions. […] If samples have been grown a different number of generations, the best approach to synthesize data across multiple conditions is to subject fitness scores to clustering analysis as in Figure 2C.”

Specific comments:Overall paper is well written and represents a significant contribution to the genomic analysis of another emerging model fungus. However, there were several issues with the statistical analysis that must be addressed.First, the issue of sufficient coverage in counts per gene needs to be included and discussed as necessary. As mentioned, the author's state fitness was measured fore 68,021 representing 6558 genes. After accounting for constraints including low reads, ~4.6 million counts were available to measure these ~70k mutations, leaving 50 counts per gene. If the counts were evenly distributed, this should be sufficient for fitness measures. We know that this is clearly not the case for sequencing reads as they are typically modeled using a noisy Poisson distribution or by a negative binomial. In the manuscript presented, neither the initial counts nor the final counts per gene are reported, not even including an example. This is a glaring omission: the authors state that the range of counts per gene varied broadly, with a mode of 10. Naively, this would imply that many mutations with significant fitness effects would be identified by ~10 counts. This seems too low for adequate gene coverage, particularly as the variance increases dramatically in this low count range, making significant fitness changes difficult to detect and accurately quantify.

The appendix reports that the typical readsper barcodeused in fitness analysis was 10, not reads per gene. To help clarify this we have added a new figure (Figure 2—figure supplement 1C) showing the distribution of average counts per gene across our experiments and the following text in the:

“This distribution of countable barcodes and sequencing depth translated to approximately 100-250 total usable reads per sample for estimating fitness of mutants in each gene (Figure 2—figure supplement 1C). This depth of usable sequence on a per-gene basis is comparable to the point at which Wetmore et al. found that additional sequence depth gave diminishing returns in terms of reduced variance in fitness scores (Wetmore et al., 2015).”

It should also be noted this is the number of reads per replicate. Thus, with three biological replicates, approximately 300 – 750 reads are present in the Time 0 samples for comparison with the experimental conditions. At this read depth, variation between biological replicate cultures and between individual barcoded insertions in the same gene are a much greater source of uncertainty than sequencing noise.

Overall, I found the statistical analysis difficult to follow and thinly presented – including details such as how the cells were grown. As manuscripts become increasingly packed with massive amounts of data, these sections of the manuscript are critical in order for the reviewer to evaluate the data quality and robustness.

We believe the experimental procedures were described adequately in the Materials and methods section, with several paragraphs devoted to culture conditions generally and procedures for the major experiments specifically. However, the above comments suggest more emphasis should be placed on an important detail of our experimental design: explicit pairing of all samples in a given condition with their own independent Time 0 starter cultures. To this end, we have amended our introduction of BarSeq analysis with auxotrophy experiments as follows:

“To establish if RB-TDNAseq could produce statistically robust results with minimal experimental replication, we recovered three independent starter cultures from frozen aliquots of the mutant pool and used each replicate to inoculate both supplemented and non-supplemented cultures. We grew these cultures for seven generations and measured fitness across the mutant pool with BarSeq.”

We also edited a relevant passage in the methods for better clarity:

“In downstream fitness or enrichment analysis, we explicitly paired each sample from an experimental condition with the Time 0 sample from the starter culture replicate from which it was seeded.”

If the reviewers or editors are unclear about any other specific details of our experimental procedures we would be happy to address them.

In the supplementary text, fitness scores are described as:"For each barcoded T-DNA insertion, we calculate the log_2_ ratio of abundance before and after competitive growth in the experimental condition. F is the average of those ratios (weighted by sequence depth) for all the insertions disrupting a given gene. T is a modified student's T-statistic, a measure of statistical significance of F that incorporates consistency between individual insertions across biological replicate cultures."

*The authors need to include at least one example of how a gene is modeled by averaging the log_2_(*T_0_*/T_after_) from different insertional mutants and include a figure that demonstrates the consistency across biological replicates.*

Another point that requires attention is why the variance and confidence in the counts is not addressed (as it is in RNA seq) prior to scoring log ratios by the average weighted counts for the insertions associated with each gene.Though the methodology used for the analysis was referenced (PMID: 25968644), the assumptions made in this analysis were not discussed here.If a gene-specific model is the intent for measuring fitness defects by relying on deep sequencing of the time zero sample, several specific issues need to be explained:How many sequence reads are required for time zero? Shouldn't replicates of time zero be included?What is the supporting statistical test? What is the range of ratios and variance for F?

As we did not develop any significant new analytical methods for processing BarSeq data, our aim was to focus the reader’s attention on the more novel aspects of this study: the implementation of random barcoded insertion mutagenesis in non-model fungi, insights into lipid metabolism in an oleaginous yeast of industrial interest, and conserved themes in lipid droplet biology across diverse eukaryotic species. But we take the reviewers point that we may have given the details of the analysis somewhat short shrift. While we think it inappropriate to completely rehash all the details of the BarSeq analysis, we have added some additional detail in the Materials and methods section.

“Briefly, for each biological replicate and condition, for any barcode with an average of at least 3 counts in Time 0 samples, a strain fitness score is calculated as F_strain_ = log_2_(C_condition_ + sqrt(P)) – log_2_(C_Time0_ + 1/sqrt(P)), where C is the raw counts for the barcode and P is a gene-specific “pseudocount” added to reduce noise in fitness scores for low-count strains. […]In general genes with data from only one or two barcodes had smaller T-statistics and thus were filtered out in later analyses.”

We have also added a new section to the appendix contrasting two examples of how data from multiple insertions are combined for a gene fitness score: 1) A case of very coherent data from many insertions and 2) a case with fewer insertions with conflicting fitness scores. This section is supported by a new supplemental figure (Figure 2—figure supplement 2), which displays the raw counts and fitness scores for the individual strains in each replicate.

“Contributions of individual strains to gene-level fitness scores

Examples of how fitness scores from individual barcoded insertion strains are combined to calculate gene-level fitness scores are shown in Figure 2—figure supplement 2.[…]However, any further global restriction on the barcodes analyzed per gene would compromise the data for other genes with fewer insertions.”

In addition to the individual examples described above we also note that Figure 3, Figure 3—figure supplement 1, and Figure 5 in the original submission all display gene fitness or enrichment scores for individual biological replicates demonstrating the consistency of these metrics.

The supplementary data mentions several different metrics and it is difficult to know which is being used in the main text, or why they are all included to begin with. For example in the auxotrophy experiments the results are presented in several different ways, the columns described by:*1) Fitness Scores (averaged between replicates), included 5 comparisons to* T_0_YPDYNB + DOCYNB + ArginineYNB + MethionineYNB + No Supplement*2) T-like Statistics versus* T_0_*; T-like test statistics for fitness/enrichment scores above YPD YNB + DOC YNB + Arginine YNB + Methionine YNB + No Supplement 3) Fitness differences vs. Control Conditions (averaged between replicates) DOC vs. No Supplement Methionine vs. No Supplement Arginine vs. No Supplement YPD vs. No Supplement 3) T-like Statistics vs. Control Conditions DOC vs. No Supplement Methionine vs. No Supplement Arginine vs. No Supplement YPD vs. No Supplement 4) Wilcoxon Signed Rank Tests Multiple Hypothesis Adjusted; Wilcoxon signed rank test between condition and* T_0_*No Supplement vs.* T_0_*DOC vs.* T_0_*Methionine vs.* T_0_*Arginine vs,* T_0_DOC vs. No SupplementMethionine vs. No SupplementArginine vs. No Supplement*YPD vs.* T_0_YPD vs. No SupplementIt is not clear which method of analysis was used on which dataset – every experiment should be annotated as such.

The *Rhodosporidium/Rhodotorula* community has relatively little systems data available to it at the moment, and we expect many researchers will want access to as much data as possible on genes of interest to them. For this reason we tried to be as transparent as possible and include all the BarSeq data we gathered in two accessible formats: as human readable, searchable, and programmatically accessible supplemental data tables; and as an interactive website, which has now been linked to the genome browser at the Joint Genome Institute. Example link: https://genome.jgi.doe.gov/cgi-bin/dispGeneModel?db=Rhoto_IFO0880_4&id=14070

For readers simply trying to dive into the experiments as presented a bit more deeply, we can see that it might be a bit difficult to extract the relevant data from the comprehensive summary. To make it easier to access the data for that purpose, we have included three new supplemental tables in Supplementary file 2, with the fitness scores that were clustered in Figure 2C, Figure 3—figure supplement 1, and Figure 5A. The supplementary tables also include the criteria by which these genes were selected (e.g. average relative fitness scores and combined relative Tstatistics on fatty acids versus glucose).

We have also removed the Wilcoxon signed rank tests on the fitness scores from the manuscript. They were offered as an alternative, very conservative measure of significance for the fitness scores that was based on fewer assumptions but they weren’t essential to understanding any of the experiments.

This presentation of the data is especially confusing because fitness scores are weighted averages by sequencing depth across all insertions and then averaged to obtain a single score. I can imagine all kinds of scenarios where this could be problematic. For example, for a single gene averaging the log_2_(T_0_ /T_after_) would seem to be vulnerable to over or underestimating the actual fitness – for example when different insertions for the same gene are conflicting in magnitude or even sign.

Indeed barcoded insertions in the same gene with differing phenotypes are common, as we would expect due to secondary mutations in the mutant pool, as discussed in the appendix. The central problem of any such analysis is to aggregate the data from several mutant strains, while minimizing the error introduced by outlier strains to accurately infer gene function. As this is not a new problem, we adapted the proven methods of Wetmore et al. to aggregate data from individual mutations in the same gene. To more clearly frame this challenge and to explain the strengths of Wetmore et al.’s approach, we added the following paragraph to the results:

“Secondary mutations are prevalent even in well-curated mutant collections (Comyn, Flibotte and Mayor, 2017) and ATMT can introduce several types of confounding mutations (see Appendix 1 for detail). […] All fitness scores and T-statistics (combined across biological replicates) are available in Supplementary file 2 and online in a dynamic fitness browser, adapted from (Price et al., 2016): http://fungalfit.genomics.lbl.gov/.”

Problems when measuring the relative importance of different genes to each other may also arise for example, if shorter genes are penalized due to having fewer insertions, or may introduce bias due to unequal numbers of insertions associated with each gene and possibly influenced by variance as well. To avoid these issues it would seem necessary to use a metric that corrects or normalizes for these issues.

The T-like statistics are indeed somewhat sensitive to gene length. Longer genes tend to have both more insertions and more total counts available for BarSeq analysis. Thus we have greater confidence in the metrics for longer genes. Note, however, that these biases result in a relatively small enrichment of longer genes amongst genes with consistent fitness scores (see Figure 2—figure supplement 3C). In general, we use the T-statistics primarily to identify genes that have consistent, reliable fitness scores in a given condition and we subsequently focus on the fitness scores as the appropriate metric to compare effects between genes. In our revised manuscript we have added more detail to the appendix discussing these biases in T-statistics and highlighting the preferred comparison of fitness scores (as opposed to T-statistics) to compare importance of different genes in a given condition:

“T-statistics have a small bias towards longer, GC rich genes

Genes with |T| > 3 for any condition versus Time 0 had a length distribution similar to that of the total genome, with a slight bias towards longer genes (Figure 2—figure supplement 3C). […] Given that a gene meets our threshold of |T| > 3, the magnitude of the fitness score is the best measure of biological importance in a given condition.”

Although there is an accompanying website, it is clearly in its early stages and no key is provided for explanation of the metrics used in these files.

The fitness browser will have increasing functionality as we add more data, particularly for analyzing co-fitness profiles between genes. The site includes a “help” section with an overview of the fitness scores and T-statistics. The information on individual conditions is necessarily minimal, but of course the relevant conditions to this manuscript are more fully described here. Once again we should reiterate that the fitness browser is cited as one of several ways to access the data, including the supplemental tables, the raw sequencing data at the NCBI short read archive and the analysis software on a third party software distribution service (bitbucket.com).

In Supplementary file 2 multiple scores reported for each gene in each experiment. However, each gene described as a weighted average of all of the insertions for a given gene. The number of independent insertions is not given except to say that there were around 10 inserts/kb for most genes.

We have added information on the total number inserts mapped in each gene and the number of inserts with sufficient counts to contribute to fitness analysis at this sequencing depth to the relevant tables in Supplementary file 2.

This raises the possibility that the assumption that genes with fewer than 2 inserts are essential – a key conclusion only briefly discussed – maybe misleading. This section needs to mention the logic and statistics behind this as well as to acknowledge previous work (e.g. PMID: 28481201 which used saturation transposition to identify essential genes).

We agree there is significant uncertainty in our provisional identification of essential genes. The list was generated from the straightforward observation that, on the gene level, insertion of TDNAs does appear to be close to truly random and the distribution of insertions per kilobase across genes reflects that, with the notable exception of a group of genes that also happen to be orthologous to known essential genes. The presence of two distinct populations of genes in the histogram shown in Figure 2B is clear. We have edited the presentation of this data in the Results section to make this logic a bit more clear, and more clearly acknowledge some uncertainty in the designation of all these genes as essentials:

“Insertions were sufficiently well dispersed to map at least one T-DNA in 93% of nuclear genes, despite some local and fine-scale biases in insertion rates (see Appendix 1 for details). […] Based on the above criterion, we identified 1337 probable essential genes, which we report in Supplementary file 1.”

We have also edited the Discussion to more clearly highlight the recent use of transposon mutagenesis in a higher resolution application of the same principle in *S. cerevisiae* (PMID: 28481201, and similar analyses in bacteria):

“We found that genes recalcitrant to T-DNA insertion were highly enriched for orthologs to known essential genes, suggesting that most genes with very low insertion rates were likely essential in our mutagenesis conditions. […]We hope the provisional list of essential genes identified here will serve as a useful resource for genetics in *R. toruloides* and relatedspecies.”

This paper relies on an advance (more barcodes) on the well-established barcoded Tn-Seq methodology pioneered by Adam Deutschbauer. It seems unnecessary and not helpful to rebrand the technique with another acronym/abbreviation.

When presenting these experiments to different audiences we’ve found that a distinction in terms between RB-TnSeq and RB-TDNAseq helps clarify the fundamental differences between the methods. While functionally equivalent to transposon mutagenesis in many respects, ATMT is more prone to introduction of more complex concatamers of the marker sequence and we find it useful to remind the audience of those differences. Also the use of TnSeq in describing an experiment that does not utilize transposons can cause some confusion.

The authors need to acknowledge the many Tn-Seq papers in many other model systems that have been successful in characterization of new genomes.

We have revised the Introduction to more clearly introduce the concept of TnSeq and distinguish the relative advantages of RB-TnSeq vs. TnSeq. The revised text includes a few more references to successful TnSeq studies and a comprehensive review. We think it better frames the context in which RB-TDNAseq may enable more widespread application of functional genomics in fungi.

“Fitness analysis of gene deletion or disruption mutants within pooled populations is a flexible, powerful approach for elucidating gene function.[…] The methods they employed were only viable for characterization of a small pool of highly enriched mutants, but they demonstrated an effective paradigm to bring high-throughput functional genomics to diverse fungi.”